# IKKε isoform switching governs the immune response against EV71 infection

Ya-Ling Chang[1,2], Yu-Wen Liao[1,2], Min-Hsuan Chen[2], Sui-Yuan Chang[1,3], Yao-Ting Huang[4 ✉], Bing-Ching Ho[2 ✉] & Sung-Liang Yu [1,2,3,5,6,7 ✉]

The reciprocal interactions between pathogens and hosts are complicated and profound. A comprehensive understanding of these interactions is essential for developing effective therapies against infectious diseases. Interferon responses induced upon virus infection are critical for establishing host antiviral innate immunity. Here, we provide a molecular mechanism wherein isoform switching of the host IKKε gene, an interferon-associated molecule, leads to alterations in IFN production during EV71 infection. We found that IKKε isoform 2 (IKKε v2) is upregulated while IKKε v1 is downregulated in EV71 infection. IKKε v2 interacts with IRF7 and promotes IRF7 activation through phosphorylation and translocation of IRF7 in the presence of ubiquitin, by which the expression of IFNβ and ISGs is elicited and virus propagation is attenuated. We also identified that IKKε v2 is activated via K63-linked ubiquitination. Our results suggest that host cells induce IKKε isoform switching and result in IFN production against EV71 infection. This finding highlights a gene regulatory mechanism in pathogen-host interactions and provides a potential strategy for establishing host first-line defense against pathogens.

[1] Department of Clinical Laboratory Sciences and Medical Biotechnology, College of Medicine, National Taiwan University, Taipei, Taiwan. [2] Centers of Genomic and Precision Medicine, National Taiwan University, Taipei, Taiwan. [3] Department of Laboratory Medicine, National Taiwan University Hospital, Taipei, Taiwan. [4] Department of Computer Science and Information Engineering, National Chung Cheng University, Chia-Yi, Taiwan. [5] Institute of Medical Device and Imaging, College of Medicine, National Taiwan University, Taipei, Taiwan. [6] Graduate Institute of Pathology, College of Medicine, National Taiwan University, Taipei, Taiwan. [7] Graduate Institute of Clinical Medicine, National Taiwan University College of Medicine, Taipei, Taiwan. ✉email: ythuang@cs.ccu.edu.tw; f94424002@gmail.com; slyu@ntu.edu.tw

Enterovirus 71 (EV71), a member of the enterovirus genus of the *Picornaviridae* family, is a nonenveloped virion encapsulating positive-strand RNAs. EV71 infection commonly causes hand, foot, and mouth disease (HFMD) in infants and young children and leads to aseptic meningitis, encephalomyelitis, acute flaccid paralysis or even neurologic and psychiatric disorders in severe subjects[1,2]. Like most RNA viruses, EV71 evolves evasion mechanisms to evade host immune attacks[3,4]. Type I interferons (IFNs) are documented as the first-line immune responses against viral infections[5,6]. Previously, we demonstrated that a cellular microRNA (miRNA), miR-146a, induced in EV71 infection inhibits type I IFN production by targeting IRAK1 and TRAF6[3]. Activation of the host innate immune response begins with recognition of pathogen-associated molecular patterns (PAMPs) by pattern recognition receptors (PRRs)[5,6]. Toll-like receptors (TLRs) and RIG-I (retinoic acid-inducible gene I)-like receptors (RLRs), two key members of PRRs, trigger the production of proinflammatory cytokines and IFNs in concert with a set of transcription factors, including nuclear factor-κB (NF-κB) and IFN regulatory factors (IRFs)[7]. The RLR family consists of three members, including RIG-I, melanoma differentiation-associated gene 5 (MDA5), and laboratory of genetics and physiology 2 (LGP2). RIG-1 and MDA5 have been identified as intracellular PRRs for RNA viruses to stimulate type-I IFN expression[8,9]. Upon RNA ligand binding, RIG-I and MDA5 interact with the mitochondrial antiviral-signaling adaptor protein (MAVS) to trigger downstream I-Kappa-B Kinase Epsilon (IKKε)/TANK binding kinase 1 (TBK1) and canonical NF-κB signaling for activation of IFN-β and inflammatory cytokines, respectively[10,11]. IKKε and TBK1, so-called noncanonical IKKs or IKK-related kinases, play critical roles in innate immunity by inducing type I IFNs[7]. Both IKKε and TBK1 enable phosphorylation of the C-terminal Ser/Thr rich regions of IRF3 and IRF7. In turn, IRF3 and IRF7 translocate into the nucleus and induce type I IFNs[7,12,13]. In addition to IFN production, recent studies have demonstrated a predominant role of IKKε in inducing a subset of IKKε-dependent IFN-stimulated genes (ISGs) by using *Ikbke*[−/−] mouse embryonic fibroblasts (MEFs)[14–16]. Type I IFN-dependent ISGs such as ISG56 and ISG54 were decreased in *Ikbke*[−/−] MEFs compared with *Ikbke*[+/+] MEFs[15,16]. However, the underlying mechanisms of IKKε activation are poorly understood and the regions of the IKKε coding DNA sequence (CDS) that are required for downstream signaling should be characterized.

## Results

To comprehensively determine the gene regulation network in EV71 infection, the whole transcriptomes were analyzed by using next-generation sequencing (NGS) technology. RD cells, a human rhabdomyosarcoma cell line, were infected with EV71, and RNA transcriptomic profiles at the indicated hours postinfection (h.p.i.) were explored. A total of 1035 differentially expressed genes in response to EV71 infection were identified by Cufflinks[17] (Supplementary data 1) and further applied to gene network analysis by MetaCore software (version 6.24.67895). The results showed that EV71 infection activated several signaling pathways, including the immune response, cellular apoptosis, cell cycle, and inflammation (Supplementary data 2), and the altered pathways were similar to those in previous reports[18–21].

**IKKε presents isoform switching in EV71 infection**. With NGS massive sequencing, not only gene expression but also alternative splicing forms can be analyzed[22,23]. Hence, we used RNA transcriptomic profiling to address whether gene isoform switching occurred and played a role in EV71 infection. The isoforms were first retrieved using Cufflinks[24], and only genes with two or more assembled isoforms were selected for further analysis. Two or more isoforms derived from a gene were considered to represent isoform switching if the proportion of one isoform was upregulated while that of the other was downregulated in EV71 infection. After filtration, 242 genes with isoform switching in response to EV71 infection were identified and further used for signaling pathway analysis (Supplementary data 3). The pathways enriched by these isoform switching genes were annotated using Kyoto Encyclopedia of Genes and Genomes (KEGG), and four signaling pathways were significantly enriched ($p$ value <0.05, Supplementary Table 1). Of these enriched pathways, the Toll-like receptor signaling pathway was ranked first ($p$ value = 0.021). Six genes belonging to this pathway, *IRAK4*, *AKT1*, *IKKε*, *MYD88*, *IRF3,* and *SPP1* showed isoform switching following EV71 infection (Supplementary Fig. 1a and Supplementary Table 1). Previous reports indicated that IKKε could promote IFN production to establish immune defense by phosphorylating IRF3 and IRF7[12,13,25–28]. Hence, we hypothesized that host cells might take control of IKKε isoform switching to establish an immune defense mechanism by regulating IFN expression. In Fig. 1a, the read depths of each IKKε exon in the mock infection group as well as at 4 and 8 h.p.i. are shown. A higher depth in IKKε exon 20 than in exon 19 was detected in the mock infection group, while a lower depth in IKKε exon 20 than in exon 19 was observed at 4 and 8 h.p.i., suggesting that IKKε isoform switching took place at 4 and 8 h.p.i. compared with mock infection (Fig. 1a). The IKKε gene possesses three isoforms: isoform 1 (IKKε v1, NM_014002; NP_054721) contains a full-length coding DNA sequence (CDS), while isoform 2 (IKKε v2, NM_001193322; NP_001180251) and isoform 3 (IKKε v3, NM_001193321; NP_001180250) lack exon 20 and exon 3 due to alternative splicing, respectively. Regarding functions, IKKε v2 is defective in the coiled-coil domain, which is important for protein–protein interactions, whereas IKKε v3 is deficient in the kinase domain (Supplementary Fig. 1b). We then validated IKKε isoform switching by digital PCR. Three primer/probe sets were used in ddPCR assay and the expression of each IKKε isoform was calculated (Supplementary Methods, Supplementary Table 2). The results, consistent with the RNA transcriptomic findings (Fig. 1a, b), indicated that IKKε v2 is upregulated while IKKε v1 is downregulated in pace with EV71 infection (Fig. 1b and Supplementary Table 3).

To understand whether IKKε isoform switching is happened in different EV71-infected cell lines, first SH-SY5Y cells, a neuroblastoma cell line, were infected with EV71 at 5 m.o.i. for single virus infection cycle and RNAs were extracted at 12 h.p.i. and 24 h.p.i.[29]. The IKKε isoforms were measured by digital PCR and the results showed upregulated IKKε v2 and downregulated IKKε v1 expression in SH-SY5Y cells during EV71 infection (Fig. 1b). Furthermore, we investigated whether IKKε isoform switching is a common characteristic in virus infections including RNA and DNA viruses. The expression of IKKε isoforms was measured in HeLa cells infected with coxsackievirus B3 (CVB3) at 5 m.o.i. for 4 h.p.i. and 6 h.p.i.[30], and herpes simplex virus-1 (HSV-1) at 1 m.o.i. for 8 h.p.i. and 24 h.p.i.[31] by RNA-Seq. The relative expression of IKKε v2 is increased while IKKε v1 is decreased both in CVB3 and HSV-1 infection (Supplementary Fig. 1c). CVB3 and HSV-1 infections induced IKKε isoform switching in a similar pattern found in EV71 infection. These data indicated that isoform switching is a common feature during virus infection at least in the cases of EV71, CVB3, and HSV-1.

To address whether the IKKε isoform switching can be detected in protein level, we performed a Click-iT AHA assay to detect nascent IKKε isoforms. Biotin-labeled newly synthesized proteins at indicated time points postinfection were purified, and

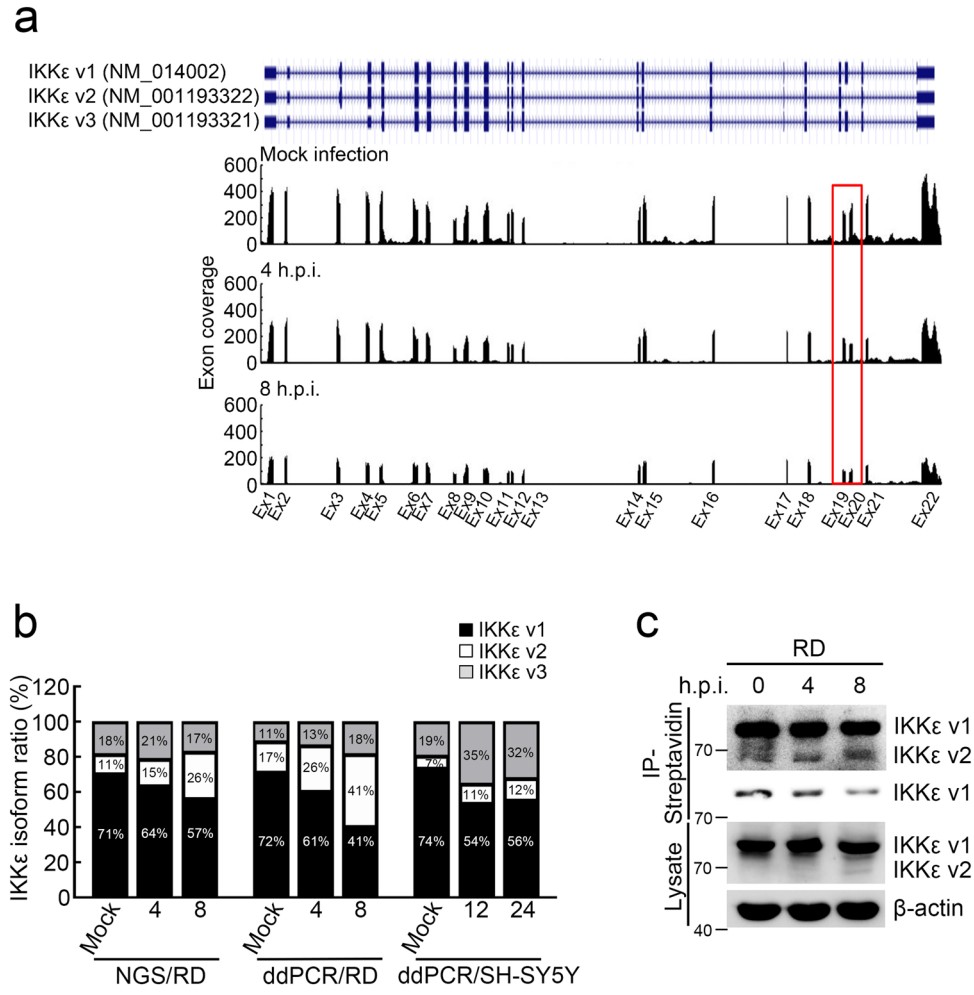

**Fig. 1 EV71 infection triggers IKKɛ isoform switching. a** IKKɛ is located at NC_000001.11 (206470243..206496890) of chromosome 1 in GRCh38. Each exon coverage of IKKɛ is illustrated and the isoform switching occurs in exon 20 at 4 and 8 h.p.i. compared with mock infection group. **b** IKKɛ isoform switching is validated by droplet digital PCR (ddPCR). IKKɛ v2 was upregulated while IKKɛ v1 was downregulated in response to EV71 infection determined by NGS (left panel for RD cells) and ddPCR (middle panel for RD cells and right panel for SH-SY5Y cells). The proportion of each IKKɛ isoform was indicated. **c** IKKɛ isoform switching is confirmed by western blotting. The Click-iT AHA assay was performed to measure newly synthesized IKKɛ v1 and v2. The synthesis of IKKɛ v2 was increased while IKKɛ v1 was decreased in EV71 infection. The upper panel of IP-Streptavidin is a long exposure while the lower panel is a short exposure.

IKKɛ isoforms were detected by Western blotting. The expression of IKKɛ v2 increased while the v1 decreased after EV71 infection, consistent with our RNA data (Fig. 1c).

**IKKɛ v2 attenuates EV71 propagation via IRF7-mediated IFNβ and ISG induction.** RIG-I and MDA5 can recognize viral RNAs and then initiate the signaling pathways leading to IKKɛ activation. IKKɛ plays a crucial role in evoking IFN production, mainly by activating IRF3/7[7,32]. In this light, we first investigated the effect of different IKKɛ isoforms on IRF3/7 activation as well as its downstream IFN or cytokine production. Three IKKɛ isoforms were constructed, and each IKKɛ-expressing construct was cotransfected with the IRF3/7-driven promoter vector. As shown in Fig. 2a, IKKɛ v2, but neither IKKɛ v1 nor IKKɛ v3, activated the promoter containing 4 repeats of IRF3/7 binding elements cloned from the IFNβ promoter (4X PRDIII/I) (Fig. 2a). Since both IRF3 and IRF7 are involved in IFNβ promoter activation[7,12,13], we introduced a promoter vector driven by IRF3 or IRF7 to determine the major activator of the IFNβ promoter in IKKɛ v2-mediated IFNβ production. Our results showed that IKKɛ v2 induced the luciferase activity of the IRF7

binding element-containing reporter vector, but not the IRF3 binding element (Fig. 2a), corresponding to the interaction of IRF3/7 and each IKKɛ isoform (Supplementary Fig. 2a). Furthermore, we introduced each IKKɛ isoform into RD cells followed by EV71 infection to evaluate whether IKKɛ v2 could increase IFNβ production upon EV71 infection. As indicated in Fig. 2b and Supplementary Fig. 2d, IFNβ expression was induced across all IKKɛ isoform transfectants and even in the vector control group at 8 h.p.i. compared to the vector control group in mock infection. Importantly, IKKɛ v2 increased IFNβ production up to 8-fold compared with the vector control at 8 h.p.i., even stronger than IKKɛ v1 or IKKɛ v3 isoforms. Taken together, IKKɛ v2 was the most effective regulator among IKKɛ isoforms to accelerate IFNβ production via IRF7 in EV71 infection. Next, we determined whether IKKɛ v2 could efficiently activate downstream IFN-stimulated genes (ISGs) in EV71 infection. It was well demonstrated that the promoter activities, as well as mRNA expression levels of ISG56 and 2′-5′-oligoadenylate synthetase 1 (OAS1), could be strongly induced by IFNs[33–38]. Hence, we measured promoter activities and mRNA expression of such ISGs by promoter activity assay and quantitative real-time PCR, respectively. The results showed that both promoter

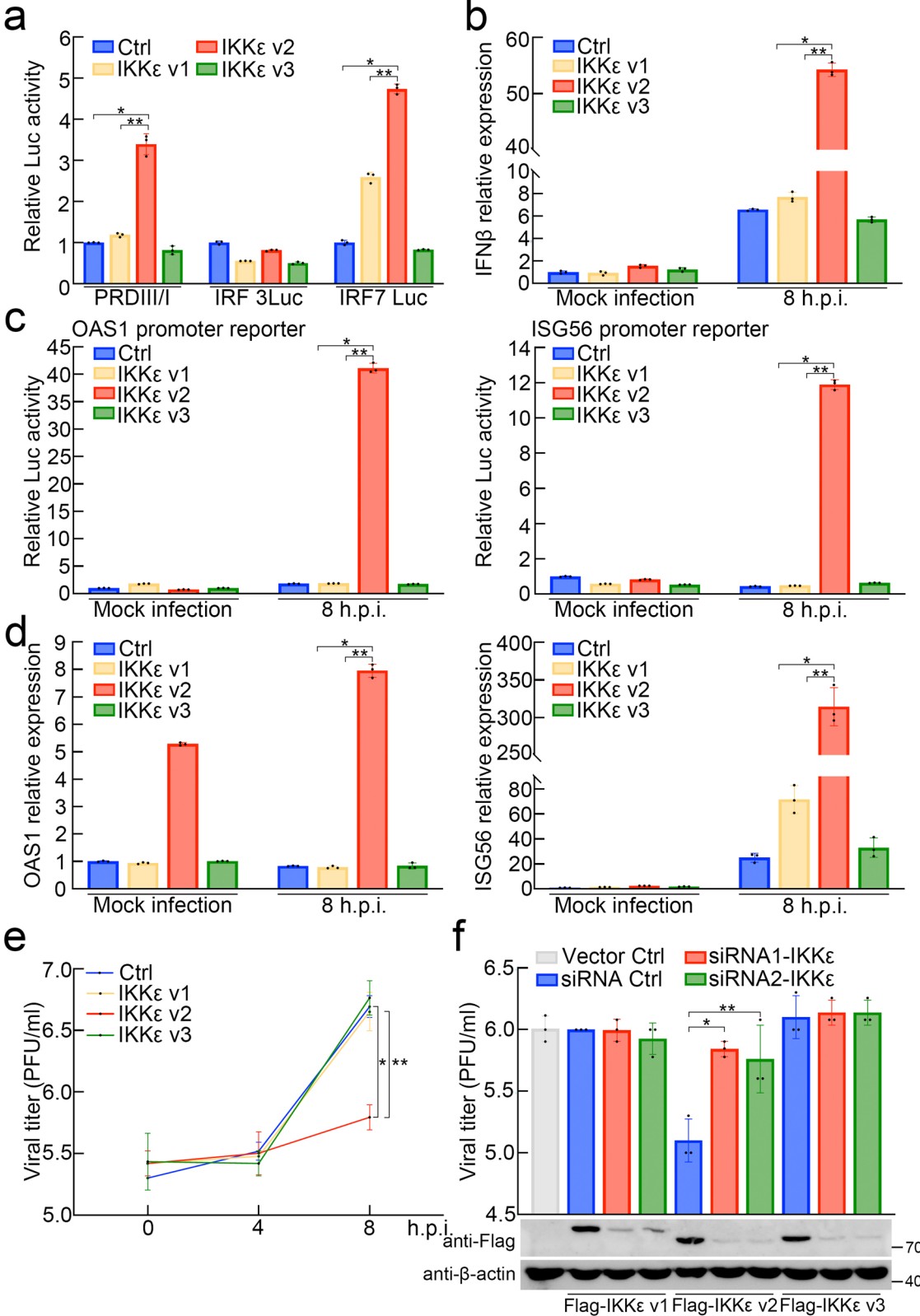

activities and mRNA expression of ISG56 and OAS1 were obviously upregulated in RD cells ectopically expressing IKKε v2 at 8 h.p.i. (Fig. 2c, d and Supplementary Fig. 2d). The other two ISGs, ISG20 and Myxovirus resistance protein A (MxA), were slightly augmented in IKKε v2 transfectants in both promoter activities and mRNA expression at 8 h.p.i. (Supplementary Fig. 2b–d). It has been reported that IFNα plays a major role in

the induction of ISG20 and MxA[39–41]. Additionally, ISG20 was also regulated by IFNγ in endothelial cells[42]. This finding suggests that the maximal expression of ISG20 or MxA might require additional supplements, such as specific stimulators or specific cellular enviornments[39–42]. This notion might partially explain why only mild inductions in ISG20 and MxA expression were observed in our results. The mentioned ISGs are well

**Fig. 2 IKKε v2 increases IRF7-mediated IFNβ and ISGs expressions in EV71 infection and attenuates virus propagation. a** IKKε v2 increases the activity of the promoter harboring IRF7 binding element. HEK293 cells were cotransfected with each IKKε isoform and luciferase reporter vector containing 4 repeats of IRF3/IRF7 binding elements obtained from IFNβ promoter (PRDIII/I), IRF7 binding element (IRF7 Luc) or IRF3 binding element (IRF3 Luc) as indicated. The data are normalized with vector control (Ctrl). All data presented are mean ± SD ($n = 3$). * and ** represent $p$ value <0.05 as compared with Ctrl group and IKKε v1 group, respectively. **b** IKKε v2 enhances IFNβ expression in EV71 infection. RD cells were transfected with each IKKε isoform followed by EV71 infection. IFNβ expression was measured by quantitative real-time PCR and normalized with Ctrl in mock infection group. All data presented are mean ± SD ($n = 3$). * and ** represent $p$ value <0.05 as compared with Ctrl group and IKKε v1 group, respectively. h.p.i., hours of postinfection. **c**, **d** IKKε v2 upregulates promoter activities and expressions of IFNβ-dependent ISGs. Each IKKε isoform was ectopically expressed in RD cells accompanied with the vector harboring indicated ISG promoter. RD cells were infected with EV71 and the luciferase activities (**c**) and the ISG expressions (**d**) were measured at indicated h.p.i.. The data are normalized with Ctrl in mock infection group. All data presented are mean ± SD ($n = 3$). * and ** represent $p$ value <0.05 as compared with Ctrl group and IKKε v1 group, respectively. **e** IKKε v2 attenuates virus propagation. RD cells expressing each IKKε isoform were infected with EV71 and virus productions were determined by plaque assay. All data presented are mean ± SD ($n = 3$). * and ** represent $p$ value <0.05 as compared with Ctrl group and IKKε v1 group, respectively. **f** Attenuation of virus titer in IKKε v2 transfectants is restored by IKKε siRNAs. The siRNAs against IKKε, siRNA1-IKKε or siRNA2-IKKε, were introduced into RD cells expressing each Flag-IKKε isoform followed by EV71 infection. The viral titers and ectopic IKKε isoform expressions were determined by plaque assay and western blotting, respectively. β-actin was served as an internal control. All data presented are mean ± SD ($n = 3$). * and ** represent $p$ value <0.05 as compared with siRNA ctrl group.

known to inhibit virus replication particularly in the case of RNA viruses[43].

Therefore, we scrutinized whether ectopic IKKε v2 could attenuate EV71 propagation. Virus tilters obtained from supernatants of each transfectant ectopically expressing IKKε isoform were measured by plaque assay, and the results exhibited a ten-fold decrease in RD cells expressing IKKε v2 compared with those expressing IKKε v1, IKKε v3 or control groups at 8 h.p.i. (Fig. 2e). In contrast, the attenuation of virus titers observed in IKKε v2 transfectants was greatly eliminated by two IKKε siRNAs, which target two common regions of IKKε isoforms, respectively (Fig. 2f). Collectively, IKKε v2 could induce IRF7-mediated IFNβ and the downstream ISGs that might attenuate EV71 propagation. This response implied that certain IKKε v2-driven mechanisms existed in EV71 infection to drive IFN and ISG production.

**IKKε v2 phosphorylates and activates IRF7 in the presence of ubiquitin.** Previous studies have clearly demonstrated that enterovirus infections promote polyubiquitination[44,45]. Furthermore, ubiquitin has been recognized as a mediator of IKKε signaling[46–48]. Accordingly, we implemented experiments to determine whether ubiquitin participated in IKKε signaling in EV71 infection. Similar to a previous study, the ubiquitination level was dramatically increased in EV71 infection (Supplementary Fig. 3)[45]. We then evaluated IRF7 phosphorylation status and translocation in each IKKε isoform transfectant in the presence of ubiquitin to mimic virus infection conditions. HEK293 cells were transfected with Flag-IRF7 and each V5-IKKε isoform expressing vectors with or without HA-ubiquitin expressing vector (HA-ubi). Flag-IRF7 was then immunoprecipitated and immunoblotted with anti-Flag and anti-phospho-serine antibodies, respectively. We observed that the phosphorylation level of Flag-IRF7 was increased in IKKε v2 transfectants in the presence of ubiquitin compared to IKKε v1 and IKKε v3 transfectants. However, in the absence of ubiquitin, no phosphorylated Flag-IRF7 was detected (Fig. 3a and Supplementary Fig. 4a). To our knowledge, IKKε is one of the major kinases for IRF7 phosphorylation and activation[16]. Ser471/472, located in the regulatory domain of IRF7, is considered a phosphorylation site by IKKε and is characterized as a vital residue in IRF7 activation[49,50]. Hence, we measured the effect of different IKKε isoforms on Ser471/472 phosphorylation of Flag-IRF7. The phosphorylation status of IRF7 Ser471/472 residues was greatly enhanced in IKKε v2 transfectants compared with that in IKKε v1 and IKKε v3 transfectants in the presence of ubiquitin (Fig. 3b and Supplementary Fig. 4b). To characterize the role of IKKε

isoform switching in EV71 infection, we examined the phosphorylation and expression of IRF7 in each ectopically IKKε isoform-expressing RD cells. IKKε v2 strongly induced IRF7 phosphorylation (Fig. 3c), whereas the RNA and protein expression levels of IRF7 were unchanged in each IKKε transfectant during EV71 infection (Supplementary Fig. 4c). Phosphorylated IRF7, in tandem, causes nuclear translocation and further induces type I IFN and ISG expression to establish an antiviral response[49,51,52]. In this light, we hypothesized that IKKε v2 strongly phosphorylated IRF7 and further augmented IRF7 translocation into the nucleus. To test this hypothesis, nucleus-cytoplasm fractionation and immunofluorescence were conducted, and more V5-tagged IRF7 was found to accumulate in the nucleus of cells expressing IKKε v2 than in those expressing IKKε v1 or IKKε v3 in the presence of ubiquitin (Fig. 3d and Supplementary Fig. 4d, f, g). Similarly, IKKε v2 also led to greater nuclear accumulation of endogenous IRF7 in the presence of ubiquitin compared with the other two IKKε isoforms (Fig. 3e and Supplementary Fig. 4e). Taken together, IKKε v2 dominantly phosphorylates and activates IRF7 rather than IKKε v1 and IKKε v3 during EV71 infection in a ubiquitin-dependent manner.

**IRF7 preferentially interacts with IKKε v2 in the presence of ubiquitin.** Having demonstrated that IKKε v2 could phosphorylate and activate IRF7 in the presence of ubiquitin, we next addressed whether IKKε could directly interact with IRF7 by using coimmunoprecipitation experiments and an in vitro pull-down assay. V5-IRF7 was cotransfected with each Flag-IKKε isoform into HEK293 cells in the presence of ubiquitin, and the cell lysates were immunoprecipitated and immunoblotted by anti-Flag and anti-V5 antibodies, respectively. With ubiquitin, the immunoprecipitates derived from IKKε v2 transfectants carried more V5-IRF7 than those from IKKε v1 and IKKε v3 transfectants (Fig. 4a and Supplementary Fig. 5a). Moreover, reciprocal experiments indicated that Flag-IRF7 could vigorously interact with IKKε v2 under ubiquitin expression (Fig. 4b and Supplementary Fig. 5b). We also determined whether endogenous IRF7 preferentially interacts with IKKε v2 in the presence of ubiquitin. The results supported a similar conclusion that IRF7 could more strongly interact with IKKε v2 than IKKε v1 and IKKε v3 in the presence of ubiquitin (Fig. 4c and Supplementary Fig. 5c). Moreover, the in vitro pull-down assay showed that C-terminal mutation of IKKε v2 did not affect the direct interaction between IKKε and IRF7 (Supplementary Fig. 5d). These evidence clearly demonstrated that IRF7 preferentially interacts with IKKε v2 in the presence of ubiquitin.

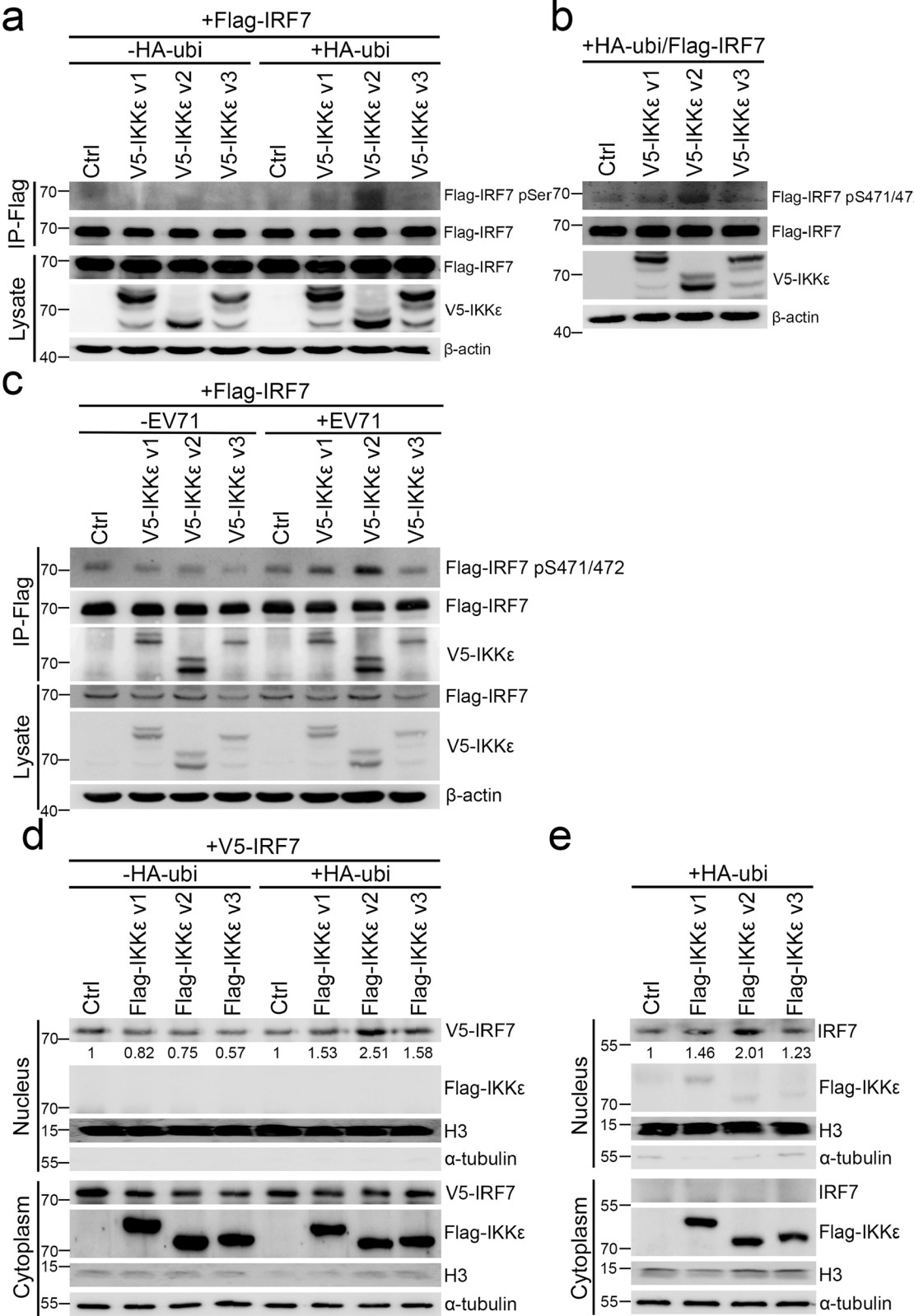

**IKKε v2 is highly phosphorylated via K63-linked ubiquitination**. It has been well demonstrated that IKKε is modified by K63-linked ubiquitin chains and this modification is essential for IKKε kinase activation[46,47]. Hence, we first assessed the effects of ubiquitination on the kinase activity of IKKε isoforms and determined the differences in K63-linked ubiquitin-mediated activation between each IKKε isoform. The phospho-serine level

served as an indicator to measure IKKε activity[53,54]. As shown in Fig. 5a, IKKε v2 showed a higher phospho-serine signal compared to IKKε v1 and IKKε v3 in the absence of ubiquitin. Notably, ubiquitin could obviously increase the phospho-serine level of IKKε v2 and then expand the differences to the other two IKKε isoforms (Fig. 5a and Supplementary Fig. 6a). To further investigate whether K63-linked ubiquitin contributed to IKKε v2

**Fig. 3 IKKε v2 increases IRF7 phosphorylation and IRF7 translocation in the presence of ubiquitin. a, b** IKKε v2 strongly phosphorylates IRF7 in the presence of ubiquitin. Flag-IRF7 and each V5-IKKε isoform were ectopically expressed in HEK293 cells in the presence or absence of ubiquitin. The lysates were subjected to immunoprecipitation with anti-Flag beads and the phosphorylated IRF7 was detected with anti-pSer antibody (**a**). Total lysates obtained from HEK293 cells ectopically expressing each V5-IKKε isoform, Flag-IRF7 and HA-ubi were subjected to immunoblot analysis. The phosphorylated IRF7 was detected by IRF7-phospho-Serine471/472 (IRF7-pS471/472) antibody (**b**). β-actin was served as an internal control. **c** IKKε v2 strongly phosphorylates IRF7 in EV71 infection. Flag-IRF7 and each V5-IKKε isoform were ectopically expressed in RD cells followed by EV71 infection. The Flag-IRF7 was immunoprecipitated with anti-Flag beads and the phosphorylation was detected by IRF7-pS471/472 antibody. The co-immunoprecipitated V5-IKKε isoform was analyzed by anti-V5 antibody. **d, e** IKKε v2 facilitates IRF7 translocation. HeLa cells were cotransfected with each Flag-IKKε isoform and V5-IRF7 in the presence or absence of HA-ubi. Nucleus and cytoplasm fractions obtained from HeLa cells were applied to immunoblot with anti-Flag and anti-V5 antibodies (**d**). Each Flag-IKKε isoform was transfected into HeLa cells along with HA-ubi. The cell lysates were adapted to nucleus and cytoplasm fractionation and immunoblot with anti-Flag and anti-IRF7 antibodies (**e**). H3 and α-tubulin were used as nuclear and cytosolic markers, respectively.

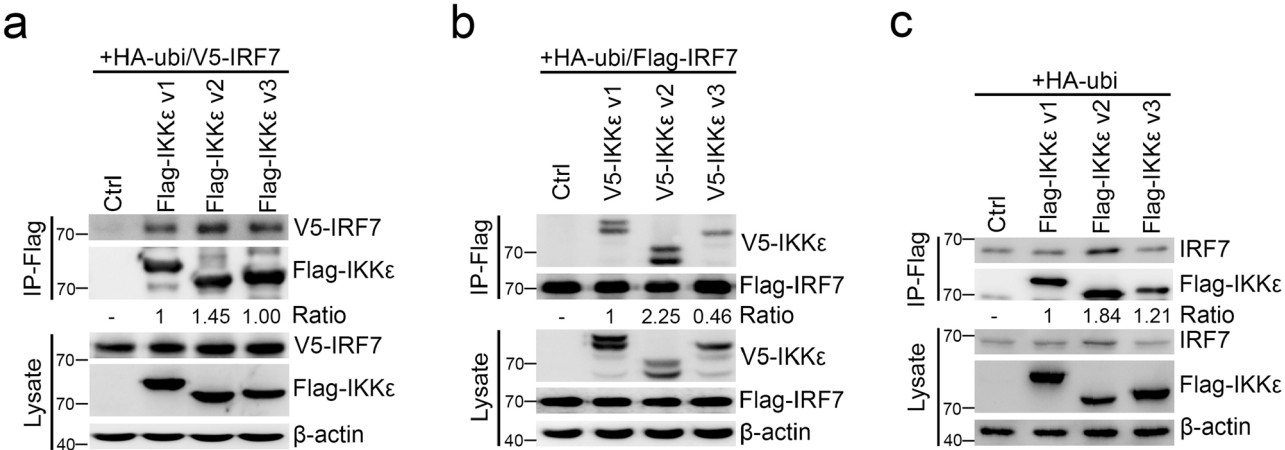

**Fig. 4 IRF7 preferentially interacts with IKKε v2. a, b** IRF7 preferentially interacts with IKKε v2 in the presence of ubiquitin. IRF7 and each IKKε isoform were ectopically expressed in HEK293 cells in the presence of ubiquitin. The lysates were subjected to immunoprecipitation to precipitate Flag-IKKε (**a**) or Flag-IRF7 (**b**) with anti-Flag beads, respectively. V5-IRF7 (**a**) and V5-IKKε isoforms (**b**) were detected by anti-V5 antibody. **c** IKKε v2 strongly interacts with endogenous IRF7 in presence of ubiquitin. RD cells were cotransfected with each Flag-IKKε isoform and HA-ubi. Total lysates were adapted to immunoprecipitation using anti-Flag beads and endogenous IRF7 was detected by anti-IRF7 antibody. β-actin was served as an internal control.

activity, we introduced each Flag-IKKε isoform along with HA-tagged wild-type (WT) ubiquitin or K63R ubiquitin mutant into HEK293 cells and determined the phospho-serine level of each IKKε isoform (Fig. 5b and Supplementary Fig. 6b). We found that the ubiquitin-mediated serine phosphorylation of IKKε v2 was obviously decreased in the case of K63R ubiquitin. The results confirmed that IKKε v2 was robustly phosphorylated compared with IKKε v1 and IKKε v3 under wild-type ubiquitin treatment, but IKKε v2 serine phosphorylation was markedly reduced in the presence of K63R ubiquitin (Fig. 5b and Supplementary Fig. 6b). Wietek et al. and Bulek et al. reported that IKKε activation could be stimulated via autophosphorylation[55,56]. As a result, each Flag-tagged IKKε isoform was purified from cells coexpressing wild-type or K63R ubiquitin and then adapted to an in vitro kinase assay. IKKε v2 phosphorylation was more obvious in wild-type ubiquitin transfectants than in K63R transfectants (Fig. 5c and Supplementary Fig. 6c). These results clearly demonstrated that K63-linked ubiquitination enhanced IKKε v2 phosphorylation. Previous studies indicated that the ubiquitin-like domain could fold back onto the IKKε kinase domain and impair its activity[57].

## Discussion

Type I IFNs provide a first line of defense against viral infections. Administration of IFNs can limit virus spreading at an early phase during virus infections, while many viruses prevent IFN attacks by the inhibition of IFN through several mechanisms. In our study, we demonstrated that IKKε isoform switching occurred in EV71 infection. IKKε v2 phosphorylated and thereby activated IRF7 to trigger IFN activation (Fig. 5d). We also

examined IKKε isoform switching in another enterovirus, CVB3, and in a DNA virus, HSV-1. The relative expression of IKKε v2 was upregulated while IKKε v1 was downregulated, upon CVB3 and HSV-1 infection. The relative abundance of the different IKKε isoforms might represent a regulatory mechanism controlling the innate immune response even though. Further studies are required to comprehensively understand how virus infection induces IKKε isoform switching remains for further investigations.

The innate immune responses provide an early phase defense against viral infections. Generally, virus triggers a cascade of signaling to product type I IFNs and proinflammatory cytokines[58]. However, it is not the case for all of viruses because certain viral infections can subvert cellular IFN induction pathways[59]. For example, influenza viral proteins PB1-F2 and PB2-S1 interact with MAVS to inhibit IFN induction[60,61] and hepatitis C virus cleaves MAVS to interfere IFN production[62]. Moreover, it has been demonstrated that dengue virus can induce IFNβ and ISG production in the brain; in contrast, IFNs are not detectable in infected dendritic cells[63,64]. It suggested that IFN induction is regulated in a cell type-specific manner.

EV71 has been demonstrated to regulate IFNβ induction by affecting the pathways mediated by RIG-1/MDA5 and TLR upon EV71 infection. EV71 3C protease (3C[pro]) was demonstrated to suppress IFN signaling by interrupting the RIG-1-IFN promoter-stimulating factor 1 (IPS1) interaction, and with nucleus translocation of IRF3[65]. Other studies have reported that 3C[pro] degrades RIG-1 and cleaves adaptor TRIF to overcome IFN production in EV71-infected cells[66,67] and EV71 2A

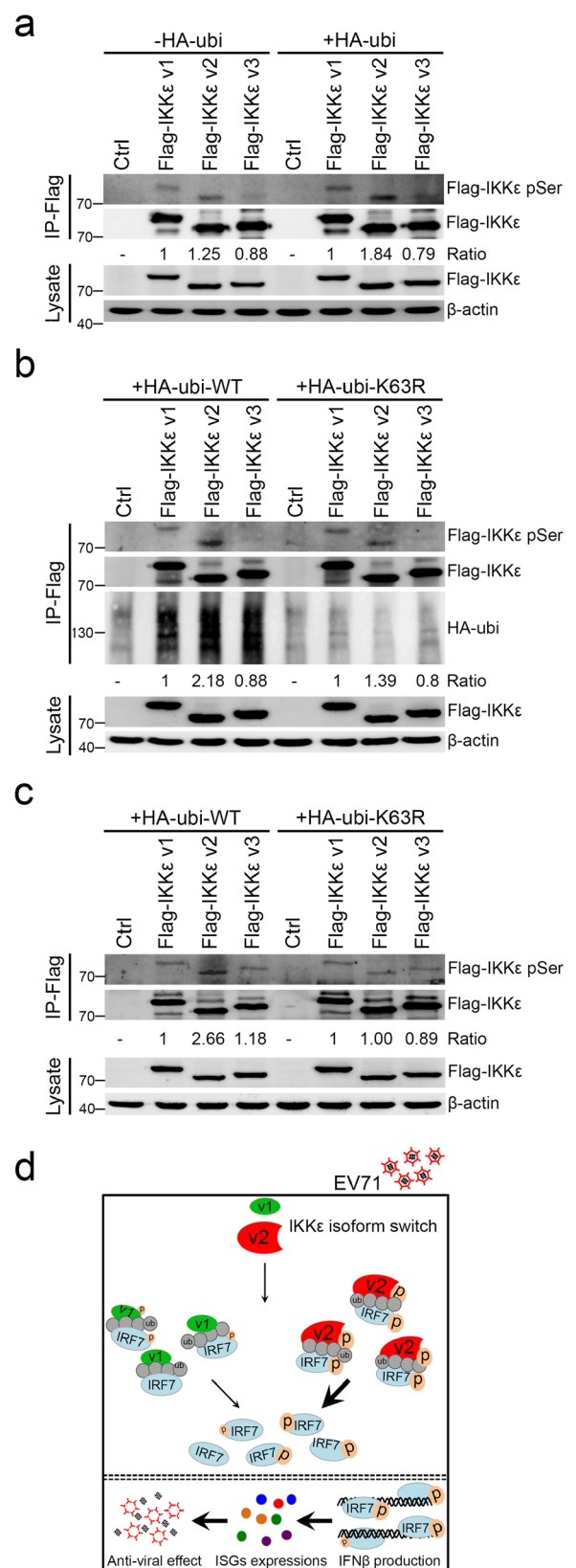

**Fig. 5 IKKε v2 processes high phosphorylation status through K63-linked ubiquitination. a** IKKε v2 presents higher phosphorylation status. HEK293 cells were transfected with each Flag-IKKε isoform accompanied with or without HA-ubi. Immunoprecipitation was performed with anti-Flag beads and the resulting products were detected by anti-phospho-Serine (pSer) and anti-Flag antibodies. **b**, **c** K63R ubiquitin markedly reduces IKKε v2 phosphorylation. HEK293 cells were transfected with each Flag-IKKε isoform accompanied with wild-type or K63R HA-ubi. Immunoprecipitation was performed with anti-Flag beads and the resulting products were further assayed by immunoblot with anti-phospho-Serine, anti-Flag, and anti-HA antibodies (**b**) and in vitro kinase assay (**c**). β-actin was served as an internal control. **d** Molecular mechanism of IKKε isoform switching in EV71 infection.

were reduced by EV71-induced miRNA-146a and resulted in suppression of IFNβ production in EV71-infected cells[3]. miR-526a targeting cylindromatosis (CYLD) was demonstrated to stimulate phosphorylations of IRF3, IκB, and IKKε, and down-regulation of miR-526a in EV71 infection impaired IFN-I production[70]. miR-548 known to regulate the host antiviral responses by directly targeting IFN-λ1 expression was shown to be suppressed upon EV71 infection[71]. miR-302 cluster suppressed EV71-induced innate immune response via direct targeting of karyopherin α2 (KPNA2)[72]. More recently, Duan et al. showed the level of miR-628-5p was increased after EV71 infection. miR-628-5p suppressed TRAF3, which mediates IRF3 and NF-κB activation, to further affect IFNβ production during EV71 infection[73]. Taken together, IFN induction is regulated in a virus- and cell type-specific manner and IFNβ expression in EV71 infection is complicated and tightly regulated.

The host IFN pathway response to EV71 has been reported in which IRF3 expression is significant reduced while IRF7 is unchanged in EV71-infected RD cells[74]. Furthermore, IRF3-CL, an isoform of IRF3, is ubiquitously expressed in all cell lines and acts as a negative regulator of IRF3 via dimerization with IRF3 in the presence of IKKε overexpression[75]. It is possible that IRF3-CL plays an inhibitory role in ectopically IKKε isoform-expressing RD cells upon EV71 infection, however, we did not measure IRF3-CL expression in our study. Hence, the limited effects of IKKε v1 on IRF3/7 signaling, IFN production and antiviral activity observed in our study may be attributed to the virus and cell type-specific IFN regulation, reduced expression of IRF3 and inhibitory effect on IRF3 by IRF3-CL in EV71-infected RD cells ectopically expressing IKKε v1. Moreover, the absent effect of IKKε v1 on immune and antiviral activities strengthens our finding that IKKε v2 presents higher activities in IRF7 interaction and activation compared with IKKε v1 and IKKε v2 possessing higher kinase activity promotes IFN and ISG production to establish host antiviral responses by interacting with, phosphorylating and activating IRF7.

Recently, TRIM6, an E3-ubiquitin ligase, was reported to synthesize K48-linked polyubiquitin chains that could interact with IKKε and lead to IKKε activation. Furthermore, it has also been shown that TRIM6 and TRIM6 synthesized K48-linked polyubiquitin chains could more efficiently interact with the kinase domain of IKKε mutants lacking the C-terminal coiled-coil domain[48]. Taken together, these findings suggest that IKKε v2 lacking a portion of the C-terminal coiled-coil domain might result in a conformation that is highly conducive to K63-linked ubiquitination, in tandem, facilitating its autoactivation. It has been proposed that the isoform switching is one of the regulatory mechanisms to fine-tune the functions of IKKε[76]. Koop's study found two artificial IKKε variants lacking exon 20-22 and exon 21–22 exhibited inhibitory effect on IRF3 signaling. The

protease (2A^pro) targets MAVS and cleaves MDA5 which is responsible for IRF3 activation to inhibit IFN production[68,69]. It is worth noting that these different regulations are demonstrated in different cell lines under different experimental conditions. On the other hand, cellular miRNAs also play roles in EV71-induced innate immune response. IRAK1 and TRAF6 proteins

discrepancy between Koop's study and our findings might be partly resulted from different IKKε constructs and different assay conditions, i.e., different cells used and EV71 challenge or not.

RIG-1/MDA5, the upstream activator of IKKε, was reported to be cleaved by EV71-encoded 2A$^{pro}$ and 3C$^{pro}$ and led to inhibition of the IFNα/β response[65,68]. However, upregulation of RIG-I ubiquitination promoted the expression of IFNβ and ISGs[77]. Another study indicated that ARRDC4 promoted K63 polyubiquitination of MDA5, consequently activating the innate immune response in EV71 infection[78]. In the present study, we found that IKKε v2 showed higher activity in the presence of K63-linked ubiquitination, and IKKε v2 promoted IRF7 activation under enhanced ubiquitination in EV71 infection. Whether RIG-1/MDA5 is involved in the regulation of IKKε isoform switching is remaining for further investigation.

Alteration of host RNA splicing is a common feature in virus infections, and the most frequently alternative splicing is exon skipping[79]. One mechanism of the alternative splicing under virus infections is directly caused by a viral manipulation of splicing machinery. 3D polymerase (3D$^{pol}$) of EV71 entered the nucleus and targeted the central pre-mRNA processing factor 8 (Prp8) to block pre-mRNA splicing and mRNA synthesis[80]. The other mechanism is related to virus-responsive regulation on splicing factors, which regulates the splicing of key players of the antiviral innate immunity. RIG-1 and stimulator of interferon genes (STING) splice variants were upregulated upon viral infection and strongly inhibited RIG-1 and STING signaling pathways, respectively[81,82]. In this study, we found that IKKε isoform switching took place in EV71, CVB3, and HSV-1 infections, and such a regulated isoform switching resulted in the controlled production of IFNβ. Whether splicing machinery is altered and IFN is involved in the regulation of IKKε isoform switching upon viral infection remain to be elucidated.

We previously reported that miR-146a is involved in blocking of type I IFN production in EV71-infected subjects[3]. In addition to miRNAs, we found that IKKε, another regulatory molecule in IFN production, exhibited isoform switching in EV71-infected cells, and these results suggested that IKKε isoform switching might play a regulatory role in the establishment of host antiviral immunity. Herein, we demonstrated that IKKε v2 presents higher activities in IRF7 interaction and activation than the other two IKKε isoforms. IKKε v2 possessing higher kinase activity promoted IFN and ISG production to establish host antiviral responses by interacting with, phosphorylating and activating IRF7. However, in nature, although IKKε isoform switching occurs in EV71 infection, IKKε v2 cannot ultimately overtake IKKε v1. Moreover, the expression level of IKKε was diminished in EV71 infection. These limitations might partially explain why EV71-infected cells could not establish a sufficient immune response against virus infection. In summary, we characterized IKKε v2 functions and provide evidence that host cells could regulate immune responses against virus infection by governing gene isoform switching. These findings might highlight a strategy for the development of antiviral therapy by manipulating gene isoform switching.

## Methods

**Cell cultures and virus infection**. Human rhabdomyosarcoma cells (RD), human embryonic kidney 293 cells (HEK293), and human cervix carcinoma cells (HeLa) were cultured in MEM and DMEM medium, respectively, supplied with 1 mM L-glutamate and 10% fetal bovine serum. RD cells were used in propagation and plaque titration of EV71. The virus infection was performed in serum-free condition. Aliquots of viral stocks were stored at −80 °C. The RD and HeLa cell lines were purchased from Bioresource Collection and Research Center (BCRC, Taiwan) and HEK293 cell line was purchased from American Type Culture Collection (ATCC, USA). All cell lines were mycoplasma negative.

**siRNA transfection**. RD cells were transiently transfected with siRNA1-IKKε or siRNA2-IKKε (s18537, s18538, ThermoFisher) at a final concentration of 50 nM using RNAiMAX (Invitrogen). Further treatments or assays were generally performed 48 h after siRNA transfection.

**Click-iT AHA assay**. RD cells were cultured in glutamine, methionine, and cystine-free medium supplemented with 50 μM Click-iT AHA (L-Azidohomoalanine) (Invitrogen). After incubation, AHA was taken up by cells and loaded onto methionine tRNAs. During translation, AHA is incorporated into newly synthesized proteins. A biotin-based tag is then added by click chemistry according to manufacturer's instructions, and the newly synthesized protein was precipitated using streptavidin dynabeads (Invitrogen) followed by western blotting analysis with antibody against IKKε (1:1000, Cell Signaling).

**Plaque assay**. EV71 plaque assays were carried out in triplicate in 6-well plates. RD cells were infected with 100 μl/per well of diluted viral stocks. After 1 h absorption, the cell monolayer was washed with phosphate-buffered saline (PBS) and incubated for 3 days in 0.3% agarose medium overlay. Cells were fixed with formaldehyde and stained with crystal violet. Plaques were counted.

**Digital PCR**. Total RNAs derived from mock- or EV71-infected RD cells were first converted to complementary DNAs (cDNAs). 500 ng cDNAs were directly analyzed by Digital PCR. Digital PCR was performed by QX200 droplet digital PCR system (Bio-Rad). Briefly, Taq polymerase PCR reaction mixtures were assembled with TaqMan probe, master mix, and cDNA samples. DG8 cartridges were loaded with 20 μl PCR reaction mixtures and 70 μl of droplet generation oil for each sample. The cartridges were placed into a droplet generator for emulsification and the emulsified samples were then transferred onto 96-well droplet PCR plate for 40 cycles PCR reaction. After PCR, the PCR plates were loaded into a droplet reader, which sequentially read droplets from each well of the plate. Analysis of digital PCR data was performed using the RED mode of the QX200 analysis software (version 1.2.10.0, Bio-Rad).

**Promoter assay**. HEK293 or RD cells were cotransfected with 4XPRDIII/I, IRF7 Luc, IRF3 Luc, ISG56, ISG20, MxA or OAS1 promoter reporter plasmid and Renilla luciferase plasmid in the presence of each IKKε isoform plasmid. Cells were harvested at indicated time points after transfection or h.p.i. and applied to Dual-Glo luciferase assay (Promega) according to the manufacturer's instructions. The luminescent signals were measured by Victor3 multilabel counter (PerkinElmer). The activity of Renilla luciferase was used as an internal control to normalize transfection efficiency, and the fold induction was calculated as the ratio of samples transfected with each IKKε isoform plasmid versus samples transfected with vector control.

**Plasmid constructions**. The coding DNA sequence (CDS) of IKKε v1 was amplified and served as a template to perform deletion PCR to generate IKKε v2 and IKKε v3. IKKε and IRF7 CDS fragments were constructed into pCMV2-tag (Stratagene) or pcNDA3.1-V5 (Thermo Fisher Scientific) mammalian expression vector. IRF3-Luc, IRF7-Luc, HA-ubiquitin-WT, and HA-ubiquitin-K63 mutant were gifted from Dr. Helene Minyi Liu (National Taiwan University, Taiwan). PRDIII/I fragment obtained from IFNβ promoter region was constructed into pGL4.17 reporter vector (Promega) with 4-time repeats. The newly generated plasmids were deposited at Addgene with ID79438.

**Individual quantitative real-time PCR**. Quantifications of IFNβ, ISG20, ISG56, OAS1, MxA, and TBP were performed by SYBR Green-based quantitative real-time PCR and TBP was served as internal control. The primer sequences were listed in Supplementary Table 2.

**Immunoprecipitation and immunoblot analysis**. For interaction detection, HEK293 cells were harvested in IP buffer (150 mM NaCl, 20 mM Tris, pH7.5, 0.1% Triton X-100, protease inhibitor cocktail, 10 mM N-ethylmaleimide, 5 mM NaF and 2 mM Na$_3$VO$_4$) and applied to immunoprecipitation assay. For detection of phosphorylated IKKε and IRF7, HEK293 cells were extracted in strict IP buffer (400 mM NaCl, 20 mM Tris, pH7.5, 0.2% Triton X-100, 0.1% SDS, protease inhibitor cocktail, 10 mM N-ethylmaleimide, 5 mM NaF and 2 mM Na$_3$VO$_4$). After extraction, the Immunoprecipitations were performed with anti-Flag M2 affinity beads (Sigma) at 4 °C for 2 h. For immunoblotting, proteins concentrations were measured by BCA protein assays. Proteins were then resolved by 8% sodium dodecyl sulfate polyacrylamide gel electrophoresis, transferred onto PVDF membranes, blocked with 5% skimmed milk in Tris-buffered saline (TBS) (20 mM Tris-HCl (pH 7.5), 150 mM NaCl, and 0.1% Tween-20) and reacted with primary antibodies for β-actin (1:5000; Sigma), Flag (1:2000; Biolegend), phosphor-Ser (1:500, Cell Signaling), IRF7 (1:500; Biolegend), phospho-IRF7(Ser471/472) (1:1000; Cell Signaling), HA (1:1000, Bethyl), H3 (1:3000; Cell Signaling), V5 (1:5000, Thermo Fisher Scientific), α-tubulin (1:2000, Biolegend) and ubiquitin (1:1000, Sigma). Immunoblotted membranes were reacted with secondary antibodies, HRP-conjugated anti-rabbit or anti-mouse IgG antibodies (1:5000, Santa

Cruz). The signals were developed by an enhanced chemiluminescence reagent (Millipore). β-actin served as an internal control. H3 and α-tubulin were used as nuclear and cytosolic markers, respectively.

**Biochemical fractionation**. Briefly, HeLa cells were harvested and resuspended in buffer A (10 mM Tris-HCl, pH 7.9, 1.5 mM $MgCl_2$, 10 mM KCl, 0.5 mM DTT, and protease inhibitor cocktail). The cells were osmotically swollen by incubation for 15 min on ice and then homogenized with ten strokes using a dounce homogenizer. The homogenized suspension was centrifuged at 8000 r.p.m. at 4 °C for 5 min. The supernatant was the cytoplasm fraction and the pellet contained the nuclei. The pellets were washed with 0.1% NP-40/PBS, and the nuclei were resuspended in buffer C (20 mM Tris-HCl, pH7.9, 0.4 M NaCl, 1.5 mM $MgCl_2$, 0.2 mM EDTA, 0.5 mM DTT, 25% glycerol and protease inhibitor cocktail) followed by ten strokes using a dounce homogenizer. The homogenized nuclei were centrifuged and the supernatant contained the nuclear extract fraction.

**Immunofluorescence staining**. RD cells cotransfected with V5-tagged IRF7 and each Flag-tagged IKKε isoform were fixed with 3.7% paraformaldehyde and permeabilized with 0.5% Triton-X100/PBS. After incubation in blocking buffer (1% BSA/PBS), slides were incubated sequentially with primary antibody (anti-Flag, Proteintech 20543-1-AP, and anti-V5, Invitrogen #R961-25) and secondary antibody (Goat anti-Rabbit Alexa 594 and Goat anti-Mouse Alexa 488). Nuclei were counterstained with Hoechst33342 (Sigma 14533).

**Image acquisition and analysis**. Fluorescence images were captured on a spinning disk confocal (Zeiss). Fluorescence signals were captured using sequential acquisition to give separate image files for each slide. Image analysis was performed using MetaMorph (Molecular Devices). More than five fields were selected for analysis of each stain. The ratio of granules in nucleus/granules in cytoplasm was measured from at least fifty cells.

**In vitro kinase assay**. The Flag-IKKε isoforms were transfected into HEK293 cells in the presence or absence of HA-ubiquitin and pulled down by immunoprecipitation using anti-Flag M2 affinity gel (Sigma) with lysis buffer (0.4 M NaCl, 20 mM Tris-HCl pH 7.5, 0.2% Triton X-100, protease inhibitor cocktail, 10 mM N-ethylmaleimide, 2 mM $Na_3VO_4$ and 5 mM NaF). The beads were washed three times in ice-cold washing buffer (0.4 M NaCl, 20 mM Tris-HCl pH 7.5, 1% Triton X-100) and twice with cold kinase assay buffer containing 20 mM HEPES pH7.5, 10 mM $MgCl_2$, 25 mM NaCl, 10 mM $Na_3VO_4$, 10 mM NaF, and 2 mM DTT. The kinase reaction was performed in the presence of 50 μM ATP and a His-tagged truncated form of human IRF7 (amino acids 428-517), in a total volume of 30 μl of kinase assay buffer at 37 °C for 45 min with gentle agitation. The kinase reaction was terminated by the addition of 10 μl 4× SDS sample buffer followed by boiling for 10 min. The samples were resolved by SDS-PAGE, transferred to PVDF membrane and probed with phospho-Ser antibody (Millipore). The blot was stripped and reprobed sequentially for IKKε isoforms with anti-Flag antibody (Biolegend).

**Transcriptome analysis**. Illumina pair-end reads of each time point (mock infection, 4, and 8 h.p.i.) were separately aligned to the human genome (hg19) using Tophat[83]. The isoforms of each gene were assembled by Cufflinks and the expression abundance both at the gene and isoform levels were estimated[17,24]. Genes and isoforms with 2-fold differential expression between any indicated h.p.i. against mock infection were identified by Cuffdiff. The RNA-Seq (mock 0, 4, 8) of this study was deposited at NCBI BioProject with PRJNA717395.

**Functional ontology enrichment and signaling pathways analysis**. A total of 1035 genes were differentially expressed in gene level with least 2-fold change (Supplementary data 1). The genes were further analyzed for signaling pathways and GO process networks by using MetaCore Analytical Suite (GeneGo). The top 50 GeneGo Maps folders altered in EV71 infection were shown in Supplementary data 2.

**Identification of isoform switching genes**. The genes with two or more splicing isoforms were selected for subsequent analysis, as genes might show isoform switching between any indicated h.p.i. against mock infection. In this study, we focused on investigating the isoforms of skipping exons. The expression levels (Fragments Per Kilobase of transcript per Million reads; FPKM) of each isoform at two time points were compared and the potentially isoform switching genes were identified if the proportion of one isoform was upregulated while that of the other was downregulated in EV71 infection. We further required that the fold change of expression levels of one isoform between two assayed time points have to be at least 1.5 fold, which leaves only 242 genes sufficient for subsequent enrichment analysis (Supplementary data 3). Note that we remove candidate genes with exon-skipping isoforms occurred at the UTRs, as the transcript assembly is less reliable at ends of each transcript. In order to reduce the analysis complexity, we apply the analysis strategy of isoform switching between mock infection and 8 h.p.i.. The KEGG pathway annotation and enrichment analysis of isoform switching genes are carried out by DAVID Bioinformatics Resources (https://david.ncifcrf.gov/) (Supplementary Table 1).

**Statistics and reproducibility**. All data in bar plots are presented as mean ± SD. Student's *t* test was used to compare the significance in promoter assays and gene expression experiments. The *p* value <0.05 for significance and two-tailed tests were used in this study. In all the figures, n refers to the number of biological replicates and is indicated in every figure legend. All statistical analyses and data wrangling were performed using Graphpad Prism software v9.0.2.

**Reporting summary**. Further information on research design is available in the Nature Research Reporting Summary linked to this article.

## Data availability

All data supporting the findings of this study are available within the paper and its Supplementary information files. Source data for the figures can be found in Supplementary Data, and full western blot images are included in Supplementary Figs. 7–16. The RNA-seq dataset has been deposited to NCBI with accession number PRJNA717395.

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

## Acknowledgements

We thank the Pharmacogenomics Laboratory of the National Research Program for Biopharmaceuticals and the NGS and Microarray Core Facility of NTU Centers of Genomic and Precision Medicine as well as the staff of the imaging core at the First Core Labs, National Taiwan University College of Medicine for technical assistance. This work was supported by MOST (MOST105-2320-B-002-066), NRPB (MOST103-2325-B-002-035, MOST104-2325-B-002-008, MOST105-2325-B-002-002), and EID (MOST104-2321-B-002-047, MOST105-2321-B-002-012, MOST106-2321-B-002-004) as well as the "Center of Precision Medicine" from The Featured Areas Research Center Program within the framework of the Higher Education Sprout Project by the Ministry of Education (MOE) in Taiwan.

## Author contributions

This study was conceptualized by Y.L.C. and S.L.Y., investigated by Y.L.C., Y.W.L., M.H.C., S.Y.C., B.C.H., and Y.T.H.. Resources were provided by S.L.Y. The manuscript was written by Y.L.C. and M.H.C. and reviewed by B.C.H. and Y.T.H.

## Competing interests

The authors declare no competing interests.
