## [Peer Review File · Communications Biology]

Reviewers' Comments:

Reviewer #1:

Remarks to the Author:

In all, this is a very interesting paper which suggests an IKKi isoform (V2) is more strongly activate the RNA virus dependent innate immune signaling. However, the following points are not very convincing to me 1) the isoform switch really happens in the virus infection model (e.g. or just the IKK-v2 isoform become dominant under any condition or in any cell types, which could suggest a physiological relevant role); 2) the mechanism of why IKKi-v2 that lacks the coil coiled domain promote such a robust activation (e.g. whether the coil coiled domain of IKKi inhibit IKKi kinase activity, or it inhibits IKKi binding to IRF7 in the presence of virus). So I suggest a revision that focuses on strengthening these two points. Please refer to the following detailed comments of the paper.

1. For Figure 1, the isoform switching is not very obvious and convincing. The changes are very small upon infection. Therefore, the dd PCR validation is very important to confirm that there is an isoform switching. It would be helpful to show the raw data/detailed analysis of the ddPCR so that the readers could also understand where the percentage of different IKKi isoforms come from. Also, if there indeed is an isoform switch, is this specific to EV71 infection? Different viruses such as Senda virus, VSV and different cell lines should be tested (e.g. fibroblasts such as MEF cells vs immune cells such as macrophages.)

2. For Figure2, Is IRF7 mRNA and protein induced by IKKi-V2? Western blot and qPCR of endogenous IRF7 expression after IKKi-V2 over expression and viral infection would be helpful to determine if IRF7 upregulation is an important downstream effect of IKKi-V2 expression. IKKi-V2 expression may cause higher basal IRF7 level which could help elicit a more rapid interferon response upon viral infection. Do you see a strong interferon-alpha production? This is actually very interesting since in pDC IRF7 is expressed at high level without infection. Maybe the authors should check if pDC alternatively expresses IKKiV2 more obviously, which would serve as another physiological relevant examples (The pDC experiment is a curious inquiry, not required for the paper if you could show convincingly in figure 1 that there is an isoform switching, but if not, this may be another physiological relevant condition you could take a look at.)

3. For figure 2 again, in order to test the function of IKKi isoforms, it would be the best to test these isoform constructs in a IKKi KO cell line, otherwise the effect of individual isoforms may be mediated through the wildtype protein (v1).

4. For figure 3, the experiments don't support the claim. Increasing the overall mono-ubiquitin level alone will not mimic viral infection since these monoubiquitin needs to be assembled into specific ub-chains and be targeted to specific proteins upon E3 ligase activation (e.g. TRIM25 for RIG-I activation and TRAF proteins downstream of MAVS). But it certainly may sensitize the signaling as it increase then availability of ubiquitin. Therefore, the experiment at least should be done with viral infection in IKKe knockout and Isoform-reconstituted cells (maybe under this condition you don't have to overexpress ubiquitin to see an effect). However, if you really want to claim IKKi V2 phosphorylates and activates IRF7 in the presence of ubiquitin, an in vitro system with all the purified components would be necessary.

Reviewer #2:

Remarks to the Author:

In this paper, author found that IKKe isoforms mRNA expression pattern was affected by EV70 infection then they found IKKeV2 mRNA is up-regulated but IKKeV3 is down-regulated in RD cell line. IKKeV2 showed pre-dominantly interact with IRF7 and phosphorylate IRF7 for induction of ISGs compared to other IKKe isoforms and that was enhanced by overexpressed ubuquitine-mediated K63 poly-ubiquitination. The manuscript is well written and figures are presented nicely. RNA-Seq data set would be useful for research in this field. RNA-Seq based finding of IKKe isoforms and its switching idea is interesting but there are some issues that require authors attention.

Major points:

- 1) In Figure 1, further RNA-Seq analysis is needed for providing the information in manuscript. Author should show the heat map of ISGs and inflammatory cytokines expression including IFN β , OAS1, ISG56 between mock, 3hpi, 8hpi to show the immune response against EV70 that authors used down-stream target genes of IKK ϵ isoforms and analyzed promoter activity in Figure 2. Author also should make the graph (-log P-value of each GO clusters) of GO database clusters analysis to clearly show their listed in Table 2. Because current Table 2 is just list of statistically different GO name (Table 2), it is not convenient and lack the kindness for paper reading. Author also should combine the analysis of viral derived RNAs in RNA-Seq.
- 2) In Figure 1, It is needed to show the protein expression of isoforms by using Western blot or Mass Spectrometry to consider the following figures such as analyzing down-stream signaling of IKK ϵ isoforms. If it is difficult to show the protein expression, at least author should address endogenous IKK ϵ isoforms by using their siRNA which used in Figure 2f to show the accuracy of their detection quality by using ddPCR.
- 3) In Figure 2e, authors analyzed the effect of IKK ϵ isoforms over-expression on EV70 infection. To show the importance of endogenous produced IKK ϵ V2 on ISGs induction and host defense against EV70 infection, authors should measure the EV70 viral titer and ISGs mRNA expression by using s IKK ϵ V2 iRNA, which they used in Figure 2f, without any overexpression plasmid. This is important issue to consider the function of IKK ϵ isoform switching in physiological condition.
- 4) In Figure 2f, author did not use siRNA for IKK ϵ V3 which is not involved in switching IKK ϵ isoforms story in this paper. It was also not affected by EV70 infection in Figure 1b and might be difficult to design the siRNA. Author should comment the reason why they skip IKK ϵ V3 knockdown in Figure 2f.
- 5) In Figure 3, Author should show the IRF7 phosphorylation in IKK ϵ isoforms over-expressed RD cell infected with EV70 which they could see the effect of IKK ϵ V2 in Figure 2f.
- 6) It is difficult to see the difference of nuclear IRF7 band intensity in Western blot data. Author should calculate the intensity and label the value or make a graph under each panel (Figure 3c and 3d). And to check the exact nuclear translocation of IRF7, author should show nuclear IRF7 by using confocal microscopy by staining IRF7 and IKK ϵ in the condition of Figure 3d.
- 7) About Figure 4 and 5, author should calculate the band intensity of V5-IRF7 in IP-Flag and normalized by V5 in lysate and indicate the value or graph for Figure 4a. The same manner calculation is needed for Figure 4b, 4c, 5a, 5b, 5c.

Minor points:

- 1) There is no information about Taqman probe and primer sequence, siRNA sequence, p-IRF7 antibody.
- 2) Author analyzed several ISGs mRNA expression using same samples (Figure 2b, 2d, S2a, S2b). To easier understanding of overall ISGs production, authors could make heat map in one panel to show the ISGs expression profile affected by IKK ϵ V2 over-expression.
- 3) Author should mention innate sensors such as RIG-I and MDA5 which recognize EV70 infection in introduction part.
- 4) What is potential mechanism of IKK ϵ isoform switching? Is there some transcription factor needed? Author can mention this point in discussion part.
- 5) Is it occurred in other viral infection and PRR ligand treatment? Author can mention this point in discussion part also.

Reviewer #3:

Remarks to the Author:

The manuscript compares the ability of three different IKK ϵ isoforms to induce IRF7 activation, IFN induction and limit EV71 replication. It is an interesting observation that EV71 infection induces isoform switching, and the manuscript also provides us with a better understanding of IKK ϵ domain function.

The data overall looks convincing and experiments well controlled.

I have some queries about the data as listed below:

1. Overall I find it surprising that IKK ϵ -v1 doesn't induce much of a response in most read-out systems, given that this is the full-length version that most researchers would have used in the

past to study IKKe. I would have expected overexpression of IKKe-v1 to upregulate read-outs in Figure 2 more strongly, and to also limit viral replication upon overexpression. Did the authors check expression levels of the different versions in their reporter assays, qPCR and viral replication assays (apart from 2f). I am equally surprised that IRF3 is not activated at all with overexpression of any IKKe variant. A positive control (TBK1?) would have been useful in this assay.

2. The molecular weight of version 3 looks different in Western Blots from Figure 3 onwards compared to blots in Figure 2.

3. Figure 3c: would we not expect some effect also in the absence of ectopic Ubiquitin expression (due to ubiquitination occurring with endogenous ubiquitin)?

4. The study heavily relies on overexpression experiments, some of which are unavoidable to delineate effects of the different variants. However: is it possible to detect expression of these IKKe isoforms with commercially available IKKe antibodies? If so, can we see a change in protein expression levels following EV71 infection? Also, in Figure 2f the authors used variant-specific siRNA to knock-down expression of their ectopically-expressed IKKe variants. Should the same siRNAs not also be able to knock down the endogenous variant mRNAs and therefore affect viral replication in the absence of ectopically-expressed flag-IKKe variants?

5. Is there a qualitative or just a quantitative difference between IKKe v1 and v2? Is one more active in IFN induction and the other more active in IFN signalling? V2 seems have even stronger effect on ISG expression compared to IFN induction?

6. Finally, some language editing might be useful and an expansion of the discussion.

7. Do the authors know what induces isoform switching? Is it an IFN-induced response? Is it RIG-1/mda-5-dependent?

Reviewers' comments:

Reviewer #1 (Remarks to the Author):

In all, this is a very interesting paper which suggests an IKKi isoform (V2) is more strongly activate the RNA virus dependent innate immune signaling. However, the following points are not very convincing to me 1) the isoform switch really happens in the virus infection model (e.g. or just the IKK-v2 isoform become dominant under any condition or in any cell types, which could suggest a physiological relevant role); 2) the mechanism of why IKKi-v2 that lacks the coil coiled domain promote such a robust activation (e.g. whether the coil coiled domain of IKKi inhibit IKKi kinase activity, or it inhibits IKKi binding to IRF7 in the presence of virus). So I suggest a revision that focuses on strengthening these two points. Please refer to the following detailed comments of the paper.

1.1 For Figure 1, the isoform switching is not very obvious and convincing. The changes are very small upon infection. Therefore, the dd PCR validation is very important to confirm that there is an isoform switching. It would be helpful to show the raw data/detailed analysis of the ddPCR so that the readers could also understand where the percentage of different IKKi isoforms come from.

Author reply:

We appreciate your constructive comments.

We totally agree with your comments. The expression levels of isoforms v1 and v2 can't be easily interpreted from Figure 1a, as it requires assembly of short reads into isoforms prior to quantification (i.e., assign each read to its originated form). The detailed analysis was mentioned in Transcriptome analysis of Methods section. After isoform assembly and quantification of the expression levels, it indicated IKK ϵ v1 in mock infection compared with EV71 infection 8 h.p.i was significantly down-regulated from 71% to 57% ($p=0.003$) while v2 was significantly up-regulated from 11% to 26% ($p=0.000001$) (revised Figure 1b IKK ϵ NGS/RD ratio). The isoform switching seems oblivious.

Droplet digital PCR (ddPCR) is a powerful technology with high sensitivity to be used for absolute quantification of targets. In the ddPCR system, the number of fluorescence-positive signals indicated how many target molecules were detected in the sample. In this study it is impossible to design the specific probe for each IKK ϵ isoform because there are no isoform-unique sequences

available for probe design. We cannot directly detect individual isoform, so we designed a strategy to calculate the copies of each IKK ϵ isoform in different assay conditions as follows:

1. Three commercially available probes were selected to detect IKK ϵ isoforms. First one detects all the three IKK ϵ isoforms (Hs01063858-m1, Thermo Fisher), second one detects both IKK ϵ v1 and v2 (Hs01069870_m1, Thermo Fisher), and the last one detects both IKK ϵ v1 and v3 (Hs01063855_g1, Thermo Fisher). The locations of these probes were labeled on schematic diagram in Supporting Figure 1.
2. After ddPCR assays, we subtracted copies of Hs01069870_m1 (detecting v1 and v2) from Hs01063858-m1 (detecting v1, v2, and v3) to have IKK ϵ v3. Similarly, we subtracted copies of Hs01063855_g1 (detecting v1 and v3) from Hs01063858-m1 (detecting v1, v2, and v3) to have IKK ϵ v2. Finally, copies of Hs01063858-m1 minus IKK ϵ v3 and IKK ϵ v2 leaves IKK ϵ v1.
3. Ratio of individual IKK ϵ isoform was calculated by copy number of each IKK ϵ isoform divided by total copies of three isoforms.

Supporting Figure 1. Schematic diagram for the locations of IKK ϵ probes used in ddPCR assay.

We have revised the related description in revised Results section and provided the detailed analysis of the ddPCR and probe information as well as ddPCR raw data in revised supplementary methods and in revised supplementary Table 5 and supplementary Table 6 as follows.

Revised Results: (Page 7, Line 15 - Page 8, Line 3)

Three primer/probe sets were used in ddPCR assay and the expression of each IKK ϵ isoform was calculated (see the detailed ddPCR analysis in supplementary methods, Supplementary Table 5). The results, consistent with the RNA transcriptomic findings (Fig. 1a, b), indicated that IKK ϵ v2 is upregulated while IKK ϵ v1 is downregulated in pace with EV71 infection (Fig. 1b and Supplementary Table 6).

Supplementary Methods

Droplet digital PCR analysis

Three commercially available probes purchased from Thermo Fisher were used to detect IKK ϵ isoforms in which Hs01063858-m1 detects all the three IKK ϵ isoforms, Hs01069870_m1 detects both IKK ϵ v1 and v2 and Hs01063855_g1 detects both IKK ϵ v1 and v3. After ddPCR assays, we subtracted copies of Hs01069870_m1 (detecting v1 and v2) from Hs01063858-m1 (detecting v1, v2, and v3) to have IKK ϵ v3. Similarly, we subtracted copies of Hs01063855_g1 (detecting v1 and v3) from Hs01063858-m1 (detecting v1, v2, and v3) to have IKK ϵ v2. Finally, copies of Hs01063858-m1 minus IKK ϵ v3 and IKK ϵ v2 leaves IKK ϵ v1. Ratio of individual IKK ϵ isoform was calculated by copy number of each IKK ϵ isoform divided by total copies of three isoforms.

Revised Supplementary Table 5

Supplementary Table 5. List of primers and probes

Gene name	Sequence (5'-3')
IFNb-SYBR-F	ATTGCCTCAAGGACAGGATG
IFNb-SYBR-R	GGCCTTCAGGTAATGCAGAA
OAS1-SYBR-F	CCCCATTATTGAAAAGTACCTGAGA
OAS1-SYBR-R	GCCGGGTCCAGGATCAC
ISG56-SYBR-F	CAGAACGGCTGCCTAATTTACA
ISG56-SYBR-R	GTGGGTCTGCTTTTTTCTCTGT
ISG20-SYBR-F	CCCTGCGGGTGCTGAGT
ISG20-SYBR-R	TGTCCAAGCAGGCTGTTCTG
Mxa-SYBR-F	GCT ACTGTGGCCCAGAAAAATC
Mxa-SYBR-R	TCATACTGGCTGCACAGGTTGT
Promoter Luc reporter	Sequence (5'-3')
OAS1-F	CCCGGTACCCTTAACAAAAAGAAAAGAGAC
OAS1-R	TTTAAGCTTTTTACCACCTTGGACACACA

ISG56-F	TAAGGTACCGCACCCAGCCAAGAATCATT
ISG56-R	CGCAAGCTTAGATCTGGCTATTCTGTCTT
ISG20-F	AAAGGTACCCCAAATCCCACTTGGTGAAA
ISG20-R	AAAAAGCTTCTCTCACCTGCCTGCCTCTG
Mxa-F	ACCGGTACCCCAAAGCTCACCAGTATCAA
Mxa-R	ATAAAGCTTCTCTGCTACCAGGCTGAGGA
Expression vector	Sequence (5'-3')
IRF7-HindIII-F	AAAAAGCTTATGGCCTTGGCTCCTGAGAGGGCAG
IRF7-BamHI-R	AAAGGATCCGGCGGGCTGCTCCAGCTCCATAAGG
Digital Probe	Target exon boundary
Hs01063858_m1	NM_014002.3 Exon 21-22
Hs01069870_m1	NM_014002.3 Exon 2-3
Hs01063855_g1	NM_014002.3 Exon 19-20

Revised Supplementary Table 6

Supplementary Table 6. The raw data of ddPCR

Sample	Probes		
	Hs01063858	Hs01069870	Hs01063855
RD_Mock infection	3330	2960	2780
RD_4 h.p.i.	5140	4450	3820
RD_8 h.p.i.	1750	1436	1026
SH-SY5Y_Mock infection	2620	2120	2440
SH-SY5Y_12 h.p.i.	3260	2120	2900
SH-SY5Y_24 h.p.i.	2440	1656	2140
	copies of each IKKε		
	IKKε v1	IKKε v2	IKKε v3
RD_Mock infection	2410	550	370
RD_4 h.p.i.	3130	1320	690
RD_8 h.p.i.	712	724	314
SH-SY5Y_Mock infection	1940	180	500
SH-SY5Y_12 h.p.i.	1760	360	1140
SH-SY5Y_24 h.p.i.	1356	300	784
	Ratio of each IKKε		
	IKKε v1	IKKε v2	IKKε v3
RD_Mock infection	72%	17%	11%
RD_4 h.p.i.	61%	26%	13%
RD_8 h.p.i.	41%	41%	18%
SH-SY5Y_Mock infection	74%	7%	19%

SH-SY5Y_12 h.p.i.	54%	11%	35%
SH-SY5Y_24 h.p.i.	56%	12%	32%

1.2 Also, if there indeed is an isoform switch, is this specific to EV71 infection? Different viruses such as Senda virus, VSV and different cell lines should be tested (e.g. fibroblasts such as MEF cells vs immune cells such as macrophages.)

Authors reply:

We appreciate your interesting comments.

The replication cycle of EV71 was approximately 8 hr at a multiplicity of infection (moi) of 10 based on our previous study (Ho et al., 2011). That is why we performed RNA Seq at 4 and 8 h.p.i in this study. To understand whether IKK ϵ isoform switching is happened in different EV71-infected cell lines, first SH-SY5Y cells, a neuroblastoma cell line, were infected with EV71 at 5 m.o.i. for single virus infection cycle and RNAs were extracted at 12 h.p.i. and 24 h.p.i. (Xu et al., 2013). The IKK ϵ isoforms were measured by digital PCR. The results indicated that the percentages of IKK ϵ v1, v2, and v3 in mock infection were 74%, 7%, and 19%, respectively. At 12 and 24 h.p.i., the percentages of IKK ϵ v1 reduced to 54% and 56% while IKK ϵ v2 increased to 11% and 12% and IKK ϵ v3 increased to 35% and 32%. These results indicated that EV71 infection indeed induces IKK ϵ isoform switching in SH-SY5Y cells, and the increasing rates of IKK ϵ v2 and IKK ϵ v3 are roughly equal. (Revised Fig. 1b).

It is very interesting whether IKK ϵ isoform switching is specific to EV71 infection or is a common phenomenon happened in different virus infections. To comprehensively address this issue we analyzed IKK ϵ isoform switching in HeLa cells infected with coxsackievirus B3 (CVB3) and Herpes simplex virus-1 (HSV-1) by RNA-Seq, however, we only focus IKK ϵ isoform switching among different virus infections in this manuscript. We will prepare another manuscript to investigate genome-wide isoform switching which are universal or virus-specific and whether these isoform-switched genes are enriched in certain categories/pathways.

To meet the single virus infection cycle, HeLa cells were infected with CVB3 at 5 m.o.i. for 4 h.p.i. and 6 h.p.i. (van Kuppeveld et al., 1997) or with HSV-1 at 1 m.o.i. for 8 h.p.i. and 24 h.p.i. (Cohen et al., 2020). The alteration of IKK ϵ isoforms was calculated by NGS at indicated time points after CVB3 and HSV-1 infection. The relative expression of IKK ϵ v2 is increased while IKK ϵ v1 is decreased both in CVB3 and HSV-1 infection (Revised Supplementary Fig.

1c). Taken together, IKK ϵ isoform switching is not only found in EV71-infected RD cells but also found in EV71-infected SH-SY5Y cells, CVB3-infected HeLa cells and HSV-1-infected HeLa cells. These data strongly indicate that isoform switching is a common feature during virus infection at least in the cases of EV71, CVB3 and HSV-1.

We have added these results in the Results section of the revised manuscript, revised Figure 1b and added new Supplementary Figure 1c as follows.

Revised Results: (Page 8, Line 4 - Page 9, Line 2)

To understand whether IKKi isoform switching is happened in different EV71-infected cell lines, first SH-SY5Y cells, a neuroblastoma cell line, were infected with EV71 at 5 m.o.i. for single virus infection cycle and RNAs were extracted at 12 h.p.i. and 24 h.p.i. ²⁹. The IKK ϵ isoforms were measured by digital PCR and the results showed upregulated IKK ϵ v2 and downregulated IKK ϵ v1 expression in SH-SY5Y cells during EV71 infection (Fig. 1b). Furthermore, we investigated whether IKK ϵ isoform switching is a common characteristic in virus infections including RNA and DNA viruses. The expression of IKK ϵ isoforms was measured in HeLa cells infected with coxsackievirus B3 (CVB3) at 5 m.o.i. for 4 h.p.i. and 6 h.p.i. ³⁰, and herpes simplex virus-1 (HSV-1) at 1 m.o.i. for 8 h.p.i. and 24 h.p.i. ³¹ by RNA-Seq. The relative expression of IKK ϵ v2 is increased while IKK ϵ v1 is decreased both in CVB3 and HSV-1 infection (Supplementary Fig. 1c). CVB3 and HSV-1 infections induced IKK ϵ isoform switching in a similar pattern found in EV71 infection. These data indicated that isoform switching is a common feature during virus infection at least in the cases of EV71, CVB3 and HSV-1.

Revised Figure 1b

Fig. 1: EV71 infection triggers IKKε isoform switching.

b. IKKε isoform switching is validated by droplet digital PCR (ddPCR). IKKε v2 was up-regulated while IKKε v1 was down-regulated in response to EV71 infection determined by NGS (left panel for RD cells) and ddPCR (middle panel for RD cells and right panel for SH-SY5Y cells). The proportion of each IKKε isoform was indicated.

Revised Supplementary Figure 1c

Supplementary Fig. 1: The pathway enriched by isoform switching genes in EV71 infection.

c. IKKε isoform switching is measured by RNA-Seq in CVB3 and HSV-1 infected HeLa cells. IKKε v2 was up-regulated while IKKε v1 was down-regulated in response to CVB3 (left panel) and HSV-1 (right panel) infection determined by RNA-Seq.

Revised References: (Page 36, Line 1-12)

29 Xu, L. J. et al. Global transcriptomic analysis of human neuroblastoma cells in response to enterovirus type 71 infection. *PLoS one* 8, e65948, doi:10.1371/journal.pone.0065948 (2013).

30 van Kuppeveld, F. J. et al. Coxsackievirus protein 2B modifies endoplasmic reticulum membrane and plasma membrane permeability and facilitates virus release. *EMBO J* 16, 3519-3532, doi:10.1093/emboj/16.12.3519 (1997).

31 Cohen, E. M., Avital, N., Shamay, M. & Kobilier, O. Abortive herpes simplex virus infection of nonneuronal cells results in quiescent viral genomes that can reactivate. *Proceedings of the National Academy of Sciences of the United States of America* 117, 635-640, doi:10.1073/pnas.1910537117 (2020).

2. For Figure 2, Is IRF7 mRNA and protein induced by IKKi-V2? Western blot and qPCR of endogenous IRF7 expression after IKKi-V2 over expression and viral infection would be helpful to determine if IRF7 upregulation is an important downstream effect of IKKi-V2 expression. IKKi-V2 expression may cause higher basal IRF7 level which could help elicit a more rapid interferon response upon viral infection. Do you see a strong interferon-alpha production? This is actually very interesting since in pDC IRF7 is expressed at high level without infection. Maybe the authors should check if pDC alternatively expresses IKKiV2 more obviously, which would serve as another physiological relevant examples (The pDC experiment is a curious inquiry, not required for the paper if you could show convincingly in figure 1 that there is an isoform switching, but if not, this may be another physiological relevant condition you could take a look at.)

Authors reply:

Thanks for your constructive suggestions.

In order to address whether IRF7 upregulation is an important downstream effect of IKK ϵ v2 overexpression, we examined the endogenous IRF7 level under the ectopic expression of each IKK ϵ isoform in EV71 infection. IRF7 protein expression was measured by Western blotting, and RNA expression was measured by real-time PCR. As the results showed, compared with IKK ϵ v1, IKK ϵ v2 did not upregulate the protein and RNA expressions of IRF7 in the presence or absence of EV71 infection. Even more IRF7 RNA expression is slightly inhibited by IKK ϵ v2 compared with the vector control during EV71 infection (Revised Supplementary Fig. 4c). Taken together, IRF7 upregulation seems not be a downstream effect of IKK ϵ v2 expression.

Regarding to IFN α issue, we have determined IFN α in EV71-infected RD cells by ELISA previously and found the IFN α level was unchanged during EV71 infection (Ho et al., 2011). Please see Table S3 of this paper. In Cell Control Group of Table S3B, the IFN α level of EV71-infected RD cells (6.67 pg/ml; 8 h.p.i.) is comparable to mock-infected cells (7.58 pg/ml; 0 h.p.i.). Moreover, Mock Transfection Group of Table S3A indicated the IFN α level is quite stable in the different infection time points (29.99, 29.49, 33.63, 33.11 pg/ml at 4 h.p.i., 8 h.p.i., 12 h.p.i., 16 h.p.i., respectively).

Table S3. The specificity of antagomiR-141 and EGR1 siRNAs

A. The expression levels of interferon alpha in antagomiR-141 transfectants

	Mock Transfection		AntagomiR-NC		AntagomiR-141	
	Interferon alpha		Interferon alpha		Interferon alpha	
	multi-subtype concentration (pg/ml)	CV (%)	multi-subtype concentration (pg/ml)	CV (%)	multi-subtype concentration (pg/ml)	CV (%)
4 h.p.i.	29.99	10.15	33.84	6.09	29.26	17.69
8 h.p.i.	29.49	17.95	33.64	3.49	31.41	17.78
12 h.p.i.	33.63	17.27	32.88	15.39	29.64	18.98
16 h.p.i.	33.11	12.88	33.18	4.73	31.43	11.67

B. The expression levels of interferon alpha in siEGR1 transfectants

Sample Name	0 h		8 h	
	Interferon Alpha Multi-subtype Concentration (pg/ml)	CV (%)	Interferon Alpha Multi-subtype Concentration (pg/ml)	CV (%)
	Cell Control	7.58	7.57	6.67
Mock Transfection	2.98	8.50	6.99	5.44
siNC	5.30	3.66	4.41	4.78
siEGR1-1	3.22	1.27	7.81	2.19
siEGR1-2	7.28	3.84	7.17	2.22
siEGR1-3	3.68	2.68	8.96	6.32

Excepted from our previous study (Cell Host Microbe, 2011: 9, 58-69.)

Here, we have provided additional strong evidence to demonstrate that isoform switching is a common feature during virus infection at least in the cases of EV71, CVB3 and HSV-1. Hence, we did not explore the role of IKK ϵ v2 in pDC cells although this issue is quite interesting and worthy to further investigation.

We have added these results in the Results section of the revised manuscript and added new Supplementary Figure 4c as follows.

Revised Results: (Page 14, Line 5-10)

To characterize the role of IKK ϵ isoform switching in EV71 infection, we examined the phosphorylation and expression of IRF7 in each ectopically IKK ϵ isoform-expressing RD cells. IKK ϵ v2 strongly induced IRF7 phosphorylation (Fig. 3c), whereas the RNA and protein expression levels of IRF7 were unchanged in each IKK ϵ transfectant during EV71 infection (Supplementary Fig. 4c).

Revised Supplementary Figure 4c

Supplementary Fig. 4: IKKε v2 increases IRF7 phosphorylation and IRF7 translocation in the presence of ubiquitin.

c. IRF7 mRNA and protein were not induced by IKKε v2. RD cells were ectopically expressed Flag- IKKε followed by EV71 infection. Total lysates were loaded to perform immunoblot with indicated antibodies. β-actin served as an internal control (left panel). RNA expression of IRF7 was measured by quantitative real-time PCR and normalized with Ctrl in mock infection group (right panel).

3. For figure 2 again, in order to test the function of IKKi isoforms, it would be the best to test these isoform constructs in a IKKi KO cell line, otherwise the effect of individual isoforms may be mediated through the wildtype protein (v1).

Author reply:

Thank you for your suggestions.

Generally, the knockout of target gene is a good strategy to characterize the function of ectopically expressed isoforms avoiding the interference induced by wild-type target gene. However, there are two reasons why we hesitate to use IKKε KO cells for characterizing the function of IKKε isoforms in this study. First, IKKε v1 is still the major isoform during EV71 infection (Please see next paragraph for details). We prefer to evaluate the role of IKKε v2 in nature context. Secondly, the dimerization of IKK-related kinases is essential for biological functions. Even if IKKε v1 is knocked out, the IKKε v2 might still form a dimer with other IKK-related counterparts. Moreover, it has been reported that the IKKε-splice variant/IKKε-WT heterodimers served as a novel regulatory mechanism suitable to shift the balance between different functions of IKKε. Instead, to strengthen the functional characterization of IKKε v2, we performed additional experiments including dual tagged co-immunoprecipitation assay to measure IKKε v2/IKKε v1(wild-type)

dimerization, in vitro pull down assay to examine IKK ϵ -IRF7 direct interaction, and IP-Western blotting assay to detect IRF7 phosphorylation induced by IKK ϵ v2 in EV71 infection.

First, based on our NGS data (Fig. 1b), the percentages of IKK ϵ v1 in mock infection, in EV71 infection at 4 h.p.i. and 8 h.p.i. were up to 71%, 64%, and 57% respectively. The finding was verified by digital PCR and the results showed that the percentage of IKK ϵ v1 in mock infection was 72%, 61% and 41% in EV71 infection at 4 h.p.i. and 8 h.p.i.. Although IKK ϵ v2 is up-regulated while IKK ϵ v1 is down-regulated in pace with EV71 infection, IKK ϵ v1 is still the major form in the presence and absence of EV71 infection.

It is known that the dimerization of IKK-related kinases is necessary to exert the biological functions (Verhelst et al., 2013). Moreover, IKK ϵ splice variants formed dimers with IKK ϵ -WT and the heterodimers served as a novel regulatory mechanism suitable to shift the balance between different functions of IKK ϵ (Koop et al., 2011). In addition, the C-terminal region of IKK ϵ was reported to involve in its dimerization (Nakatsu et al., 2014). Mutant IKK ϵ with amino acid substitutions at C-terminal resulted in defect in dimer formation and concomitantly affected the ability to induce IFN β promoter activity. As our analysis of IKK ϵ isoform switching in EV71 infection, we could suspect that IKK ϵ v2 and IKK ϵ v3 might form heterodimers with IKK ϵ -WT, and the heterodimers are the major populations during virus infection.

The lack of exon 20 in IKK ϵ v2 leads to a frame shift and results in a C-terminus truncated protein. In order to understand whether the C-terminal mutation of IKK ϵ v2 would have an effect on IKK ϵ dimerization, V5-tagged IKK ϵ v1 (IKK ϵ -WT) and Flag-tagged IKK ϵ isoforms were co-transfected into HEK293T cells followed by immunoprecipitation. As shown in Supporting Figure 2, three Flag-tagged IKK ϵ isoforms (v1, v2, v3) immunoprecipitated V5-tagged IKK ϵ -WT equally. In addition, we performed the in vitro pull assay to examine the direct interaction of each IKK ϵ isoform and IRF7. We observed an approximately equal amount of IRF7 interacting with each IKK ϵ isoform (Revised Supplementary Fig. 5d).

Previous studies have clearly demonstrated that enteroviruses infections promoted polyubiquitination, and we have demonstrated that IRF7 preferentially interacted with IKK ϵ v2 in the presence of ubiquitin. We further examined the interaction of IKK ϵ isoforms and IRF7 in EV71 infection. During EV71 infection, Flag-IRF7 immunoprecipitated more IKK ϵ v2 compared to IKK ϵ v1 and IKK ϵ v3 (Revised Fig. 3c). Furthermore, the higher phosphorylation of IRF7 Ser471/472 residues was detected in IKK ϵ v2

transfectants than in IKK ϵ v1 and IKK ϵ v3 transfectants in EV71 infection (Revised Fig. 3c).

In summary, the C-terminal mutation of IKK ϵ v2 does not hinder IKK ϵ dimerization and the direct interaction with IRF7, while IKK ϵ v2 could much strongly interacts with IRF7 compared to IKK ϵ v1 and v3 during EV71 infection.

Supporting Figure 2. The binding ability of IKK ϵ isoforms with IKK ϵ v1.

V5-tagged IKK ϵ -v1 and Flag-tagged IKK ϵ isoforms were co-transfected into HEK293T cells. Cells lysates were immunoprecipitated with Flag-agarose and V5-tagged IKK ϵ -v1 were detected with an anti-V5 antibody (top panel). V5 and Flag-tagged IKK ϵ in whole cell lysate (Lysate) were also detected as indicated in the lower panel.

We have described these results in the Results section and added new Supplementary Figure 5d and new Figure 3c in the revised manuscript as follows.

Revised Results:

(Page 16, Line 4-8)

Moreover, the in vitro pull-down assay showed that C-terminal mutation of IKK ϵ v2 did not affect the direct interaction between IKK ϵ and IRF7 (Supplementary Fig. 5d). These evidences clearly demonstrated that IRF7 preferentially interacts with IKK ϵ v2 in the presence of ubiquitin.

(Page 14, Line 5-10)

To characterize the role of IKK ϵ isoform switching in EV71 infection, we examined the phosphorylation and expression of IRF7 in each ectopically IKK ϵ isoform-expressing RD cells. IKK ϵ v2 strongly induced IRF7 phosphorylation (Fig. 3c), whereas the RNA and protein expression levels of IRF7 were unchanged in each IKK ϵ transfectant during EV71 infection (Supplementary Fig. 4c).

Revised Supplementary Figure 5d

Supplementary Fig. 5: The interaction of IKK isoforms and IRF7.

d. The direct interaction of IRF7 and IKK ϵ isoforms. Each Flag-IKK ϵ isoform was purified by anti-Flag beads from Flag-IKK ϵ -expressed HEK293T cells and incubated with 1 μ g of His-IRF7, which was purified from *E. coli*. for 2 hours at 4°C. After washing for three times, the bound His-IRF7 was analyzed by Western blotting with anti-His antibody.

Revised Figure 3c

Fig. 3: IKK ϵ v2 increases IRF7 phosphorylation and IRF7 translocation in the presence of ubiquitin.

c. IKK ϵ v2 strongly phosphorylates IRF7 in EV71 infection. Flag-IRF7 and each V5-IKK ϵ isoform were ectopically expressed in RD cells followed by EV71 infection. The Flag-IRF7 was immunoprecipitated with anti-Flag beads

and the phosphorylation was detected by IRF7-pS471/472 antibody. The co-immunoprecipitated V5-IKK ϵ isoform was analyzed by anti-V5 antibody.

4. For figure 3, the experiments don't support the claim. Increasing the overall mono-ubiquitin level alone will not mimic viral infection since these monoubiquitin needs to be assembled into specific ub-chains and be targeted to specific proteins upon E3 ligase activation (e.g. TRIM25 for RIG-I activation and TRAF proteins downstream of MAVS). But it certainly may sensitize the signaling as it increase then availability of ubiquitin. Therefore, the experiment at least should be done with viral infection in IKK ϵ knockout and Isoform-reconstituted cells (maybe under this condition you don't have to overexpress ubiquitin to see an effect). However, if you really want to claim IKKi V2 phosphorylates and activates IRF7 in the presence of ubiquitin, an in vitro system with all the purified components would be necessary.

Author reply:

Thank you for your constructive comments.

We totally agree your suggestions. The addition of ubiquitin alone can not mimic viral infection. Hence, we measured the phosphorylation of IRF7 Ser471/472 in IKK ϵ isoform-expressed RD cells during EV71 infection. The data indicated that the phosphorylation of IRF7 Ser471/472 was enhanced by IKK ϵ v2 compared to IKK ϵ v1, IKK ϵ v3 and Ctrl group at 8 h.p.i..

We have described these results in the Results section and added new Figure 3c in the revised manuscript as follows.

Results: (Page 14, Line 5-10)

To characterize the role of IKK ϵ isoform switching in EV71 infection, we examined the phosphorylation and expression of IRF7 in each ectopically IKK ϵ isoform-expressing RD cells. IKK ϵ v2 strongly induced IRF7 phosphorylation (Fig. 3c), whereas the RNA and protein expression levels of IRF7 were unchanged in each IKK ϵ transfectant during EV71 infection (Supplementary Fig. 4c).

Revised Figure 3c

Fig. 3: IKK ϵ v2 increases IRF7 phosphorylation and IRF7 translocation in the presence of ubiquitin.

c. IKK ϵ v2 strongly phosphorylates IRF7 in EV71 infection. Flag-IRF7 and each V5-IKK ϵ isoform were ectopically expressed in RD cells followed by EV71 infection. The Flag-IRF7 was immunoprecipitated with anti-Flag beads and the phosphorylation was detected by IRF7-pS471/472 antibody. The co-immunoprecipitated V5-IKK ϵ isoform was analyzed by anti-V5 antibody.

References:

- Cohen, E.M., Avital, N., Shamay, M., and Kobilier, O. (2020). Abortive herpes simplex virus infection of nonneuronal cells results in quiescent viral genomes that can reactivate. *Proc Natl Acad Sci U S A* 117, 635-640.
- Ho, B.C., Yu, S.L., Chen, J.J., Chang, S.Y., Yan, B.S., Hong, Q.S., Singh, S., Kao, C.L., Chen, H.Y., Su, K.Y., et al. (2011). Enterovirus-induced miR-141 contributes to shutoff of host protein translation by targeting the translation initiation factor eIF4E. *Cell Host Microbe* 9, 58-69.
- Koop, A., Lepenies, I., Braum, O., Davarnia, P., Scherer, G., Fickenscher, H., Kabelitz, D., and Adam-Klages, S. (2011). Novel splice variants of human IKKepsilon negatively regulate IKKepsilon-induced IRF3 and NF-kB activation. *Eur J Immunol* 41, 224-234.
- Nakatsu, Y., Matsuoka, M., Chang, T.H., Otsuki, N., Noda, M., Kimura, H., Sakai, K., Kato, H., Takeda, M., and Kubota, T. (2014). Functionally distinct

effects of the C-terminal regions of IKKepsilon and TBK1 on type I IFN production. PLoS One 9, e94999.

van Kuppeveld, F.J., Hoenderop, J.G., Smeets, R.L., Willems, P.H., Dijkman, H.B., Galama, J.M., and Melchers, W.J. (1997). Coxsackievirus protein 2B modifies endoplasmic reticulum membrane and plasma membrane permeability and facilitates virus release. EMBO J 16, 3519-3532.

Verhelst, K., Verstrepen, L., Carpentier, I., and Beyaert, R. (2013). IkkappaB kinase epsilon (IKKepsilon): a therapeutic target in inflammation and cancer. Biochem Pharmacol 85, 873-880.

Xu, L.J., Jiang, T., Zhang, F.J., Han, J.F., Liu, J., Zhao, H., Li, X.F., Liu, R.J., Deng, Y.Q., Wu, X.Y., et al. (2013). Global transcriptomic analysis of human neuroblastoma cells in response to enterovirus type 71 infection. PLoS One 8, e65948.

Reviewer #2 (Remarks to the Author):

In this paper, author found that IKK ϵ isoforms mRNA expression pattern was affected by EV70 infection then they found IKK ϵ V2 mRNA is up-regulated but IKK ϵ V3 is down-regulated in RD cell line. IKK ϵ V2 showed pre-dominantly interact with IRF7 and phosphorylate IRF7 for induction of ISGs compared to other IKK ϵ isoforms and that was enhanced by overexpressed ubiquitine-mediated K63 poly-ubiquitination. The manuscript is well written and figures are presented nicely. RNA-Seq data set would be useful for research in this field. RNA-Seq based finding of IKK ϵ isoforms and its switching idea is interesting but there are some issues that require authors attention.

Major points:

1) In Figure 1, further RNA-Seq analysis is needed for providing the information in manuscript. Author should show the heat map of ISGs and inflammatory cytokines expression including IFN β , OAS1, ISG56 between mock, 3hpi, 8hpi to show the immune response against EV70 that authors used down-stream target genes of IKK ϵ isoforms and analyzed promoter activity in Figure 2. Author also should make the graph (-log P-value of each GO clusters) of GO database clusters analysis to clearly show their listed in Table 2. Because current Table 2 is just list of statistically different GO name (Table 2), it is not convenient and lack the kindness for paper reading. Author also should combine the analysis of viral derived RNAs in RNA-Seq.

Authors reply:

Thank you for your advice.

Given there are many reports to investigate the impact of EV71 infection on host's response by transcriptomic analysis, in this study we focused to explore the biological meaning of isoform switching during EV71 infection. We apologize that we did not provide more information of RNA-Seq. To provide comprehensive information of RNA-Seq in this revised manuscript, we not only provided "-log P-value" of each pathway and also listed all of genes within each pathway (revised Supplementary Table 2) The readers can easily found the relative expression data of those genes in the interested pathways (Supplementary Table1). Moreover, to provide more assistance for the readers we deposited all of RNA-Seq data (mock, 4 h.p.i., 8 h.p.i.) which can be found at <http://bioinfo.cs.ccu.edu.tw/bioinfo/EV71/> (please see below). By accessing

this website, the readers can free download RNA-Seq data and process this data based as they wish.

Index of /bioinfo/EV71

Name	Last modified	Size	Description
 Parent Directory		-	
 EV71_FCA_0.tar.gz	2016-02-02 00:24	62G	
 EV71_FCA_4.tar.gz	2016-02-02 00:33	58G	
 EV71_FCA_8.tar.gz	2016-02-02 00:42	61G	

Supporting Figure 1

The directory of deposited RNA-Seq data.

We have revised the Methods section and Supplementary Table 2 as follows.

Revised Methods: (Page 29, Line 11-13)

The RNA-Seq (mock 0, 4, 8) of this study can be found at <http://bioinfo.cs.ccu.edu.tw/bioinfo/EV71/>.

Supplementary Table 2. Enriched pathways with differentially expressed genes in EV71 infection

Rank	Pathway	-Log p value	Genes
1	Immune response_TLR signaling pathway	7.655	TAB2, UEV1A, IL-6, Ubiquitin, AP-1, c-Jun/c-Fos, IKK-alpha, c-Jun, IL-12 alpha, CD14, TNF-alpha, HSP70
2	Immune response_IL-1 signaling pathway	7.637	TAB2, Endothelin-1, UEV1A, IL-6, Ubiquitin, AP-1, IKK-alpha, PAI1, c-Jun, c-Jun/c-Jun, TNF-alpha
3	Immune response_TLR2 and TLR4 signaling pathways	6.396	TAB2, UEV1A, IL-6, Ubiquitin, AP-1, c-Jun/c-Fos, IKK-alpha, c-Jun, CD14, TNF-alpha, TLR10
4	Cell cycle_ESR1 regulation of G1/S transition	5.586	Skp2/TrCP/FBXW, Ubiquitin, c-Fos, c-Jun/c-Fos, c-Jun, Cyclin A, c-Myc, CRM1
5	Immune response_TLR5, TLR7, TLR8 and TLR9 signaling pathways	5.221	TAB2, UEV1A, IL-6, Ubiquitin, AP-1, c-Jun/c-Fos, IKK-alpha, c-Jun, TNF-alpha
6	Immune response_IL-33 signaling pathway	4.587	TAB2, IL-6, Ubiquitin, Histone H2A, Histone H2B, IKK-alpha, c-Jun, TNF-alpha, IL-13
7	Signal transduction_PTMs in BAFF-induced non-canonical NF-kB signaling	4.451	Skp2/TrCP/FBXW, UEV1A, Ubiquitin, IKK-alpha, c-IAP2, c-IAP1, BAFF(TNFSF13B)
8	Development_TGF-beta-dependent induction of EMT via MAPK	4.378	Endothelin-1, NOX4, AP-1, c-Fos, c-Jun/c-Fos, PAI1, c-Jun, SNAIL1
9	Immune response_IL-2 activation and signaling pathway	4.243	AP-1, c-Fos, c-Jun/c-Fos, IKK-alpha, c-Myc, SHP-1, SOCS3, EGR1
10	Development_GM-CSF signaling	4.178	c-Fos, IKK-alpha, GM-CSF receptor, CSF2RB, c-Myc, CSF2RA, EGR1, Hck
11	Apoptosis and survival_Role of PKR in stress-induced apoptosis	3.991	TAB2, C/EBP zeta, IKK-alpha, TNF-alpha, PP2A regulatory, c-Myc, ATF-4, ATF-3

Rank	Pathway	-Log p value	Genes
13	Immune response_CCL2 signaling	3.932	IL-6, AP-1, c-Fos, c-Jun/c-Fos, c-Jun, TNF-alpha, CCBP2 (CCR9), Claudin-5
14	Immune response_CD16 signaling	3.915	Calcineurin B (regulatory), TNF-alpha, SHP-1, Cytohesin1
15	Immune response_IL-3 activation and signaling pathway	3.727	AP-1, c-Fos, c-Jun/c-Fos, SHP-1, SOCS3, EGR1
16	NETosis in SLE	3.727	Histone H4, Histone H1, Histone H2A, Histone H2, Histone H1.2, Leukocyte elastase
17	Effect of H. pylori infection on gastric epithelial cell proliferation	3.709	Histone H4, Skp2/TrCP/FBXW, AP-1, c-Jun, Calcineurin B (regulatory), c-Myc, HB-EGF, EGR1
18	Immune response_TNF-R2 signaling pathways	3.639	AP-1, c-Jun/c-Fos, IKK-alpha, c-Jun, c-IAP2, c-IAP1, TNF-alpha
19	Immune response_IL-18 signaling	3.605	TAB2, IL-6, AP-1, c-Fos, c-Jun/c-Fos, IKK-alpha, c-Jun, TNF-alpha
20	Expression targets of tissue factor signaling in cancer	3.527	PAI1, CTGF, Coagulation factor V, Coagulation factor X, Cyr61
21	Immune response_Histamine H1 receptor signaling in immune response	3.460	IL-6, c-Fos, c-Jun/c-Fos, IKK-alpha, c-Jun, Calcineurin B (regulatory), TNF-alpha
22	Neuroprotective action of lithium	3.456	nNOS, WNT, c-Jun, Calcineurin B (regulatory), NR2, NR2B, Frizzled, HSP70
23	Transcription_N-CoR/ SMRT complex-mediated epigenetic gene silencing	3.404	Histone H4, Ubiquitin, Histone H2B, AP-1, c-Fos, c-Jun/c-Fos, c-Jun
24	Immune response_HMGB1/TLR signaling pathway	3.356	TAB2, UEV1A, IL-6, Ubiquitin, IKK-alpha, TNF-alpha
25	Signal transduction_NF-kB activation pathways	3.295	TAB2, Ubiquitin, IKK-alpha, c-IAP2, c-IAP1, TNF-alpha, BAFF(TNFSF13B)
26	Development_WNT signaling pathway (part 2)	3.191	PYGO2, WNT, c-Jun, Frizzled, SNAIL1, c-Myc, ENC1

Rank	Pathway	-Log p value	Genes
27	Cell cycle_Regulation of G1/S transition (part 2)	3.169	c-Fos, IKK-alpha, c-Jun, Cyclin A, p107
28	Signal transduction_PTMs (ubiquitination and phosphorylation) in TNF-alpha-induced NF-kB signaling	3.162	TAB2, UEV1A, Ubiquitin, c-IAP2, c-IAP1, TNF-alpha
29	Apoptosis and survival_Anti-apoptotic TNFs/NF-kB/IAP pathway	3.090	IKK-alpha, c-IAP2, c-IAP1, TNF-alpha, TACI(TNFRSF13B)
30	Immune response_Role of PKR in stress-induced antiviral cell response	2.998	TAB2, IL-6, IKK-alpha, c-Jun, TNF-alpha, BAFF(TNFSF13B), c-Myc
31	Role of cell adhesion in vaso-occlusion in Sickle cell disease	2.931	ITGAM, Thrombospondin 1, CD14, TNF-alpha, Fc gamma RI, P-selectin
32	Signal transduction_PTMs in BAFF-induced canonical NF-kB signaling	2.931	TAB2, UEV1A, Ubiquitin, IKK-alpha, BAFF(TNFSF13B), TACI(TNFRSF13B)
33	Regulation of Tissue factor signaling in cancer	2.931	AP-1, c-Jun, c-Jun/c-Jun, TNF-alpha, EGR1, P-selectin
34	Immune response_TREM1 signaling pathway	2.908	IL-6, Ubiquitin, c-Fos, c-Jun/c-Fos, IKK-alpha, TREM1, TNF-alpha
35	Colorectal cancer (general schema)	2.873	WNT, IL-6, Frizzled, TNF-alpha, Ephrin-B receptors
36	Development_G-CSF-induced myeloid differentiation	2.873	ITGAM, c-Jun, c-Myc, SHP-1, SOCS3
37	Apoptosis and survival_Role of IAP-proteins in apoptosis	2.807	Ubiquitin, c-IAP2, c-IAP1, TNF-alpha, HSP70
38	Signal transduction_PTMs in IL-17-induced CIKS-dependent MAPK signaling pathways	2.743	TAB2, UEV1A, Ubiquitin, AP-1, IKK-alpha
39	Development_Regulation of epithelial-to-mesenchymal transition (EMT)	2.699	WNT, Endothelin-1, PAI1, c-Jun, Frizzled, SNAIL1, TNF-alpha

Rank	Pathway	-Log p value	Genes
40	Immune response_ETV3 affect on CSF1-promoted macrophage differentiation	2.682	AP-1, c-Fos, c-Jun/c-Fos, c-Jun, c-Myc
41	Signal transduction_Activin A signaling regulation	2.682	Histone H4, Ubiquitin, Histone H2, Activin A, SMAD7
42	Signal transduction_PTMs in IL-12 signaling pathway	2.678	GADD45 beta, Ubiquitin, c-Jun, IL-12 alpha, c-Myc, SOCS3
43	Immune response_IL-22 signaling pathway	2.622	c-Fos, c-Jun/c-Fos, c-Jun, c-Myc, SOCS3
44	Immune response_C5a signaling	2.586	AP-1, c-Fos, c-Jun/c-Fos, PAI1, c-Jun, TNF-alpha
45	PDE4 regulation of cyto/chemokine expression in inflammatory skin diseases	2.586	IL-6, IL-12 alpha, MIG, TNF-alpha, IL23A, IL-13
46	Immune response_MIF-mediated glucocorticoid regulation	2.528	IL-6, c-Fos, c-Jun, TNF-alpha
47	HBV signaling via protein kinases leading to HCC	2.510	AP-1, c-Fos, c-Jun/c-Fos, c-Jun, c-Myc
48	Cell adhesion_ECM remodeling	2.499	PAI1, MMP-12, MMP-15, Kallikrein 1, HB-EGF, Kallikrein 3 (PSA)
49	Immune response_IL-12 signaling pathway	2.454	Ubiquitin, c-Jun, SOCS3, P-selectin
50	Proteolysis_Putative ubiquitin pathway	2.454	FBXW7, UEV1A, Ubiquitin, HSP70

2) In Figure 1, It is needed to show the protein expression of isoforms by using Western blot or Mass Spectrometry to consider the following figures such as analyzing down-stream signaling of IKK ϵ isoforms. If it is difficult to show the protein expression, at least author should address endogenous IKK ϵ isoforms by using their siRNA which used in Figure 2f to show the accuracy of their detection quality by using ddPCR.

Authors reply:

We appreciate your constructive comment.

Based on a previous report, the protein expression of IKK ϵ v1 and v2 could be detected by IKK ϵ antibody (Cell Signaling) (Koop et al., 2011). In order to precisely examine the newly synthesized endogenous IKK ϵ isoform switching in EV71 infection, the Click-iT AHA assay, a non-radioactive method for the detection of nascent protein, was performed. As shown in revised Figure 1c, the isoform switching of IKK ϵ v1 and v2 was confirmed at both RNA and

protein levels.

We have added the revised descriptions in the Results and Methods sections of the revised manuscript, and added new Supplementary Figure 1c as follows.

Revised Results: (Page 9, Line 3-8)

To address whether the IKK ϵ isoform switching can be detected in protein level, we performed a Click-iT AHA assay to detect nascent IKK ϵ isoforms. Biotin-labeled newly synthesized proteins at indicated time points postinfection were purified, and IKK ϵ isoforms were detected by Western blotting. The expression of IKK ϵ v2 increased while the v1 decreased after EV71 infection, consistent with our RNA data (Fig. 1c).

Revised Methods: (Page 22, Line 12 – Page 23, Line 4)

Click-iT AHA assay. RD cells were cultured in glutamine, methionine, and cystine free medium supplemented with 50uM Click-iT AHA (L-Azidohomoalanine) (Invitrogen). After incubation, AHA was taken up by cells and loaded onto methionine tRNAs. During translation, AHA is incorporated into newly synthesized proteins. A biotin-based tag is then added by click chemistry according to manufacturer's instructions, and the newly synthesized protein was precipitated using streptavidin dynabeads (Invitrogen) followed by Western blotting analysis with antibody against IKK ϵ (1:1000, Cell Signaling).

Revised Figure 1c

Fig. 1: EV71 infection triggers IKK ϵ isoform switching.

c. IKK ϵ isoform switching is confirmed by Western blotting. The Click-iT AHA assay was performed to measure newly synthesized IKK ϵ v1 and v2. The

synthesis of IKK ϵ v2 was increased while IKK ϵ v1 was decreased in EV71 infection.

3) In Figure 2e, authors analyzed the effect of IKK ϵ isoforms over-expression on EV70 infection. To show the importance of endogenous produced IKK ϵ V2 on ISGs induction and host defense against EV70 infection, authors should measure the EV70 viral titer and ISGs mRNA expression by using s IKK ϵ V2 iRNA, which they used in Figure 2f, without any overexpression plasmid. This is important issue to consider the function of IKK ϵ isoform switching in physiological condition.

Author reply:

We appreciate your constructive comment and apologize for the unclear label of the siRNAs against IKK ϵ .

The siRNAs used in Figure 2f are not variant-specific. siRNA-IKK ϵ -1 and siRNA-IKK ϵ -2 are two unique siRNAs that target non-overlapping regions of all the three IKK ϵ isoforms. Hence, we cannot use these siRNAs to knock down the endogenous variant mRNAs. To avoid the misunderstanding caused by unclear label we changed the name of “siRNA-IKK ϵ -1” to “siRNA1-IKK ϵ ” and the name of “siRNA-IKK ϵ -2” to “siRNA2-IKK ϵ ” in the revised manuscript and revised Figure 2f.

It is an excellent strategy to use isoform-specific siRNA for studying the importance of IKK ϵ v2 in ISGs induction and host defense against EV71 infection, however, to the best of our knowledge, it is impossible to design isoform-specific siRNAs. The strategy used to design the specific siRNAs was described below:

(1) siRNA design for IKK ϵ isoform 2

The exon 20 annotated in IKK ϵ isoform 1 (NM_014002) was lacked in IKK ϵ isoform 2 (NM_001193322), the exon 19 and exon 21 spanning was used for isoform-specific siRNA design (please see below Figure).

Exon 19 sequence

GGTGGTGCACGAGACCAGGAACCACCTGCGCCTGGTTGGCTGTTCTGT

GGCTGCCTGTAACACAGAAGCCCAGGGGGTCCAGGAGAGTCTCAGCAA
G

Exon 21 sequence

CATGCAAGAGCTCTGCGAGGGGATGAAGCTGCTGGCATCTGACCTCCTG
GACAACAACCGCATCATCGAACG

Exon 19 and 21 spanning

GGTGGTGCACGAGACCAGGAACCACCTGCGCCTGGTTGGCTGTTCTGT
GGCTGCCTGTAACACAGAAGCCCAGGGGGTCCAGGAGAGTCTCAGCAA
GCATGCAAGAGCTCTGCGAGGGGATGAAGCTGCTGGCATCTGACCTCCT
GGACAACAACCGCATCATCGAACG

Custom Dice-Substate siRNA (DsiRNA) designer (IDT, https://sg.idtdna.com/site/order/designtool/index/DSIRNA_PREDESIGN) was used for siRNA design and all designed siRNAs fulfilling design criteria were listed as follows. However, none of designed siRNA located at exon 19-21 spanning.

Designed siRNA	Sequence position	Sequence	Exon 19-21 spanning
1	132-157	5' GCAUCUGACCUCCUGGACAACAACC 3'	No
2	143-168	5' CCUGGACAACAACCGCAUCAUCGAA 3'	No
3	133-158	5' CAUCUGACCUCCUGGACAACAACCG 3'	No
4	144-169	5' CUGGACAACAACCGCAUCAUCGAAC 3'	No
5	134-159	5' AUCUGACCUCCUGGACAACAACCGC 3'	No
6	141-166	5' CUCCUGGACAACAACCGCAUCAUCG 3'	No
7	136-161	5' CUGACCUCCUGGACAACAACCGCAT 3'	No
8	131-156	5' GGCAUCUGACCUCCUGGACAACAAC 3'	No
9	140-165	5' CCUCCUGGACAACAACCGCAUCATC 3'	No

(2) siRNA design for IKKε isoform 3

The exon 3 annotated in IKKε isoform 1 (NM_014002) was lacked in IKKε isoform 3 (NM_001193321), the exon 2 and exon 4 spanning was used for isoform-specific siRNA design (please see below Figure)..

Exon 2 sequence

CTCAGCTCCTGGACGTGCCACAGACAGAAAGCATAACATACTCGCCA
GGAAGAGCCTTTGCCTGACTCAGGGCAGCTCAGAGTGTGGG

Exon 4 sequence

AAATCCGGAGAGCTGGTTGCTGTGAAGGTCTTCAACACTACCAGCTACCT
GCGGCCCGCGAGGTGCAAGTGAGGGAGTTTGAGGTCCTGCGGAAGCT
GAACCACCAGAACATTGTCAAGCTCTTTGCGGTGGAGGAGACG

Exon 2 and 4 spanning

CTCAGCTCCTGGACGTGCCACAGACAGAAAGCATAACATACTCGCCA
GGAAGAGCCTTTGCCTGACTCAGGGCAGCTCAGAGTGTGGGAAATCCG
GAGAGCTGGTTGCTGTGAAGGTCTTCAACACTACCAGCTACCTGCGGCC
CCGCGAGGTGCAAGTGAGGGAGTTTGAGGTCCTGCGGAAGCTGAACCA
CCAGAACATTGTCAAGCTCTTTGCGGTGGAGGAGACG

Custom Dice-Substate siRNA (DsiRNA) designer (IDT, https://sg.idtdna.com/site/order/designtool/index/DSIRNA_PREDESIGN) was used for siRNA design and all designed siRNAs fulfilling design criteria were listed as follows. However, none of designed siRNA located at exon 2-4 spanning.

Designed siRNA	Sequence position	Sequence	Exon 2-4 spanning
1	189-214	5' GAACCACCAGAACAUUGUCAAGCTC 3'	No
2	185-210	5' AGCUGAACCACCAGAACAUUGUCA 3'	No
3	190-215	5' AACCACCAGAACAUUGUCAAGCUCT 3'	No
4	114-139	5' GAAGGUCUUAACACUACCAGCUAC 3'	No
5	113-138	5' UGAAGGUCUUAACACUACCAGCTA 3'	No
6	184-209	5' AAGCUGAACCACCAGAACAUUGUCA 3'	No
7	106-131	5' GUUGCUGUGAAGGUCUUAACACTA 3'	No
8	186-211	5' GCUGAACCACCAGAACAUUGUCAAG 3'	No
9	117-142	5' GGUCUUAACACUACCAGCUACCTG 3'	No
10	112-137	5' GUGAAGGUCUUAACACUACCAGCT 3'	No
11	115-140	5' AAGGUCUUAACACUACCAGCUACC 3'	No
12	192-217	5' CCACCAGAACAUUGUCAAGCUCUTT 3'	No
13	118-143	5' GUCUUAACACUACCAGCUACCUGC 3'	No
14	195-220	5' CCAGAACAUUGUCAAGCUCUUUGCG 3'	No
15	187-212	5' CUGAACCACCAGAACAUUGUCAAGC 3'	No
16	191-216	5' ACCACCAGAACAUUGUCAAGCUCTT 3'	No
17	116-141	5' AGGUCUUAACACUACCAGCUACCT 3'	No

18	193-218	5' CACCAGAACAUUGUCAAGCUCUUTG 3'	No
19	183-208	5' GAAGCUGAACCACCAGAACAUUGTC 3'	No

We have revised the sections of Results and Methods as well as Figure 2f in revised descriptions as follows.

Revised Results: (Page 12, Line 8-11)

In contrast, the attenuation of virus titers observed in IKK ϵ v2 transfectants was greatly eliminated by two IKK ϵ siRNAs, which target two common regions of IKK ϵ isoforms, respectively (Fig. 2f).

Revised Methods: (Page 22, Line 8-11)

siRNA transfection. RD cells were transiently transfected with siRNA1-IKK ϵ or siRNA2-IKK ϵ (s18537, s18538, ThermoFisher) at a final concentration of 50 nM using RNAiMAX (Invitrogen). Further treatments or assays were generally performed 48 h after siRNA transfection.

Revised Figure 2f

Fig. 2: IKK ϵ v2 increases IRF7-mediated IFN β and ISGs expressions in EV71 infection and attenuates virus propagation.

f. Attenuation of virus titer in IKK ϵ v2 transfectants is restored by IKK ϵ siRNAs. The siRNAs against IKK ϵ , siRNA1-IKK ϵ or siRNA2-IKK ϵ , were introduced into RD cells expressing each Flag-IKK ϵ isoform followed by EV71 infection. The viral titers and ectopic IKK ϵ isoform expressions were determined by plaque assay and Western blotting, respectively. β -actin was served as an internal control. All data presented are mean \pm SD (n=3). * and ** represent p value <0.05 as compared with siRNA ctrl group.

4) In Figure 2f, author did not use siRNA for IKK ϵ V3 which is not involved in switching IKK ϵ isoforms story in this paper. It was also not affected by EV70 infection in Figure 1b and might be difficult to design the siRNA. Author should comment the reason why they skip IKK ϵ V3 knockdown in Figure 2f.

Authors reply:

We terribly apologize for the unclear label of the siRNAs against IKK ϵ .

The siRNAs used in Figure 2f are not variant-specific. siRNA-IKK ϵ -1 and siRNA-IKK ϵ -2 are two unique siRNAs that target non-overlapping regions of all the three IKK ϵ isoforms. The reason why we did not use IKK ϵ isoform-specific siRNAs in this study has explained. Please see “Authors reply” to comment 3.

5) In Figure 3, Author should show the IRF7 phosphorylation in IKK ϵ isoforms over-expressed RD cell infected with EV70 which they could see the effect of IKK ϵ V2 in Figure 2f.

Author reply:

Thank you for your constructive comments.

Based on your suggestion, we measured the phosphorylation of IRF7 Ser471/472 in IKK ϵ isoform-expressed RD cells during EV71 infection. The data indicated that the phosphorylation of IRF7 Ser471/472 was enhanced by IKK ϵ v2 compared to IKK ϵ v1, IKK ϵ v3 and Ctrl group at 8 h.p.i..

We have described these results in the Results section and added new Figure 3c in the revised manuscript as follows.

Revised Results: (Page 14, Line 5-10)

To characterize the role of IKK ϵ isoform switching in EV71 infection, we examined the phosphorylation and expression of IRF7 in each ectopically IKK ϵ isoform-expressing RD cells. IKK ϵ v2 strongly induced IRF7 phosphorylation (Fig. 3c), whereas the RNA and protein expression levels of IRF7 were unchanged in each IKK ϵ transfectant during EV71 infection (Supplementary Fig. 4c).

Revised Figure 3c

Fig. 3: IKK ϵ v2 increases IRF7 phosphorylation and IRF7 translocation in the presence of ubiquitin.

c. IKK ϵ v2 strongly phosphorylates IRF7 in EV71 infection. Flag-IRF7 and each V5-IKK ϵ isoform were ectopically expressed in RD cells followed by EV71 infection. The Flag-IRF7 was immunoprecipitated with anti-Flag beads and the phosphorylation was detected by IRF7-pS471/472 antibody. The co-immunoprecipitated V5-IKK ϵ isoform was analyzed by anti-V5 antibody.

6) It is difficult to see the difference of nuclear IRF7 band intensity in Western blot data. Author should calculate the intensity and label the value or make a graph under each panel (Figure 3c and 3d). And to check the exact nuclear translocation of IRF7, author should show nuclear IRF7 by using confocal microscopy by staining IRF7 and IKK ϵ in the condition of Figure 3d.

Author reply:

Thank you for the kind suggestions.

We have quantified the intensity of each band by densitometer and added the information to the revised Figure. 3d and Figure 3e. We also observed IRF7 nuclear translocation by confocal microscopy and showed the immunofluorescence images and quantified data in Supplementary Figure 4f and 4g. Exactly, there is an obvious IRF7 nuclear translocation found in IKK ϵ v2 expressed cells.

We have added the confocal results in Results and Methods sections. We also added new Supplementary Figure 4f and 4g and revised Figure 3d and 3e as follows.

Revised Results: (Page 14, Line 14 – Page 15, Line 5)

To test this hypothesis, nucleus-cytoplasm fractionation and immunofluorescence were conducted, and more V5-tagged IRF7 was found to accumulate in the nucleus of cells expressing IKK ϵ v2 than in those expressing IKK ϵ v1 or IKK ϵ v3 in the presence of ubiquitin (Fig. 3d and Supplementary Fig. 4d,f,g). Similarly, IKK ϵ v2 also led to greater nuclear accumulation of endogenous IRF7 in the presence of ubiquitin compared with the other two IKK ϵ isoforms (Fig. 3e and Supplementary Fig. 4e). Taken together, IKK ϵ v2 dominantly phosphorylates and activates IRF7 rather than IKK ϵ v1 and IKK ϵ v3 during EV71 infection in a ubiquitin-dependent manner.

Methods: (Page 27, Line 11 – Page 28, Line 5)

Immunofluorescence staining. RD cells co-transfected with V5-tagged IRF7 and each Flag-tagged IKK ϵ isoform were fixed with 3.7% paraformaldehyde and permeabilized with 0.5% Triton-X100/PBS. After incubation in blocking buffer (1%BSA/PBS), slides were incubated sequentially with primary antibody (anti-Flag, Proteintech 20543-1-AP and anti-V5, Invitrogen #R961-25) and secondary antibody (Goat anti-Rabbit Alexa 594 and Goat anti-Mouse Alexa 488). Nuclei were counterstained with Hoechst33342 (Sigma 14533).

Image acquisition and analysis. Fluorescence images were captured on a spinning disk confocal (Zeiss). Fluorescence signals were captured using sequential acquisition to give separate image files for each slide. Image analysis was performed using MetaMorph (Molecular Devices). More than five fields were selected for analysis of each stain. The ratio of granules in nucleus/granules in cytoplasm was measured from at least fifty cells.

Revised Figure 3d and 3e

Fig. 3: IKKε v2 increases IRF7 phosphorylation and IRF7 translocation in the presence of ubiquitin.

d, e. IKKε v2 facilitates IRF7 translocation. HeLa cells were co-transfected with each Flag-IKKε isoform and V5-IRF7 in the presence or absence of HA-ubi. Nucleus and cytoplasm fractions obtained from HeLa cells were applied to immunoblot with anti-Flag and anti-V5 antibodies (d). Each Flag-IKKε isoform was transfected into HeLa cells along with HA-ubi. The cell lysates were adapted to nucleus and cytoplasm fractionation and immunoblot with anti-Flag and anti-IRF7 antibodies (e). H3 and α-tubulin were used as nuclear and cytosolic markers, respectively.

Revised Supplementary Figure 4f

Supplementary Fig. 4: IKK ϵ v2 increases IRF7 phosphorylation and IRF7 translocation in the presence of ubiquitin.

f. IKK ϵ v2 facilitates IRF7 translocation by immunofluorescence analysis. The immunofluorescence images of cells treated with the same condition used in Figure 3d.

Revised Supplementary Figure 4g

Supplementary Fig. 4: IKK ϵ v2 increases IRF7 phosphorylation and IRF7 translocation in the presence of ubiquitin.

g. The ratio of nuclear IRF7 to cytoplasmic IRF7 in Supplementary Figure 3f. The ratios of F- IKK ϵ isoforms were normalized to that of Ctrl.

7) About Figure 4 and 5, author should calculate the band intensity of V5-IRF7 in IP-Flag and normalized by V5 in lysate and indicate the value or graph for Figure 4a. The same manner calculation is needed for Figure 4b, 4c, 5a, 5b, 5c.

Author reply:

We apologize for the unclear description of the methods. We describe in detail how immunoprecipitation and co-immunoprecipitation performed as followed:

1. Lyse cells in 1X cell lysis buffer.
2. Determine the protein concentration of the cell lysate.
3. Prepare the lysate input with 40ug cell lysate.
4. Heat the lysate input to 95-100°C for 5 minutes and microcentrifuge for 1 minute at 14,000 X g.
5. Take at least 1.2 mg cell lysate and add Anti-FLAG® M2 affinity gel beads (5 µl of 50% bead slurry, Sigma A2220). Incubate with gentle rocking for 2 hours at 4°C.
6. Collect the beads by pulse centrifugation. Discard the supernatant and wash the beads 3 times with 1ml ice-cold 1X cell lysis buffer.
7. Resuspend the pellet with 20 µl SDS sample buffer. Vortex, then microcentrifuge for 30 seconds.
8. Heat the IP products to 95-100°C for 5 minutes and microcentrifuge for 1 minute at 14,000 X g.
9. Analyze sample by Western blotting.

Since the M2 beads were saturated by excess cell lysate, the phosphorylation levels and the associations of the immunoprecipitated target could be normalized by Flag in IP-Flag as the listed references (Luo et al., 2020; Wang et al., 2018). We counted the band intensity of V5-IRF7 in IP-Flag and normalized by V5 in IP-Flag. We then indicated the value in Revised Figure 4a. The calculation in the same manner is used for Figure 4b, 4c, 5a, 5b, 5c.

We revised the Figures as follows:

Revised Figure 4a and 4b

a

b

Revised Figure 4c

c

Revised Figure 5a

Revised Figure 5b

Revised Figure 5c

C

Minor points:

1) There is no information about Taqman probe and primer sequence, siRNA sequence, p-IRF7 antibody.

Authors reply:

We apologize for our carelessness.

We have added the information of siRNA, p-IRF7 antibody, Tagman probe and primer sequences in the Methods section and revised Supplementary Table 5 of the revised manuscript as follows.

Revised Methods:

(Page 22, Line8-11)

siRNA transfection. RD cells were transiently transfected with siRNA1-IKKε or siRNA2-IKKε (s18537, s18538, ThermoFisher) at a final concentration of 50 nM using RNAiMAX (Invitrogen). Further treatments or assays were generally performed 48 h after siRNA transfection.

(Page 26, Line 8-9)

phospho-IRF7(Ser471/472) (1:1000; Cell Signaling)

Revised Supplementary Table 5

Supplementary Table 5. List of primer and probe

Gene name	Sequence (5'-3')
IFNb-SYBR-F	ATTGCCTCAAGGACAGGATG
IFNb-SYBR-R	GGCCTTCAGGTAATGCAGAA

OAS1-SYBR-F	CCCCATTATTGAAAAGTACCTGAGA
OAS1-SYBR-R	GCCGGGTCCAGGATCAC
ISG56-SYBR-F	CAGAACGGCTGCCTAATTTACA
ISG56-SYBR-R	GTGGGTCTGCTTTTTCTCTGT
ISG20-SYBR-F	CCCTGCGGGTGCTGAGT
ISG20-SYBR-R	TGTCCAAGCAGGCTGTTCTG
Mxa-SYBR-F	GCT ACTGTGGCCAGAAAAATC
Mxa-SYBR-R	TCATACTGGCTGCACAGGTTGT
Promoter Luc reporter	Sequence (5'-3')
OAS1-F	CCCGGTACCCTTAACAAAAAGAAAAGAGAC
OAS1-R	TTTAAGCTTTTTACCACCTTGACACACA
ISG56-F	TAAGGTACCGCACCCAGCCAAGAATCATT
ISG56-R	CGCAAGCTTAGATCTGGCTATTCTGTCTT
ISG20-F	AAAGGTACCCCAAATCCCACTTGGTGAAA
ISG20-R	AAAAAGCTTCTCTCACCTGCCTGCCTCTG
Mxa-F	ACCGGTACCCCAAAGCTCACCAGTATCAA
Mxa-R	ATAAAGCTTCTCTGCTACCAGGCTGAGGA
Expression vector	Sequence (5'-3')
IRF7-HindIII-F	AAAAAGCTTATGGCCTTGGCTCCTGAGAGGGCAG
IRF7-BamHI-R	AAAGGATCCGGCGGGCTGCTCCAGCTCCATAAGG
Digital Probe	Target exon boundary
Hs01063858_m1	NM_014002.3 Exon 21-22
Hs01069870_m1	NM_014002.3 Exon 2-3
Hs01063855_g1	NM_014002.3 Exon 19-20

2) Author analyzed several ISGs mRNA expression using same samples (Figure 2b, 2d, S2a, S2b). To easier understanding of overall ISGs production, authors could make heat map in one panel to show the ISGs expression profile affected by IKK ϵ V2 over-expression.

Authors reply:

Thank you for the suggestion.

We have added the heat map of ISGs expression profile in ectopically IKK ϵ -expressed cells into the revised Results section of the revised manuscript and added new Supplementary Figure 2d as follows.

Revised Results: (Page 11, Line 7-10)

The results showed that both promoter activities and mRNA expression of ISG56 and OAS1 were obviously upregulated in RD cells ectopically

expressing IKK ϵ v2 at 8 h.p.i. (Fig. 2c,d and Supplementary Fig. 2d). The other two ISGs, ISG20 and Myxovirus resistance protein A (MxA), were slightly augmented in IKK ϵ v2 transfectants in both promoter activities and mRNA expression at 8 h.p.i. (Supplementary Fig. 2b-d).

Revised Supplementary Fig. 2d

Supplementary Fig. 2: IKK ϵ v2 transcriptionally induces ISGs expressions in EV71 infection through IRF7.

d. The heat map of ISGs expressions in ectopically IKK ϵ isoform-expressed cells.

3) Author should mention innate sensors such as RIG-I and MDA5 which recognize EV70 infection in introduction part.

Authors reply:

Thanks for your suggestions.

We have added RIG-1 and MDA5 description in the Introduction section of the revised manuscript as follows.

Revised Introduction: (Page 3, Line 17 – Page 4, Line 7)

The RLR family consists of three members, including RIG-I, melanoma differentiation-associated gene 5 (MDA5), and laboratory of genetics and physiology 2 (LGP2). RIG-1 and MDA5 have been identified as intracellular PRRs for RNA viruses to stimulate type-I IFN expression^{8,9}. Upon RNA ligand binding, RIG-I and MDA5 interact with the mitochondrial antiviral-signaling adaptor protein (MAVS) to trigger downstream I-Kappa-B Kinase Epsilon (IKK ϵ)/TANK binding kinase 1 (TBK1) and canonical NF- κ B signaling for activation of IFN- β and inflammatory cytokines, respectively^{10,11}.

Revised References: (Page 33, Line 27 – Page 34, Line 12)

8 Kang, D. C. et al. mda-5: An interferon-inducible putative RNA helicase with double-stranded RNA-dependent ATPase activity and melanoma growth-suppressive properties. *Proceedings of the National Academy of Sciences of the United States of America* 99, 637-642, doi:10.1073/pnas.022637199 (2002).

9 Yoneyama, M. et al. The RNA helicase RIG-I has an essential function in double-stranded RNA-induced innate antiviral responses. *Nat Immunol* 5, 730-737, doi:10.1038/ni1087 (2004).

10 Goubau, D. et al. Antiviral immunity via RIG-I-mediated recognition of RNA bearing 5'-diphosphates. *Nature* 514, 372-375, doi:10.1038/nature13590 (2014).

11 Jin, Y., Zhang, R., Wu, W. & Duan, G. Antiviral and Inflammatory Cellular Signaling Associated with Enterovirus 71 Infection. *Viruses* 10, doi:10.3390/v10040155 (2018).

4) What is potential mechanism of IKK ϵ isoform switching? Is there some transcription factor needed? Author can mention this point in discussion part.

Authors reply:

Thanks for your suggestions.

In this study, we first provided evidence of IKK ϵ isoform switching in virus infection, and we demonstrated that IKK ϵ isoform switching plays a role in innate immune response. Although we did not study what induces IKK ϵ isoform switching in this manuscript, we have extended the possible regulatory mechanisms in the revised manuscript.

We have added the description in the Discussion section of the revised manuscript as follows.

Revised Discussion: (Page 19, Line 9 – Page 20, Line 2)

RIG-1/MDA5, the upstream activator of IKK ϵ , was reported to be cleaved by EV71-encoded 2A and 3C protease^(3C^{pro}) and led to inhibition of the IFN- α/β response^{58,59}. However, upregulation of RIG-I ubiquitination promoted the expression of IFN- β and ISGs⁶⁰. Another study indicated that ARDC4 promoted K63 polyubiquitination of MDA5, consequently activating the innate immune response in EV71 infection⁶¹. In the present study, we found that IKK ϵ v2 showed higher activity in the presence of K63-linked ubiquitination, and IKK ϵ v2 promoted IRF7 activation under enhanced ubiquitination in EV71 infection. Whether RIG-1/MDA5 is involved in the regulation of IKK ϵ isoform

switching is remaining for further investigation.

Revised References: (Page 38, Line 27 – Page 39, Line 3)

58 Lei, X. et al. The 3C protein of enterovirus 71 inhibits retinoid acid-inducible gene I-mediated interferon regulatory factor 3 activation and type I interferon responses. *Journal of virology* 84, 8051-8061, doi:10.1128/JVI.02491-09 (2010).

59 Feng, Q. et al. Enterovirus 2Apro targets MDA5 and MAVS in infected cells. *Journal of virology* 88, 3369-3378, doi:10.1128/JVI.02712-13 (2014).

60 Chen, N. et al. Enterovirus 71 inhibits cellular type I interferon signaling by inhibiting host RIG-I ubiquitination. *Microb Pathog* 100, 84-89, doi:10.1016/j.micpath.2016.09.001 (2016).

61 Meng, J. et al. ARRDC4 regulates enterovirus 71-induced innate immune response by promoting K63 polyubiquitination of MDA5 through TRIM65. *Cell Death Dis* 8, e2866, doi:10.1038/cddis.2017.257 (2017).

5) Is it occurred in other viral infection and PRR ligand treatment? Author can mention this point in discussion part also.

Authors reply:

Thanks for your suggestions.

Based on your comments, we detected whether IKK ϵ isoform switching is happened in coxsackievirus B3 (CVB3) and (Herpes simplex virus-1) HSV-1 infected cells by RNA-Seq. The results indicated that the relative expression of IKK ϵ v2 is increased while IKK ϵ v1 is decreased in both CVB3 and HSV-1 infection (Revised Supplementary Fig. 1c).

We have revised the Results and Discussion of the revised manuscript and added new Supplementary Figure. 1c as follows.

Revised Results: (Page 8, Line 10 – Page 9, Line 2)

Furthermore, we investigated whether IKK ϵ isoform switching is a common characteristic in virus infections including RNA and DNA viruses. The expression of IKK ϵ isoforms was measured in HeLa cells infected with coxsackievirus B3 (CVB3) at 5 m.o.i. for 4 h.p.i. and 6 h.p.i.³⁰, and herpes simplex virus-1 (HSV-1) at 1 m.o.i. for 8 h.p.i. and 24 h.p.i.³¹ by RNA-Seq. The relative expression of IKK ϵ v2 is increased while IKK ϵ v1 is decreased both in CVB3 and HSV-1 infection (Supplementary Fig. 1c). CVB3 and HSV-1 infections induced IKK ϵ isoform switching in a similar pattern found in EV71 infection. These data indicated that isoform switching is a common feature during virus infection at least in the cases of EV71, CVB3 and HSV-1.

Revised Discussion: (Page 18, Line 5-17)

Type I IFNs provide a first line of defense against viral infections. Administration of IFNs can limit virus spreading at an early phase during virus infections, while many viruses prevent IFN attacks by the inhibition of IFN through several mechanisms. In our study, we demonstrated that IKK ϵ isoform switching occurred in EV71 infection. IKK ϵ v2 phosphorylated and thereby activated IRF7 to trigger IFN activation. We also examined IKK ϵ isoform switching in another enterovirus, CVB3, and in a DNA virus, HSV-1. The relative expression of IKK ϵ v2 was upregulated while IKK ϵ v1 was downregulated, upon CVB3 and HSV-1 infection. The relative abundance of the different IKK ϵ isoforms might represent a novel regulatory mechanism controlling the innate immune response. Further studies are required to comprehensively understand how virus infection induces IKK ϵ isoform switching.

Supplementary Figure 1c

Supplementary Fig. 1: The pathway enriched by isoform switching genes in EV71 infection.

c. IKK ϵ isoform switching is measured by RNA-Seq in CVB3 and HSV-1 infected HeLa cells. IKK ϵ v2 was up-regulated while IKK ϵ v1 was down-regulated in response to CVB3 (left panel) and HSV-1 (right panel) infection determined by RNA-Seq.

Revised References: (Page 36, Line 4-12)

30 van Kuppeveld, F. J. et al. Coxsackievirus protein 2B modifies

endoplasmic reticulum membrane and plasma membrane permeability and facilitates virus release. *EMBO J* 16, 3519-3532, doi:10.1093/emboj/16.12.3519 (1997).

31 Cohen, E. M., Avital, N., Shamay, M. & Kobiler, O. Abortive herpes simplex virus infection of nonneuronal cells results in quiescent viral genomes that can reactivate. *Proceedings of the National Academy of Sciences of the United States of America* 117, 635-640, doi:10.1073/pnas.1910537117 (2020).

References:

Koop, A., Lepenies, I., Braum, O., Davarnia, P., Scherer, G., Fickenscher, H., Kabelitz, D., and Adam-Klages, S. (2011). Novel splice variants of human IKKepsilon negatively regulate IKKepsilon-induced IRF3 and NF-kB activation. *Eur J Immunol* 41, 224-234.

Luo, H., Yu, Q., Liu, Y., Tang, M., Liang, M., Zhang, D., Xiao, T.S., Wu, L., Tan, M., Ruan, Y., et al. (2020). LATS kinase-mediated CTCF phosphorylation and selective loss of genomic binding. *Sci Adv* 6, eaaw4651.

Wang, C., Wang, H., Zhang, D., Luo, W., Liu, R., Xu, D., Diao, L., Liao, L., and Liu, Z. (2018). Phosphorylation of ULK1 affects autophagosome fusion and links chaperone-mediated autophagy to macroautophagy. *Nat Commun* 9, 3492.

Reviewer #3 (Remarks to the Author):

The manuscript compares the ability of three different IKK ϵ isoforms to induce IRF7 activation, IFN induction and limit EV71 replication. It is an interesting observation that EV71 infection induces isoform switching, and the manuscript also provides us with a better understanding of IKK ϵ domain function.

The data overall looks convincing and experiments well controlled.

I have some queries about the data as listed below:

1.1 Overall I find it surprising that IKK ϵ -v1 doesn't induce much of a response in most read-out systems, given that this is the full-length version that most researchers would have used in the past to study IKK ϵ . I would have expected overexpression of IKK ϵ -v1 to upregulate read-outs in Figure 2 more strongly, and to also limit viral replication upon overexpression. Did the authors check expression levels of the different versions in their reporter assays, qPCR and viral replication assays (apart from 2f).

Author reply:

Thank you for your comment.

We have checked the expression levels of different IKK ϵ isoforms accompanied with real-time PCR for IFN β , four ISGs and viral replication assays shown in Figure 2. We apologized we did not present the Western data in Figure 2 due to space limitation. Supporting Figure 1 showed that all of IKK ϵ isoforms are successfully expressed in transfected RD cells at mock infection and EV71 infection at 8 h.p.i. We also checked whether the virus infection is successful and the increase of cleaved PARP in EV71 infected RD cells indicated the virus infection works. Taken together, these data exclude the possibility that the different responses of IKK ϵ isoforms are caused by unequal expressions of IKK ϵ isoforms.

Supporting Figure 1. The expression level of IKKε referred to Fig. 2.

RD cells were transfected with different IKKε isoform-expressed vectors or empty vectors followed by EV71 infection. Total lysates were loaded to perform immunoblot with anti-Flag antibody. Cleaved PARP (PARP-C) served as an indicator for EV71 infection. β-actin served as an internal control.

1.2 I am equally surprised that IRF3 is not activated at all with overexpression of any IKKε variant. A positive control (TBK1?) would have been useful in this assay.

Author Reply:

Thank you for your comment.

To provide more convincing evidence for determining IRF3 or IRF7 plays the key role in IKKε v2-mediated IFNβ production, we performed IRF/IKKε immunoprecipitation assays by which the interaction between IKKε isoforms and IRF3/IRF7 was measured. We found that all the three IKKε can bind to IRF7 rather than IRF3. We thought it might be the reason why IRF3-driven promoter reporter was not activated in three IKKε isoform transfectants.

We have added IP data in Supplementary Figure 2a and revised descriptions in the Results section of the revised manuscript as follows.

Revised Results: (Page 10, Line 6-9)

Our results showed that IKKε v2 induced the luciferase activity of the IRF7 binding element-containing reporter vector, but not the IRF3 binding element (Fig. 2a), corresponding to the interaction of IRF3/7 and each IKKε isoform (Supplementary Fig. 2a).

Revised Supplementary Figure 2a

Supplementary Fig. 2: IKK ϵ v2 transcriptionally induces ISGs expressions in EV71 infection through IRF7.

a. IKK ϵ preferentially binds to IRF7. Each Flag-IKK ϵ isoform was co-transfected with V5-IRF3 or V5-IRF7 in HEK293T cells. Flag-IKK ϵ was immunoprecipitated with anti-Flag beads. V5-IRF3 (left panel) and V5-IRF7 (right panel) were detected by anti-V5 antibody.

2. The molecular weight of version 3 looks different in Western Blots from Figure 3 onwards compared to blots in Figure 2.

Author reply:

Thank you for your comment.

The different molecular weight of IKK ϵ v3 results from different fusion tags used. In the case of Flag tag, the molecular weight of IKK ϵ v1 is slightly higher than that of IKK ϵ v2 and IKK ϵ v3 and the molecular weights of IKK ϵ v2 and IKK ϵ v3 are similar (please refer to Figure 2f, Figure 3d and Figure 3e). In the case of V5 tag, the molecular weight of IKK ϵ v2 is much lower than that of IKK ϵ v1 and IKK ϵ v3 and the molecular weights of IKK ϵ v1 and IKK ϵ v3 are similar (please refer to Figure 3a-3c). We also provided the full picture of Figure 2f for reference (Supporting Fig 2).

Supporting Figure 2. The full Western picture of IKKε referred to Fig. 2f.

Molecular weights of Flag-IKKε isoforms

Figure 3d and 3e

Molecular weight of V5-IKK ϵ isoforms

Figure 3a-3c

3. Figure 3c: would we not expect some effect also in the absence of ectopic Ubiquitin expression (due to ubiquitination occurring with endogenous ubiquitin)?

Author reply:

Thank you for your constructive comments.

We totally agree your suggestion. As we know IRF7 phosphorylation at Ser471/472 is required for IRF7 nuclear translocation. Hence we measured the phosphorylation of IRF7 Ser471/472 in IKK ϵ isoform-expressed RD cells in absence of exogenous ubiquitin during EV71 infection. The data showed that the phosphorylation of IRF7 Ser471/472 was enhanced by IKK ϵ v2 compared to IKK ϵ v1, IKK ϵ v3 and Ctrl group in EV71 infection even if there is no ectopic ubiquitin expression.

We have described these results in the Results section and added new Figure 3c in the revised manuscript as follows.

Results: (Page 14, Line 5-10)

To characterize the role of IKK ϵ isoform switching in EV71 infection, we examined the phosphorylation and expression of IRF7 in each ectopically IKK ϵ isoform-expressing RD cells. IKK ϵ v2 strongly induced IRF7 phosphorylation (Fig. 3c), whereas the RNA and protein expression levels of IRF7 were unchanged in each IKK ϵ transfectant during EV71 infection (Supplementary Fig. 4c).

Revised Figure 3c

Fig. 3: IKK ϵ v2 increases IRF7 phosphorylation and IRF7 translocation in the presence of ubiquitin.

c. IKK ϵ v2 strongly phosphorylates IRF7 in EV71 infection. Flag-IRF7 and each V5-IKK ϵ isoform were ectopically expressed in RD cells followed by EV71 infection. The Flag-IRF7 was immunoprecipitated with anti-Flag beads and the phosphorylation was detected by IRF7-pS471/472 antibody. The co-immunoprecipitated V5-IKK ϵ isoform was analyzed by anti-V5 antibody.

4.1 The study heavily relies on overexpression experiments, some of which are unavoidable to delineate effects of the different variants. However: is it possible to detect expression of these IKK ϵ isoforms with commercially available IKK ϵ antibodies? If so, can we see a change in protein expression

levels following EV71 infection?

Author reply:

We appreciate your constructive comment.

Yes, the protein expression of endogenous IKK ϵ v1 and v2 could be detected by IKK ϵ antibody (Koop et al., 2011). In order to precisely examine the newly synthesized endogenous IKK ϵ isoform switching in EV71 infection, the Click-iT AHA assay, a non-radioactive method for the detection of nascent protein, was performed. As shown in revised Figure. 1c, the isoform switching of IKK ϵ v1 and v2 was confirmed at both RNA and protein levels.

We have added the revised descriptions in the Results and Methods sections of the revised manuscript and added new Figure 1c as follows.

Revised Results: (Page 9, Line 3-8)

To address whether the IKK ϵ isoform switching can be detected in protein level, we performed a Click-iT AHA assay to detect nascent IKK ϵ isoforms. Biotin-labeled newly synthesized proteins at indicated time points postinfection were purified, and IKK ϵ isoforms were detected by Western blotting. The expression of IKK ϵ v2 increased while the v1 decreased after EV71 infection, consistent with our RNA data (Fig. 1c).

Revised Methods: (Page 22, Line 12 – Page 23, Line 4)

Click-iT AHA assay. RD cells were cultured in glutamine, methionine, and cystine free medium supplemented with 50uM Click-iT AHA (L-Azidohomoalanine) (Invitrogen). After incubation, AHA was taken up by cells and loaded onto methionine tRNAs. During translation, AHA is incorporated into newly synthesized proteins. A biotin-based tag is then added by click chemistry according to manufacturer's instructions, and the newly synthesized protein was precipitated using streptavidin dynabeads (Invitrogen) followed by Western blotting analysis with antibody against IKK ϵ (1:1000, Cell Signaling).

Revised Figure 1c

Fig. 1: EV71 infection triggers IKKε isoform switching.

c. IKKε isoform switching is confirmed by Western blotting. The Click-iT AHA assay was performed to measure newly synthesized IKKε v1 and v2. The synthesis of IKKε v2 was increased while IKKε v1 was decreased in EV71 infection.

4.2 Also, in Figure 2f the authors used variant-specific siRNA to knock-down expression of their ectopically-expressed IKKε variants. Should the same siRNAs not also be able to knock down the endogenous variant mRNAs and therefore affect viral replication in the absence of ectopically-expressed flag-IKKε variants?

Author reply:

We appreciate your constructive comment and apologize for the unclear label of the siRNAs against IKKε.

The siRNAs used in Figure 2f are not variant-specific. siRNA-IKKε-1 and siRNA-IKKε-2 are two unique siRNAs that target non-overlapping regions of all the three IKKε isoforms. Hence, we cannot use these siRNAs to knock down the endogenous variant mRNAs. To avoid the misunderstanding caused by unclear label we changed the name of “siRNA-IKKε-1” to “siRNA1-IKKε” and the name of “siRNA-IKKε-2” to “siRNA2-IKKε” in the revised manuscript and revised Figure 2f.

It is a good idea to use isoform-specific siRNA for studying the role of different isoforms in viral replication and exploring the underlying mechanism. Actually, we have tried to design isoform-specific siRNAs, however, we failed to design isoform-specific siRNAs. The strategy used to design the specific siRNAs was

described below:

(1) siRNA design for IKKε isoform 2

The exon 20 annotated in IKKε isoform 1 (NM_014002) was lacked in IKKε isoform 2 (NM_001193322), the exon 19 and exon 21 spanning was used for isoform-specific siRNA design (please see below Figure).

Exon 19 sequence

GGTGGTGCACGAGACCAGGAACCACCTGCGCCTGGTTGGCTGTTCTGT
GGCTGCCTGTAACACAGAAGCCCAGGGGGTCCAGGAGAGTCTCAGCAA
G

Exon 21 sequence

CATGCAAGAGCTCTGCGAGGGGATGAAGCTGCTGGCATCTGACCTCCTG
GACAACAACCGCATCATCGAACG

Exon 19 and 21 spanning

GGTGGTGCACGAGACCAGGAACCACCTGCGCCTGGTTGGCTGTTCTGT
GGCTGCCTGTAACACAGAAGCCCAGGGGGTCCAGGAGAGTCTCAGCAA
GCATGCAAGAGCTCTGCGAGGGGATGAAGCTGCTGGCATCTGACCTCCT
GGACAACAACCGCATCATCGAACG

Custom Dice-Substate siRNA (DsiRNA) designer (IDT, https://sg.idtdna.com/site/order/designtool/index/DSIRNA_PREDESIGN) was used for siRNA design and all designed siRNAs fulfilling design criteria were listed as follows. However, none of designed siRNA located at exon 19-21 spanning.

Designed siRNA	Sequence position	Sequence	Exon 19-21 spanning
1	132-157	5' GCAUCUGACCUCCUGGACAACAACC 3'	No
2	143-168	5' CCUGGACAACAACCGCAUCAUCGAA 3'	No
3	133-158	5' CAUCUGACCUCCUGGACAACAACCG 3'	No
4	144-169	5' CUGGACAACAACCGCAUCAUCGAAC 3'	No
5	134-159	5' AUCUGACCUCCUGGACAACAACCGC 3'	No

6	141-166	5' CUCCUGGACAACAACCGCAUCAUCG 3'	No
7	136-161	5' CUGACCUCCUGGACAACAACCGCAT 3'	No
8	131-156	5' GGCAUCUGACCUCCUGGACAACAAC 3'	No
9	140-165	5' CCUCCUGGACAACAACCGCAUCATC 3'	No

(2) siRNA design for IKK ϵ isoform 3

The exon 3 annotated in IKK ϵ isoform 1 (NM_014002) was lacked in IKK ϵ isoform 3 (NM_001193321), the exon 2 and exon 4 spanning was used for isoform-specific siRNA design (please see below Figure)..

Exon 2 sequence

CTCAGCTCCTGGACGTGCCACAGACAGAAAGCATAACATACTCGCCA
 GGAAGAGCCTTTGCCTGACTCAGGGCAGCTCAGAGTGTGGG

Exon 4 sequence

AAATCCGGAGAGCTGGTTGCTGTGAAGGTCTTCAACACTACCAGCTACCT
 GCGGCCCGCGAGGTGCAAGTGAGGGAGTTTGAGGTCCTGCGGAAGCT
 GAACCACCAGAACATTGTCAAGCTCTTTGCGGTGGAGGAGACG

Exon 2 and 4 spanning

CTCAGCTCCTGGACGTGCCACAGACAGAAAGCATAACATACTCGCCA
 GGAAGAGCCTTTGCCTGACTCAGGGCAGCTCAGAGTGTGGGAAATCCG
 GAGAGCTGGTTGCTGTGAAGGTCTTCAACACTACCAGCTACCTGCGGCC
 CCGCGAGGTGCAAGTGAGGGAGTTTGAGGTCCTGCGGAAGCTGAACCA
 CCAGAACATTGTCAAGCTCTTTGCGGTGGAGGAGACG

Custom Dice-Substate siRNA (DsiRNA) designer (IDT, https://sg.idtdna.com/site/order/designtool/index/DSIRNA_PREDESIGN) was used for siRNA design and all designed siRNAs fulfilling design criteria were listed as follows. However, none of designed siRNA located at exon 2-4 spanning.

Designed siRNA	Sequence position	Sequence	Exon 2-4 spanning
1	189-214	5' GAACCACCAGAACAUUGUCAAGCTC 3'	No

2	185-210	5' AGCUGAACCACCAGAACAUAUGUCA 3'	No
3	190-215	5' AACCAACCAGAACAUAUGUCAAGCUCT 3'	No
4	114-139	5' GAAGGUCUUCAACACUACCAGCUAC 3'	No
5	113-138	5' UGAAGGUCUUCAACACUACCAGCTA 3'	No
6	184-209	5' AAGCUGAACCACCAGAACAUAUGUCA 3'	No
7	106-131	5' GUUGCUGUGAAGGUCUUCAACACTA 3'	No
8	186-211	5' GCUGAACCACCAGAACAUAUGUCAAG 3'	No
9	117-142	5' GGUCUUCAACACUACCAGCUACCTG 3'	No
10	112-137	5' GUGAAGGUCUUCAACACUACCAGCT 3'	No
11	115-140	5' AAGGUCUUCAACACUACCAGCUACC 3'	No
12	192-217	5' CCACCAGAACAUAUGUCAAGCUCUTT 3'	No
13	118-143	5' GUCUUCAACACUACCAGCUACCUGC 3'	No
14	195-220	5' CCAGAACAUAUGUCAAGCUCUUUGCG 3'	No
15	187-212	5' CUGAACCACCAGAACAUAUGUCAAGC 3'	No
16	191-216	5' ACCACCAGAACAUAUGUCAAGCUCTT 3'	No
17	116-141	5' AGGUCUUCAACACUACCAGCUACCT 3'	No
18	193-218	5' CACCAGAACAUAUGUCAAGCUCUUTG 3'	No
19	183-208	5' GAAGCUGAACCACCAGAACAUAUGTC 3'	No

We have revised the sections of Results and Methods as well as Figure 2f in revised manuscript as follows.

Revised Results: (Page 12, Line 8-11)

In contrast, the attenuation of virus titers observed in IKK ϵ v2 transfectants was greatly eliminated by two IKK ϵ siRNAs, which target two common regions of IKK ϵ isoforms, respectively. (Fig. 2f).

Revised Methods: (Page 22, Line 8-11)

siRNA transfection. RD cells were transiently transfected with siRNA1-IKK ϵ or siRNA2-IKK ϵ (s18537, s18538, ThermoFisher) at a final concentration of 50 nM using RNAiMAX (Invitrogen). Further treatments or assays were generally performed 48 h after siRNA transfection.

Revised Figure 2f

Fig. 2: IKKε v2 increases IRF7-mediated IFNβ and ISGs expressions in EV71 infection and attenuates virus propagation.

f. Attenuation of virus titer in IKKε v2 transfectants is restored by IKKε siRNAs. The siRNAs against IKKε, siRNA1-IKKε or siRNA2-IKKε, were introduced into RD cells expressing each Flag-IKKε isoform followed by EV71 infection. The viral titers and ectopic IKKε isoform expressions were determined by plaque assay and Western blotting, respectively. β-actin was served as an internal control. All data presented are mean ± SD (n=3). * and ** represent p value <0.05 as compared with siRNA ctrl group.

5. Is there a qualitative or just a quantitative difference between IKKε v1 and v2? Is one more active in IFN induction and the other more active in IFN signalling? V2 seems have even stronger effect on ISG expression compared to IFN induction?

Authors reply:

Thank you for your comment.

We think there might be qualitative differences between IKKε v1 and IKKε v2. In this study, we found EV71 infection induces IKKε isoform switching from IKKε v1 to IKKε v2. According to our real-time PCR analysis, we found that IKKε v2 is superior to IKKε v1 on induction of IFN-β expression during EV71 infection (Fig. 2b). Again, IKKε v2 seemed to have stronger activity than IKKε v1 on ISG56, ISG20, OAS1 and MxA induction (Fig. 2d and Supplementary Fig. 2c). These data implied that IKKε v2 seems to be more active in both IFN induction and IFN signaling.

6. Finally, some language editing might be useful and an expansion of the

discussion.

Authors reply:

We apologize for the unprofessional writing. The revised manuscript has been edited by a native English speaking editor of American Journal Experts (AJE).

AJE. **Editing Certificate**

This document certifies that the manuscript

IKKε isoform switching governs immune response against EV71 infection

prepared by the authors

Ya-Ling Chang, Min-Hsuan Chen, Yao-Ting Huang, Sung-Liang Yu

was edited for proper English language, grammar, punctuation, spelling, and overall style by one or more of the highly qualified native English speaking editors at AJE.

This certificate was issued on **September 16, 2020** and may be verified on the AJE website using the verification code **092B-54C0-EA46-1D8F-3795**.

 Neither the research content nor the authors' intentions were altered in any way during the editing process. Documents receiving this certification should be English-ready for publication; however, the author has the ability to accept or reject our suggestions and changes. To verify the final AJE edited version, please visit our verification page at aje.com/certificate. If you have any questions or concerns about this edited document, please contact AJE at support@aje.com.

AJE provides a range of editing, translation, and manuscript services for researchers and publishers around the world. For more information about our company, services, and partner discounts, please visit aje.com.

We have expanded the Discussion section in revised descriptions as follows.

Revised Discussion:

(Page 18, Line 5-17)

Type I IFNs provide a first line of defense against viral infections. Administration of IFNs can limit virus spreading at an early phase during virus infections, while many viruses prevent IFN attacks by the inhibition of IFN through several mechanisms. In our study, we demonstrated that IKKε isoform switching occurred in EV71 infection. IKKε v2 phosphorylated and thereby activated IRF7 to trigger IFN activation. We also examined IKKε isoform switching in another enterovirus, CVB3, and in a DNA virus, HSV-1. The relative expression of IKKε v2 was upregulated while IKKε v1 was downregulated, upon CVB3 and HSV-1 infection. The relative abundance of the different IKKε isoforms might represent a novel regulatory mechanism controlling the innate immune response. Further studies are required to

comprehensively understand how virus infection induces IKK ϵ isoform switching.

(Page 19, Line 9 – Page 20, Line 2)

RIG-1/MDA5, the upstream activator of IKK ϵ , was reported to be cleaved by EV71-encoded 2A and 3C protease^(3C_{pro}) and led to inhibition of the IFN- α / β response^{58,59}. However, upregulation of RIG-I ubiquitination promoted the expression of IFN- β and ISGs⁶⁰. Another study indicated that ARRDC4 promoted K63 polyubiquitination of MDA5, consequently activating the innate immune response in EV71 infection⁶¹. In the present study, we found that IKK ϵ v2 showed higher activity in the presence of K63-linked ubiquitination, and IKK ϵ v2 promoted IRF7 activation under enhanced ubiquitination in EV71 infection. Whether RIG-1/MDA5 is involved in the regulation of IKK ϵ isoform switching is remaining for further investigation.

Revised References: (Page 38, Line 27 – Page 39, Line 3)

58 Lei, X. et al. The 3C protein of enterovirus 71 inhibits retinoid acid-inducible gene I-mediated interferon regulatory factor 3 activation and type I interferon responses. *Journal of virology* 84, 8051-8061, doi:10.1128/JVI.02491-09 (2010).

59 Feng, Q. et al. Enterovirus 2Apro targets MDA5 and MAVS in infected cells. *Journal of virology* 88, 3369-3378, doi:10.1128/JVI.02712-13 (2014).

60 Chen, N. et al. Enterovirus 71 inhibits cellular type I interferon signaling by inhibiting host RIG-I ubiquitination. *Microb Pathog* 100, 84-89, doi:10.1016/j.micpath.2016.09.001 (2016).

61 Meng, J. et al. ARRDC4 regulates enterovirus 71-induced innate immune response by promoting K63 polyubiquitination of MDA5 through TRIM65. *Cell Death Dis* 8, e2866, doi:10.1038/cddis.2017.257 (2017).

7. Do the authors know what induces isoform switching? Is it an IFN-induced response? Is it RIG-1/mda-5-dependent?

Author reply:

Thank you for your constructive comment.

Although we did not study what induces IKK ϵ isoform switching in this manuscript, we have extended the possible regulatory mechanisms in the revised manuscript.

We have added the description in the Discussion section of the revised manuscript as follows.

Revised Discussion: (Page 19, Line 9 – Page 20, Line 2)

RIG-1/MDA5, the upstream activator of IKK ϵ , was reported to be cleaved by EV71-encoded 2A and 3C protease (^{3C_{pro}}) and led to inhibition of the IFN- α/β response^{58,59}. However, upregulation of RIG-I ubiquitination promoted the expression of IFN- β and ISGs⁶⁰. Another study indicated that ARRDC4 promoted K63 polyubiquitination of MDA5, consequently activating the innate immune response in EV71 infection⁶¹. In the present study, we found that IKK ϵ v2 showed higher activity in the presence of K63-linked ubiquitination, and IKK ϵ v2 promoted IRF7 activation under enhanced ubiquitination in EV71 infection. Whether RIG-1/MDA5 is involved in the regulation of IKK ϵ isoform switching is remaining for further investigation.

Revised References: (Page 38, Line 27 – Page 39, Line 3)

58 Lei, X. et al. The 3C protein of enterovirus 71 inhibits retinoid acid-inducible gene I-mediated interferon regulatory factor 3 activation and type I interferon responses. *Journal of virology* 84, 8051-8061, doi:10.1128/JVI.02491-09 (2010).

59 Feng, Q. et al. Enterovirus 2A_{pro} targets MDA5 and MAVS in infected cells. *Journal of virology* 88, 3369-3378, doi:10.1128/JVI.02712-13 (2014).

60 Chen, N. et al. Enterovirus 71 inhibits cellular type I interferon signaling by inhibiting host RIG-I ubiquitination. *Microb Pathog* 100, 84-89, doi:10.1016/j.micpath.2016.09.001 (2016).

61 Meng, J. et al. ARRDC4 regulates enterovirus 71-induced innate immune response by promoting K63 polyubiquitination of MDA5 through TRIM65. *Cell Death Dis* 8, e2866, doi:10.1038/cddis.2017.257 (2017).

References:

Xu, L.J., Jiang, T., Zhang, F.J., Han, J.F., Liu, J., Zhao, H., Li, X.F., Liu, R.J., Deng, Y.Q., Wu, X.Y., et al. (2013). Global transcriptomic analysis of human neuroblastoma cells in response to enterovirus type 71 infection. *PLoS One* 8, e65948.

Reviewers' Comments:

Reviewer #1:

Remarks to the Author:

All my comments are carefully addressed. I have no future comments. It is an interesting finding.

Reviewer #2:

None

Reviewer #3:

Remarks to the Author:

I think the manuscript has been improved by the revisions.

Inclusion of Figure 1c strengthened the evidence for infection-induced isoform switching.

I have some remaining reservations:

1. The authors have not commented on the discrepancy of their findings versus those published by Koop et al 2011. It looks like V2 in this manuscript is the same as their sv2? How do the authors explain that this splice variant was found to be inhibitory in the Koop et al. study, yet shows enhanced signalling in the present study? At the very least this discrepancy should be addressed in the discussion.

2. The explanation about the different molecular weights of flag-tagged V1/2/3 versus V5-tagged V1/2/3 is not convincing. While this might explain a difference in absolute MW (although 8aa for a flag-tag versus 14aa for a V5 tag should not make a big difference here either), it cannot explain why the relationship of the different isoforms to each other changes. With the V5-tag isoform 3 runs close to isoform 1, but with the flag-tag it runs similar to isoform 2. Isoform 2 seems much smaller than isoform 1 with the V5-tag, but only slightly smaller with the flag-tag. I find this quite concerning.

3. The explanation provided for the absent/very small effects of V1 on the read-outs used is also not very convincing. Many, many publications have shown before that overexpression of IKKe (v1 I am assuming) strongly drives also IRF3-related read-outs, so it is still not clear to me why we see hardly any effects of v1 on IFN β induction and viral titers in figure 1.

4: Similarly it has been shown before that IKKe can also interact with IRF3 (even at endogenous levels), so it is unclear why the authors do not see this in their IP with overexpressed proteins. In Supp Fig 2a: have V5-IRF3 and V5-IRF7, which are shown in two different panels been detected on the same membrane/film and can thus be compared in intensity, or not?

5. Despite two reviewers asking for this, the authors have not addressed the question about the mechanism for isotype switching. Not experimentally, and not much on this has been added to the discussion either. If different viruses induce this, it would seem quite obvious to test whether this is an IFN-mediated effect?

Minor issues:

In Supp Figure 2: there are two actin blots shown in the IKKe IP. Duplicated by mistake?

What is different about Figure 3a and 3b?

Supplementary Figure 5: this pulldown experiment does not seem to be very well controlled.

Figure 1c: is the second V1 panel a lower exposure? If so, please label this clearly.

I think the manuscript has been improved by the revisions.

Inclusion of Figure 1c strengthened the evidence for infection-induced isoform switching.

I have some remaining reservations:

1. The authors have not commented on the discrepancy of their findings versus those published by Koop et al 2011. It looks like V2 in this manuscript is the same as their sv2? How do the authors explain that this splice variant was found to be inhibitory in the Koop et al. study, yet shows enhanced signalling in the present study? At the very least this discrepancy should be addressed in the discussion.

Author reply:

Thank you for your constructive comment.

Based on the gene structure, our IKK ϵ -v2 lacking exon 20 is the same as IKK ϵ -sv2 in the study published by Koop et al 2011 (Supporting Figure 1). Both IKK ϵ -v2 and IKK ϵ -sv2 are natural isoforms, and their expressions were also verified in cells by Koop's and our groups, respectively. To investigate the physiological function, we constructed the tagged IKK ϵ -v2 (Supporting Figure 1A), while Koop's group generated the expression construct of IKK ϵ -D647, which is an artificial C-terminal truncated form lacking exon 20, 21, and 22, to characterize the C-terminal function of IKK ϵ (Supporting Figure 1B). IKK ϵ -D647 was demonstrated to have an inhibitory effect on functions of IKK ϵ , as a result, Koop and his colleagues indirectly indicated that IKK ϵ -sv2 is an antagonistic isoform. However, we have demonstrated that the natural form IKK ϵ -v2 showed an enhanced signaling in EV71 infection. The possible reasons for discrepancy of our findings versus those published by Koop's group were described as follows:

1. The different constructs.

In our study, we generated three natural isoforms constructs identified by NGS during EV71 infection, which is IKK ϵ v1 contains a full-length coding DNA sequence, IKK ϵ v2 lacks exon 20 and IKK ϵ v3 lacks exon 3. In contrast to the natural isoforms, Koop's group created IKK ϵ constructs with C-terminal truncation, and they used the IKK ϵ -D647 instead of IKK ϵ -sv2 to perform all the experiments in their report. Compared with the gene structure of IKK ϵ v2, not only exon 20, but also truncated exon 21 and 22 were deleted in IKK ϵ -D647 construct. The structural difference might contribute to the discrepant function between IKK ϵ v2 and IKK ϵ sv2.

2. The different experimental designs.

In our study, we explore the physiological role of IKK ϵ isoforms in EV71 infection. Hence, we evaluated the functions of constructs in EV71 host cells (RD cells). In our report, we demonstrated that IKK ϵ v2 presents higher activities in IRF7 signaling than the other two IKK ϵ isoforms and possesses higher antiviral activity in EV71 infected cells. On the other hand, Koop's group characterized the function of exon 20 or 21 in IKK ϵ isoforms. Therefore, Koop's group transfected the C-terminal truncated constructs, the IKK ϵ -D647 (lacking exon 20-22) and IKK ϵ -D684 (lacking exon 21-22) instead of IKK ϵ -sv2 and IKK ϵ -sv1 respectively, into HEK293T and 293/TLR3 cells to evaluate the functions of exon 20, 21 and 22. Based on their *in vitro* data, they concluded that the splice variants of IKK ϵ have the potential to inhibit the activity of the full-length protein, however, they did not directly examine physiological function of IKK ϵ -sv2.

A.

B.

Supporting Figure 1. Structure of IKK ϵ splice variants and expression constructs.

A. Illustration of IKK ϵ isoforms in our study

B. Illustration of IKK ϵ isoforms of published reference by Koop et al 2011

We have discussed the discrepancy between our study and Koop's paper in the revised discussion section as follows.

Revised Discussion: (Page 22, Line 11 - Page 22, Line 16)

It has been proposed that the isoform switching is one of regulatory mechanisms to fine-tune the functions of IKK ϵ ⁷⁶. Koop's study found two artificial IKK ϵ variants lacking exon 20-22 and exon 21-22 exhibited inhibitory effect on IRF3 signaling. The discrepancy between Koop's study and our findings might be partly resulted from different IKK ϵ constructs and different assay conditions, i.e. different cells used and EV71 challenge or not.

Revised References:

76 Koop, A. *et al.* Novel splice variants of human IKKepsilon negatively regulate IKKepsilon-induced IRF3 and NF-kB activation. *Eur J Immunol* **41**, 224-234, doi:10.1002/eji.201040814 (2011).

2. The explanation about the different molecular weights of flag-tagged V1/2/3 versus V5-tagged V1/2/3 is not convincing. While this might explain a difference in absolute MW (although 8aa for a flag-tag versus 14aa for a V5 tag should not make a big difference here either), it cannot explain why the relationship of the different isoforms to each other changes. With the V5-tag isoform 3 runs close to isoform 1, but with the flag-tag it runs similar to isoform 2. Isoform 2 seems much smaller than isoform 1 with the V5-tag, but only slightly smaller with the flag-tag. I find this quite concerning.

Author reply:

Thank you for your comment. We totally agree your opinion.

To understand why the molecular weights (MW) of Flag-tagged IKK ϵ v1/2/3 versus V5-tagged IKK ϵ v1/2/3 are different and whether these differences influence our findings, we discussed in three aspects, 1. unexpected mutations in V5-tagged IKK ϵ v2, 2. The degradation of V5-tagged IKK ϵ v2, 3. different post translational modifications (PTMs) of V5-tagged IKK ϵ v2.

First, to check whether any unexpected mutations generated in plasmid amplification, although we have verified all of constructs before, we re-sequenced V5-tagged IKK ϵ v2 and Flag-tagged IKK ϵ v2 constructs used in transfection experiments by Sanger sequencing and performed Blast comparison. There are no unexpected mutations found in both constructs (Supporting Figure 2).

Secondly, to directly compare the MWs of V5-tagged IKK ϵ v2 and Flag-tagged IKK ϵ v2, we detected Flag-tagged V1/2/3 and V5-tagged V1/2/3 by Western blotting on the same Immuno-Blot membrane. As shown in Supporting Figure 3, there are two sharp bands in V5-tagged IKK ϵ v2, which is the same as Fig. 4b. The MW of the upper one is similar to Flag-tagged IKK ϵ v2, however, the major band of V5-tagged IKK ϵ v2 is smaller than Flag-tagged IKK ϵ v2. Additionally, V5-tagged v1/3 show similar MWs with Flag-tagged IKK ϵ v1/3. Moreover, we explored whether the major band of V5-tagged IKK ϵ v2 is a degraded form. As shown in Supporting Figure 3, the MW pattern of the V5-tagged IKK ϵ v2 is the same in the absence (middle panel) or presence (right panel) of MG132. These data indicated that the major form of V5-tagged IKK ϵ v2 is not a product of protein degradation.

Thirdly, the V5-tagged vector we used is pcDNATM3.1/V5-His A, B, and C (Invitrogen, V810-20). It is well-known that poly-His tags always nonspecifically bind to endogenous proteins with histidine clusters (Mahmood and Xie, 2015). Therefore, the non-specific hydrophobic interactions may result in unusual PTMs on V5-His tagged IKK ϵ by enzymes associated with basic V5-His, but not acidic Flag. The non-specific interaction may interfere sumoylation, ubiquitination, multiple phosphorylations or a combination of different PTMs. Thus, the MW difference among these recombinant isoforms might be attributed to different PTMs. Moreover, it is possible that V5-His tagged IKK ϵ may have different PTMs for the three isoforms, and here, we only focused on the significant MW difference of V5-His tagged IKK ϵ v2. According to previous reports, IKK ϵ and TBK1 share 61% overall homology to each other (Hiscott et al., 2006) (Supporting Figure 4), and the C-terminal coiled-coil domain of TBK1 is demonstrated to be responsible for its sumoylation (Saul et al., 2015). As indicated in Supporting Figure 5, IKK ϵ v2 is a C-terminus truncated protein with several Lysine residues lost compared to IKK ϵ v1 and IKK ϵ v3. Hence, it is possible that some small ubiquitin-like modifiers (i.e. SUMO enzyme) may non-specifically interact with V5-His tagged IKK ϵ isoforms to result in IKK ϵ v1 and IKK ϵ v3 MW shift while these enzymes might not modify IKK ϵ v2.

In summary, we excluded the possibilities that the related “smaller” MW of V5-tagged IKK ϵ v2 is due to unexpected mutations or protein degradation. The different molecular weights between Flag-tagged IKK ϵ v2 and V5-tagged IKK ϵ v2 might result from nonspecific protein binding by His tag. In addition, to minimize the artificial effect induced by tag types and tag positions, we used two different tags, V5 and Flag, which are respectively located at C- and

N-terminal of recombinant proteins to characterize the functions of IKKε isoforms. Our data indicated that both Flag-tagged IKKε v2 and V5-tagged IKKε v2 exhibit stronger activities on IRF7 signaling.

Query: V5 tagged IKKε
Subject: Flag tagged IKKε

Score	Expect	Identities	Gaps	Strand	Frame
3840 bits(1971)	0.00	197/197(100%)	0/197(0%)	Plus/Plus	
Query 1	ATCGAGACACAGCCAAATTCCTCTGGACACAGATCACTCTCGGGCAGGGGCGCCACT	60			
Sbjct 1	ATCGAGACACAGCCAAATTCCTCTGGACACAGATCACTCTCGGGCAGGGGCGCCACT	60			
Query 61	GGCAGTGTGTACAGGCGCCCAACAGAAATCCGGAGAGCTGGTCTCTGAAGTCTTC	120			
Sbjct 61	GGCAGTGTGTACAGGCGCCCAACAGAAATCCGGAGAGCTGGTCTCTGAAGTCTTC	120			
Query 121	AACACTACAGACTACCTGGGGCCCGGGAGGTCCAGGTGAGGGAGTTGAGTCTGGGG	180			
Sbjct 121	AACACTACAGACTACCTGGGGCCCGGGAGGTCCAGGTGAGGGAGTTGAGTCTGGGG	180			
Query 181	AACTGCAACACACAGACATCTCAAGCTCTTTCCTCCAGACAGCCCGGGAGCCGG	240			
Sbjct 181	AACTGCAACACACAGACATCTCAAGCTCTTTCCTCCAGACAGCCCGGGAGCCGG	240			
Query 241	CAGAGGTACTGTGATGGAGTACTCTCCAGTGGAGCTCTGATGCTGTCTGGAGAG	300			
Sbjct 241	CAGAGGTACTGTGATGGAGTACTCTCCAGTGGAGCTCTGATGCTGTCTGGAGAG	300			
Query 301	CTGAGAAAGTACTTGGGTGACTGAGAGAGTCTGCTGCTGCTGCTGCTGCTGCTG	360			
Sbjct 301	CTGAGAAAGTACTTGGGTGACTGAGAGAGTCTGCTGCTGCTGCTGCTGCTGCTG	360			
Query 361	GGCAGCAATGACACTCCGGGAGAGCCGATCTGCTGCTGCTGCTGCTGCTGCTG	420			
Sbjct 361	GGCAGCAATGACACTCCGGGAGAGCCGATCTGCTGCTGCTGCTGCTGCTGCTG	420			
Query 421	ATCATGCCCTGTAGGGAGGAGGGCCAGAGATCTACAGCTGACAGACTTCCGGGCT	480			
Sbjct 421	ATCATGCCCTGTAGGGAGGAGGGCCAGAGATCTACAGCTGACAGACTTCCGGGCT	480			
Query 481	GGCCGGAGCTGGATGATGATGAGAAAGTCTGCTGCTGCTGCTGCTGCTGCTG	540			
Sbjct 481	GGCCGGAGCTGGATGATGATGAGAAAGTCTGCTGCTGCTGCTGCTGCTGCTG	540			
Query 541	CATCCCGACATCTAGCGGGGGGGTCTCCGAAAGCCGAGCAAAAGGCTTGGGGGT	600			
Sbjct 541	CATCCCGACATCTAGCGGGGGGGTCTCCGAAAGCCGAGCAAAAGGCTTGGGGGT	600			
Query 601	ACTCTGCACTCTGAGCACTGAGTGCACCTCTGACCATGAGCCACTGGCAGCTGGCC	660			
Sbjct 601	ACTCTGCACTCTGAGCACTGAGTGCACCTCTGACCATGAGCCACTGGCAGCTGGCC	660			
Query 661	TTCTATCCCTTTGGTGGGGCCGGGGAGAGAGGAGTCACTGATCCAGCCAGGGAG	720			
Sbjct 661	TTCTATCCCTTTGGTGGGGCCGGGGAGAGAGGAGTCACTGATCCAGCCAGGGAG	720			
Query 721	AAGCCGGCTGGGGCAATTCAGGTGCCAGAGGGGGGGAGAGCCGGGCTCTGGAGTGG	780			
Sbjct 721	AAGCCGGCTGGGGCAATTCAGGTGCCAGAGGGGGGGAGAGCCGGGCTCTGGAGTGG	780			
Query 781	TACACCTCCCACTCACTCCGAGCTCTCACTGGGGCTCCAGAGCCAGCTGGTCCCACT	840			
Sbjct 781	TACACCTCCCACTCACTCCGAGCTCTCACTGGGGCTCCAGAGCCAGCTGGTCCCACT	840			
Query 841	CTGGCCAACTCTGGAGTGGAGAGCCAAAGTCTGGGGCTTGGCCACTCTCTTGG	900			
Sbjct 841	CTGGCCAACTCTGGAGTGGAGAGCCAAAGTCTGGGGCTTGGCCACTCTCTTGG	900			
Query 901	GAGACAGTACAGTCTGGAGAGTCTCTGATGATCTCTCTGCTCTCTCTGAGGAGTC	960			
Sbjct 901	GAGACAGTACAGTCTGGAGAGTCTCTGATGATCTCTCTGCTCTCTCTGAGGAGTC	960			

Query 961	CTGCACCACATCTATATCCATGCCCCAACAGCATAGCCATTTTCAGGAGGCCCTGCAC	1020
Sbjct 961	CTGCACCACATCTATATCCATGCCCCAACAGCATAGCCATTTTCAGGAGGCCCTGCAC	1020
Query 1021	AAGCAGACCCAGTGGGGCCCCCGACACAGGAGTACCTCTTTGAGGGTCACTCTGTGTC	1080
Sbjct 1021	AAGCAGACCCAGTGGGGCCCCCGACACAGGAGTACCTCTTTGAGGGTCACTCTGTGTC	1080
Query 1081	CTCGAGCTGAGGCTCTGAGCAGAGTACATCTCCACAGACGGCCAGGAGCCCTGACCC	1140
Sbjct 1081	CTCGAGCTGAGGCTCTGAGCAGAGTACATCTCCACAGACGGCCAGGAGCCCTGACCC	1140
Query 1141	CTCTTCAGCAGCCATCCCTAAGGGGCTGGCTTCAGGAGCCCTGCTTGGAGTCTCC	1200
Sbjct 1141	CTCTTCAGCAGCCATCCCTAAGGGGCTGGCTTCAGGAGCCCTGCTTGGAGTCTCC	1200
Query 1201	AAGTTCGTCCCAAGTGGAGCTGAGGGGATACACACTGGCAGGGGGCTGTGGGG	1260
Sbjct 1201	AAGTTCGTCCCAAGTGGAGCTGAGGGGATACACACTGGCAGGGGGCTGTGGGG	1260
Query 1261	GGCCGTACAGGGCTGGGGTGGAGGGGGCTCTGGATGGAGAGAGCTAACTTT	1320
Sbjct 1261	GGCCGTACAGGGCTGGGGTGGAGGGGGCTCTGGATGGAGAGAGCTAACTTT	1320
Query 1321	GGGGGCTGCACTGGGTCTGAGAGCTCTCCAGCCACATCCAGAGCACTGGAGTGC	1380
Sbjct 1321	GGGGGCTGCACTGGGTCTGAGAGCTCTCCAGCCACATCCAGAGCACTGGAGTGC	1380
Query 1381	GCAGGACATCCCTCTCTTACCTCAGCAGGAGCTGGGAACTGAGAGGTTGAGCAGG	1440
Sbjct 1381	GCAGGACATCCCTCTCTTACCTCAGCAGGAGCTGGGAACTGAGAGGTTGAGCAGG	1440
Query 1441	GCTGGAAGCCCTGAGATCCAGGAATCGAAGGGGCTCGAGAACTGAGGCTCAGGCTGG	1500
Sbjct 1441	GCTGGAAGCCCTGAGATCCAGGAATCGAAGGGGCTCGAGAACTGAGGCTCAGGCTGG	1500
Query 1501	ACTCTAGCGAGGCTCTCTCTGATCTCCCAAAATATCCAGGAGCCCGAGGAGGCTG	1560
Sbjct 1501	ACTCTAGCGAGGCTCTCTCTGATCTCCCAAAATATCCAGGAGCCCGAGGAGGCTG	1560
Query 1561	AGCAGCCGAAACGGGAGCTGGTGAAGAGCCGGATCAGGATCAGAGGAGGAGC	1620
Sbjct 1561	AGCAGCCGAAACGGGAGCTGGTGAAGAGCCGGATCAGGATCAGAGGAGGAGC	1620
Query 1621	CAGCAGATTCAGTCTCTTGGCAAGATCACTCTCACTACAACAGCTTCAAGAGCT	1680
Sbjct 1621	CAGCAGATTCAGTCTCTTGGCAAGATCACTCTCACTACAACAGCTTCAAGAGCT	1680
Query 1681	AGATGAGGGCCAGGGCTGGTCAACAGGAGGAGTCAACAGGCTGGATGAGTGGAT	1740
Sbjct 1681	AGATGAGGGCCAGGGCTGGTCAACAGGAGGAGTCAACAGGCTGGATGAGTGGAT	1740
Query 1741	TTGAGTCAATTTGCAAAAAGCTCTCCAGGTTGTTCAGGAGGAGTGGCTCCAGAGTAT	1800
Sbjct 1741	TTGAGTCAATTTGCAAAAAGCTCTCCAGGTTGTTCAGGAGGAGTGGCTCCAGAGTAT	1800
Query 1801	CAGGCTCTTGTACACACGGGAGGAGTGAAGGTTGTTCAGGAGGAGTGGCTCCAGAG	1860
Sbjct 1801	CAGGCTCTTGTACACACGGGAGGAGTGAAGGTTGTTCAGGAGGAGTGGCTCCAGAG	1860
Query 1861	CTGGCCAACTCTGGAGTGGAGAGCCAAAGTCTGGGGCTTGGCCACTCTCTTGG	1920
Sbjct 1861	CTGGCCAACTCTGGAGTGGAGAGCCAAAGTCTGGGGCTTGGCCACTCTCTTGG	1920
Query 1921	AGCTCTGAGCAGATCCAGAGCTCTCCAGGGGATGAGCTCTCTGGGATC	1971
Sbjct 1921	AGCTCTGAGCAGATCCAGAGCTCTCCAGGGGATGAGCTCTCTGGGATC	1971

Supporting Figure 2. The sequence comparison of V5-tagged IKKε v2 and Flag-tagged IKKε v2 constructs

Supporting Figure 3. Molecular weight characterization of Flag-tagged and V5-tagged IKKε isoforms

Supporting Figure 4. Comparison of the IKK family

Supporting Figure 5. The sequence alignment of V5-tagged IKKε isoforms

3. The explanation provided for the absent/very small effects of V1 on the read-outs used is also not very convincing. Many, many publications have shown before that overexpression of IKKε (v1 I am assuming) strongly drives also IRF3-related read-outs, so it is still not clear to me why we see hardly any effects of v1 on IFNβ induction and viral titers in figure 1.

Author reply:

Thank you for your comment. We apologize we only provided the data to exclude the possibility that the different responses of IKK ϵ isoforms were caused by unequal expressions of IKK ϵ isoforms and did not provide a convincing explanation for very small effects of IKK ϵ v1 on IRF3/7 signaling, IFN production and antiviral activity in last response letter. Here, we reviewed IRF3/IFN related studies thoroughly and provided a reasonable explanation to support our findings.

IFN induction is regulated in a virus- and cell type-specific manner

The innate immune responses provide an early phase defense against viral infections. Viral pathogens can be detected through pattern-recognition receptors (PRRs), including Toll-like receptors (TLRs), NOD-like receptors (NLRs), retinoid acid-inducible gene I (RIG-I), and melanoma differentiation-associated gene 5 (MDA-5) (Broz and Monack, 2013; Holm et al., 2013; Takeuchi and Akira, 2010). After recognition of viral genome by these receptors, TIR domain-containing adapter-inducing interferon- β (TRIF) and mitochondrial antiviral signaling protein (MAVS) form a complex with TNF receptor-associated factor 3 (TRAF3) and lead to activation of TANK-binding kinase 1 (TBK1) and I κ B kinase ϵ (IKK ϵ) (Takeda and Akira, 2005). Subsequent activation of interferon regulatory factors (IRFs) by TBK1/IKK ϵ results in expression of type I IFNs and proinflammatory cytokines (Tailor et al., 2006). However, it is not the case for all of viruses because certain virus infections can subvert cellular IFN induction pathways (Garcia-Sastre, 2017). For example, influenza viral proteins PB1-F2 and PB2-S1 interact with MAVS to inhibit IFN induction (Varga et al., 2012; Yamayoshi et al., 2016); another virus such as hepatitis C virus can cleave MAVS to interfere IFN production (Li et al., 2005). In addition, accumulating evidence indicates innate immune responses are regulated in a cell type-specific manner. Previous studies have demonstrated that dengue virus can induce IFN β and ISG production in the brain; in contrast, IFNs are not detectable in infected dendritic cells (Al-Shujairi et al., 2017; Rodriguez-Madoz et al., 2010), which suggests that IFN induction is disparately regulated in different cells.

EV71 has been demonstrated to regulate IFN β induction by affecting the pathways mediated by RIG-1/MDA5 and TLR upon EV71 infection. EV71 3C protease (3C^{pro}) was demonstrated to suppress IFN signaling by interrupting the RIG-1-IFN promoter-stimulating factor 1 (IPS1) interaction, and with nucleus translocation of IRF3 (Lei et al., 2010). Other studies have reported that 3C^{pro} degrades RIG-1 and cleaves adaptor TRIF to overcome IFN

production in EV71 infected cells (Barral et al., 2009; Wang et al., 2016), and EV71 2A protease ($2A^{pro}$) targets MAVS and cleaves MDA5 which is responsible for IRF3 activation to inhibit IFN production (Feng et al., 2014; Kuo et al., 2013). It is worth noting that these different regulations are demonstrated in different cell lines under different experimental conditions.

Currently, several reports indicated that cellular miRNAs also play roles in the EV71-induced innate immune response. miRNA-146a was reported to be induced upon EV71, poliovirus 3, and CVB3 infections (Ho et al., 2014). IRAK1 and TRAF6 proteins were demonstrated to be reduced by miRNA-146a and resulted in suppression of IFN β production in EV71 infected cells. miR-526a targeting cylindromatosis (CYLD) was demonstrated to stimulate phosphorylations of IRF3, I κ B, and IKK ϵ , and downregulation of miR-526a in EV71 infection impaired IFN-I production (Xu et al., 2014). miR-548 known to regulate the host antiviral responses by directly targeting IFN- λ 1 expression was shown to be suppressed upon EV71 infection (Li et al., 2013). miR-302 cluster suppressed the EV71-induced innate immune response via direct targeting to karyopherin α 2 (KPNA2) (Peng et al., 2018). More recently, Duan et al. showed the level of miR-628-5p was increased after EV71 infection. miR-628-5p suppressed TRAF3, which mediates IRF3 and NF- κ B activation, to further affect IFN β production during EV71 infection (Li et al., 2020). Taken together, IFN β expression in EV71 infection is complicated and tightly regulated.

IRF3 signaling is attenuated in EV71 infection

Based on our experimental design, we performed physiological experiments in RD cells, which were demonstrated to have high sensitivity to enterovirus and most widely be used in EV71 studies (Chen et al., 2012; Perez-Ruiz et al., 2003; Zhou et al., 2019). The host IFN pathway response to EV71 was investigated and the genes associated with the IFN pathway were characterized (Zhang et al., 2014). As showed in Supporting Figure 6, the expression of IRF3 was significant reduced (-3.91 fold in EV71 infected vs. mock RD cells) while IRF7 was unchanged in EV71 infected RD cells. Furthermore, a previous study has reported IRF3-CL, an isoform of IRF3, is ubiquitously expressed in all cell lines and acts as a negative regulator of IRF3 via dimerization with IRF3 during virus infection or in the presence of IKK ϵ overexpression (Li et al., 2011). It is possible that IRF3-CL plays an inhibitory role in ectopically IKK ϵ isoform-expressing RD cells upon EV71 infection, however, we did not measure the expression of IRF3-CL in our study.

Hence, we speculate the reasons for the limited effects of IKK ϵ v1 on IRF3/7 signaling, IFN production and antiviral activity may be attributed to the virus and cell type-specific IFN regulation, reduced expression of IRF3 and inhibitory effect on IRF3 by IRF3-CL in EV71 infected RD cells ectopically expressing IKK ϵ v1. Moreover, the absent effect of IKK ϵ v1 on the read-outs assayed strengthens our finding that IKK ϵ v2 presents higher activities in IRF7 interaction and activation than IKK ϵ v1 and promotes IFN and ISG productions to establish host antiviral responses.

Supplementary Material Table 1: Genes modulated in IFN treated and/or EV71 or CA16 infected RD cells

Gene Symbol	Gene Description	Average fold change*
		EV71 infected vs. Mock (RD cell)
IFNB1	Interferon, beta 1, fibroblast	10.42
IRF1	Interferon regulatory factor 1	-2.70
IRF2	Interferon regulatory factor 2	-2.53
IRF3	Interferon regulatory factor 3	-3.91
IRF5	Interferon regulatory factor 5	-3.87
IRF7	Interferon regulatory factor 7	1.05
IRF9	Interferon regulatory factor 9	-1.05

* Numbers denote fold increase, (-) values denote fold reduction.
Relative quantification of gene expression was determined using the 2^{- $\Delta\Delta C_t$} method.

Supporting Figure 6. Genes modulated in IFN treated and/or EV71 or CA16 infected RD cells. (Excerpt from Supplementary Material Table 1 of Zhang et al., 2014)

We have discussed why IKK ϵ v1 did not have effects on IRF3/7 signaling, IFN production and antiviral activity in the revised discussion section as follows.

Revised Discussion: (Page 19, Line 1 - Page 22, Line 2)

The innate immune responses provide an early phase defense against viral infections. Generally, virus triggers a cascade of signaling to product type I IFNs and proinflammatory cytokines⁵⁸. However, it is not the case for all of viruses because certain viral infections can subvert cellular IFN induction pathways⁵⁹. For example, influenza viral proteins PB1-F2 and PB2-S1 interact with MAVS to inhibit IFN induction^{60,61} and hepatitis C virus cleaves MAVS to interfere IFN production⁶². Moreover, it has been demonstrated that dengue virus can induce IFN β and ISG production in the brain; in contrast, IFNs are not detectable in infected dendritic cells^{63,64}. It suggested that IFN induction is regulated in a cell type-specific manner.

EV71 has been demonstrated to regulate IFN β induction by affecting the pathways mediated by RIG-1/MDA5 and TLR upon EV71 infection. EV71 3C protease (3C^{pro}) was demonstrated to suppress IFN signaling by interrupting the RIG-1-IFN promoter-stimulating factor 1 (IPS1) interaction, and with nucleus translocation of IRF3⁶⁵. Other studies have reported that 3C^{pro} degrades RIG-1 and cleaves adaptor TRIF to overcome IFN production in EV71 infected cells^{66,67} and EV71 2A protease (2A^{pro}) targets MAVS and cleaves MDA5 which is responsible for IRF3 activation to inhibit IFN production^{68,69}. It is worth noting that these different regulations are demonstrated in different cell lines under different experimental conditions. On the other hand, cellular miRNAs also play roles in EV71-induced innate immune response. IRAK1 and TRAF6 proteins were reduced by EV71 induced miRNA-146a and resulted in suppression of IFN β production in EV71 infected cells³. miR-526a targeting cylindromatosis (CYLD) was demonstrated to stimulate phosphorylations of IRF3, I κ B, and IKK ϵ , and downregulation of miR-526a in EV71 infection impaired IFN-I production⁷⁰. miR-548 known to regulate the host antiviral responses by directly targeting IFN- λ 1 expression was shown to be suppressed upon EV71 infection⁷¹. miR-302 cluster suppressed EV71-induced innate immune response via direct targeting of karyopherin α 2 (KPNA2)⁷². More recently, Duan et al. showed the level of miR-628-5p was increased after EV71 infection. miR-628-5p suppressed TRAF3, which mediates IRF3 and NF- κ B activation, to further affect IFN β production during EV71 infection⁷³. Taken together, IFN induction is regulated in a virus- and cell type-specific manner and IFN β expression in EV71 infection is complicated and tightly regulated.

The host IFN pathway response to EV71 has been reported in which IRF3 expression is significant reduced while IRF7 is unchanged in EV71 infected RD cells⁷⁴. Furthermore, IRF3-CL, an isoform of IRF3, is ubiquitously expressed in all cell lines and acts as a negative regulator of IRF3 via dimerization with IRF3 in the presence of IKK ϵ overexpression⁷⁵. It is possible that IRF3-CL plays an inhibitory role in ectopically IKK ϵ isoform-expressing RD cells upon EV71 infection, however, we did not measure IRF3-CL expression in our study. Hence, the limited effects of IKK ϵ v1 on IRF3/7 signaling, IFN production and antiviral activity observed in our study may be attributed to the virus and cell type-specific IFN regulation, reduced expression of IRF3 and inhibitory effect on IRF3 by IRF3-CL in EV71 infected RD cells ectopically expressing IKK ϵ v1. Moreover, the absent effect of IKK ϵ v1 on immune and antiviral activities strengthens our finding that IKK ϵ v2 presents higher

activities in IRF7 interaction and activation compared with IKK ϵ v1 and IKK ϵ v2 possessing higher kinase activity promotes IFN and ISG production to establish host antiviral responses by interacting with, phosphorylating and activating IRF7.

Revised References:

58 Tailor, P., Tamura, T. & Ozato, K. IRF family proteins and type I interferon induction in dendritic cells. *Cell Res* **16**, 134-140, doi:10.1038/sj.cr.7310018 (2006).

59 Garcia-Sastre, A. Ten Strategies of Interferon Evasion by Viruses. *Cell Host Microbe* **22**, 176-184, doi:10.1016/j.chom.2017.07.012 (2017).

60 Varga, Z. T., Grant, A., Manicassamy, B. & Palese, P. Influenza virus protein PB1-F2 inhibits the induction of type I interferon by binding to MAVS and decreasing mitochondrial membrane potential. *Journal of virology* **86**, 8359-8366, doi:10.1128/JVI.01122-12 (2012).

61 Yamayoshi, S., Watanabe, M., Goto, H. & Kawaoka, Y. Identification of a Novel Viral Protein Expressed from the PB2 Segment of Influenza A Virus. *Journal of virology* **90**, 444-456, doi:10.1128/JVI.02175-15 (2016).

62 Li, X. D., Sun, L., Seth, R. B., Pineda, G. & Chen, Z. J. Hepatitis C virus protease NS3/4A cleaves mitochondrial antiviral signaling protein off the mitochondria to evade innate immunity. *Proceedings of the National Academy of Sciences of the United States of America* **102**, 17717-17722, doi:10.1073/pnas.0508531102 (2005).

63 Al-Shujairi, W. H. *et al.* Intracranial Injection of Dengue Virus Induces Interferon Stimulated Genes and CD8⁺ T Cell Infiltration by Sphingosine Kinase 1 Independent Pathways. *PloS one* **12**, e0169814, doi:10.1371/journal.pone.0169814 (2017).

64 Rodriguez-Madoz, J. R., Bernal-Rubio, D., Kaminski, D., Boyd, K. & Fernandez-Sesma, A. Dengue virus inhibits the production of type I interferon in primary human dendritic cells. *Journal of virology* **84**, 4845-4850, doi:10.1128/JVI.02514-09 (2010).

65 Lei, X. *et al.* The 3C protein of enterovirus 71 inhibits retinoid acid-inducible gene I-mediated interferon regulatory factor 3 activation and type I interferon responses. *Journal of virology* **84**, 8051-8061, doi:10.1128/JVI.02491-09 (2010).

66 Barral, P. M., Sarkar, D., Fisher, P. B. & Racaniello, V. R. RIG-I is cleaved during picornavirus infection. *Virology* **391**, 171-176, doi:10.1016/j.virol.2009.06.045 (2009).

- 67 Wang, C. *et al.* Differential Regulation of TLR Signaling on the Induction of Antiviral Interferons in Human Intestinal Epithelial Cells Infected with Enterovirus 71. *PloS one* **11**, e0152177, doi:10.1371/journal.pone.0152177 (2016).
- 68 Feng, Q. *et al.* Enterovirus 2Apro targets MDA5 and MAVS in infected cells. *Journal of virology* **88**, 3369-3378, doi:10.1128/JVI.02712-13 (2014).
- 69 Kuo, R. L., Kao, L. T., Lin, S. J., Wang, R. Y. & Shih, S. R. MDA5 plays a crucial role in enterovirus 71 RNA-mediated IRF3 activation. *PloS one* **8**, e63431, doi:10.1371/journal.pone.0063431 (2013).
- 70 Xu, C. *et al.* Downregulation of microRNA miR-526a by enterovirus inhibits RIG-I-dependent innate immune response. *Journal of virology* **88**, 11356-11368, doi:10.1128/JVI.01400-14 (2014).
- 71 Li, Y. *et al.* MicroRNA-548 down-regulates host antiviral response via direct targeting of IFN-lambda1. *Protein Cell* **4**, 130-141, doi:10.1007/s13238-012-2081-y (2013).
- 72 Peng, N. *et al.* MicroRNA-302 Cluster Downregulates Enterovirus 71-Induced Innate Immune Response by Targeting KPNA2. *J Immunol* **201**, 145-156, doi:10.4049/jimmunol.1701692 (2018).
- 73 Li, D. *et al.* MicroRNA-628-5p Facilitates Enterovirus 71 Infection by Suppressing TRAF3 Signaling. *Cell Mol Immunol*, doi:10.1038/s41423-020-0453-4 (2020).
- 74 Zhang, W., Zhang, L., Wu, Z. & Tien, P. Differential interferon pathway gene expression patterns in Rhabdomyosarcoma cells during Enterovirus 71 or Coxsackievirus A16 infection. *Biochem Biophys Res Commun* **447**, 550-555, doi:10.1016/j.bbrc.2014.04.021 (2014).
- 75 Li, C., Ma, L. & Chen, X. Interferon regulatory factor 3-CL, an isoform of IRF3, antagonizes activity of IRF3. *Cell Mol Immunol* **8**, 67-74, doi:10.1038/cmi.2010.55 (2011).

4: Similarly it has been shown before that IKK ϵ can also interact with IRF3 (even at endogenous levels), so it is unclear why the authors do not see this in their IP with overexpressed proteins. In Supp Fig 2a: have V5-IRF3 and V5-IRF7, which are shown in two different panels been detected on the same membrane/film and can thus be compared in intensity, or not?

Author reply:

Thank you for your comment.

At the beginning of this study, we checked the interaction of IRF3 or IRF7 and IKK ϵ isoforms to verify the downstream effector of IKK ϵ isoforms. In the co-IP

experiment, we loaded equal amount of lysate for probing anti-V5 and anti-Flag respectively to avoid the bias from the stripping and re-probe. Furthermore, to perform protein electrophoresis and protein transfer under the same condition, we put all co-IP samples on the same membrane indicated in Supporting Figure 7 (left membrane) and the V5-IRF7 and V5-IRF3 lysate controls were loaded on different membranes due to the space limitation as shown in Supporting Figure 7. After protein transfer, the V5-IRF7 and V5-IRF3 panels were “back-to-back” incubated with antibodies in the same hybridization bag. That is the reason we showed V5-IRF7 and V5-IRF3 lysates in two panels. Importantly, in Suppl Fig. 2a, the Flag-IKK ϵ in IP-Flag panel (the second panel) showed comparable intensities but none V5-IRF3 bound to Flag-tagged IKK ϵ isoforms (the top panel). Although the V5-IRF7 and V5-IRF3 lysate controls were loaded on separate membranes, the protein loading and procedures of transfer, hybridization and development were identical. Therefore, the intensities of V5-IRF3 and V5-IRF7 in the Lysate panel of Supplementary Figure 2a are comparable.

In order to understand why IKK ϵ can not associate with IRF3 in our study, we first re-checked our V5-IRF3 construct by Sanger sequencing. After alignment, we confirmed that the sequence of our V5-IRF3 (Supporting Figure 8) is consistent with NCBI reference sequence NM_001571. It suggests that the accuracy of V5-IRF3 should not be concerned. Moreover, we looked up references for explaining why we did not co-IP IRF3 by IKK ϵ isoforms. In order to characterize functions of IKK ϵ isoforms, we needed to ectopically express IKK ϵ isoforms while it is impossible to design isoform-specific siRNAs. (Please refer to the description in last response letter.) However, previous studies have reported IRF3-CL, an isoform of IRF3, forms a heterodimer with IRF3 when IKK ϵ is overexpressed in HEK293. Furthermore, IRF3-CL may function as a negative regulator of IRF3 by inhibiting IKK ϵ -mediated nuclear translocation of IRF3 (Li et al., 2011). It is possible that once we ectopically expressed IKK ϵ , IRF3-CL associated with IRF3 and disrupted the interaction of IRF3 and IKK ϵ in HEK293. As a result, neither the interaction between IKK ϵ isoforms and IRF3 nor the promoter activation driven by IRF3 can be observed.

**Supporting Fig 7. The sample order of Supplementary Figure 2a**

**Supporting Figure 8. The sequence alignment of V5-IRF3 and NM_001571**

5. Despite two reviewers asking for this, the authors have not addressed the question about the mechanism for isotype switching. Not experimentally, and not much on this has been added to the discussion either. If different viruses induce this, it would seem quite obvious to test whether this is an IFN-mediated effect?

**Author reply**:

Thanks for your suggestions.

Up to our knowledge, there is no report for IKKε isoform switching. In our study, we demonstrated IKKε switches not only in EV71 infection but also in CVB3 and HSV-1 infections. It suggests that the mechanism of IKKε isoform switching may be universal, and it might be worthy of further investigation in

the future. Hence, we have extended the possible regulatory mechanisms in the revised manuscript.

We have added the description in the Discussion section of the revised manuscript as follows:

Revised Discussion: (Page 23, Line 11 – Page 24, Line 8)

Alteration of host RNA splicing is a common feature in virus infections, and the most frequently alternative splicing is exon skipping⁷⁹. One mechanism of the alternative splicing under virus infections is directly caused by a viral manipulation of splicing machinery. 3D polymerase (3D^{pol}) of EV71 entered the nucleus and targeted the central pre-mRNA processing factor 8 (Prp8) to block pre-mRNA splicing and mRNA synthesis⁸⁰. The other mechanism is related to virus-responsive regulation on splicing factors, which regulates the splicing of key players of the antiviral innate immunity. RIG-1 and stimulator of interferon genes (STING) splice variants were upregulated upon viral infection and strongly inhibited RIG-1 and STING signaling pathways, respectively^{81,82}. In this study, we found that IKK ϵ isoform switching took place in EV71, CVB3 and HSV-1 infections, and such a regulated isoform switching resulted in the controlled production of IFN β . Whether splicing machinery is altered and IFN is involved in the regulation of IKK ϵ isoform switching upon viral infection remain to be elucidated.

Revised References:

79 Black, D. L. Mechanisms of alternative pre-messenger RNA splicing. *Annu Rev Biochem* **72**, 291-336, doi:10.1146/annurev.biochem.72.121801.161720 (2003).

80 Liu, Y. C. *et al.* Cytoplasmic viral RNA-dependent RNA polymerase disrupts the intracellular splicing machinery by entering the nucleus and interfering with Prp8. *PLoS Pathog* **10**, e1004199, doi:10.1371/journal.ppat.1004199 (2014).

81 Gack, M. U. *et al.* Roles of RIG-I N-terminal tandem CARD and splice variant in TRIM25-mediated antiviral signal transduction. *Proceedings of the National Academy of Sciences of the United States of America* **105**, 16743-16748, doi:10.1073/pnas.0804947105 (2008).

82 Rodriguez-Garcia, E. *et al.* TMEM173 Alternative Spliced Isoforms Modulate Viral Replication through the STING Pathway. *Immunohorizons* **2**, 363-376, doi:10.4049/immunohorizons.1800068 (2018).

Minor issues:

1. In Supp Figure 2: there are two actin blots shown in the IKKε IP. Duplicated by mistake?

Author reply:

Thank you for your notice.

After double check, we did not find two actin blots in Suppl Figure 2. The Suppl Figure 2 was showed as follows:

Revised Supplementary Figure 2a

Supplementary Fig. 2: IKKε v2 transcriptionally induces ISGs expressions in EV71 infection through IRF7.

a. IKKε preferentially binds to IRF7. Each Flag-IKKε isoform was co-transfected with V5-IRF3 or V5-IRF7 in HEK293T cells. Flag-IKKε was immunoprecipitated with anti-Flag beads. V5-IRF3 (left panel) and V5-IRF7 (right panel) were detected by anti-V5 antibody.

2. What is different about Figure 3a and 3b?

Author reply:

Thank you for your comment.

The difference of phospho-IRF7 in Figure 3a and 3b resulted from different antibodies. To understand how IKKε v2 activates IRF7 step by step, we first addressed whether ubiquitin plays a role on IRF7 phosphorylation by using anti-phospho-Serine antibody (Cell signaling, 9631). As the data showed in

Figure 3a, the phosphorylation level of Serine on Flag-IRF7 was increased in IKK ϵ v2 transfectants in the presence of ubiquitin compared to IKK ϵ v1 and IKK ϵ v3 transfectants. Next, we addressed whether IKK ϵ v2 activates IRF7 by using anti-phospho-IRF7 (Ser471/472) specific antibody (Cell signaling, 5184). Ser471/472 is considered a phosphorylation site by IKK ϵ and is characterized as a vital residue in IRF7 activation. Figure 3b indicated that IKK ϵ v2 greatly activates IRF7 and then induces IRF7 translocation in Figure 3d.

3. Supplementary Figure 5: this pulldown experiment does not seem to be very well controlled.

Author reply:

Thank you for your kind suggestion.

We apologize for our carelessness. We revised the Supplementary Figure 5d as follows:

Revised Figure 5.

Supplementary Fig. 5: The interaction of IKK isoforms and IRF7.

d. The direct interaction of IRF7 and IKK ϵ isoforms. Each Flag-IKK ϵ isoform was purified by anti-Flag beads from Flag-IKK ϵ -expressed HEK293T cells and incubated with 1 μ g of His-IRF7, which was purified from *E. coli*. for 2 hours at 4°C. After washing for three times, the bound His-IRF7 was analyzed by Western blotting with anti-His antibody.

4. Figure 1c: is the second V1 panel a lower exposure? If so, please label this clearly.

Author reply:

Thank you for the kind suggestions.

We revised the Figure 1c as follows:

Revised Figure legend: (Page 46, Line 10 – Page 46, Line 14)

Fig. 1: EV71 infection triggers IKK ϵ isoform switching.

c. IKK ϵ isoform switching is confirmed by Western blotting. The Click-iT AHA assay was performed to measure newly synthesized IKK ϵ v1 and v2. The synthesis of IKK ϵ v2 was increased while IKK ϵ v1 was decreased in EV71 infection. The upper panel of IP-Streptavidin is a long exposure while the lower panel is a short exposure.

References:

- Al-Shujairi, W.H., Clarke, J.N., Davies, L.T., Alsharifi, M., Pitson, S.M., and Carr, J.M. (2017). Intracranial Injection of Dengue Virus Induces Interferon Stimulated Genes and CD8+ T Cell Infiltration by Sphingosine Kinase 1 Independent Pathways. *PLoS One* 12, e0169814.
- Barral, P.M., Sarkar, D., Fisher, P.B., and Racaniello, V.R. (2009). RIG-I is cleaved during picornavirus infection. *Virology* 391, 171-176.
- Broz, P., and Monack, D.M. (2013). Newly described pattern recognition receptors team up against intracellular pathogens. *Nat Rev Immunol* 13, 551-565.
- Chen, P., Song, Z., Qi, Y., Feng, X., Xu, N., Sun, Y., Wu, X., Yao, X., Mao, Q., Li, X., *et al.* (2012). Molecular determinants of enterovirus 71 viral entry: cleft around GLN-172 on VP1 protein interacts with variable region on scavenger receptor B 2. *J Biol Chem* 287, 6406-6420.
- Feng, Q., Langereis, M.A., Lork, M., Nguyen, M., Hato, S.V., Lanke, K., Emdad, L., Bhoopathi, P., Fisher, P.B., Lloyd, R.E., *et al.* (2014). Enterovirus 2Apro targets MDA5 and MAVS in infected cells. *J Virol* 88, 3369-3378.
- Garcia-Sastre, A. (2017). Ten Strategies of Interferon Evasion by Viruses. *Cell Host Microbe* 22, 176-184.
- Hiscott, J., Nguyen, T.L., Arguello, M., Nakhaei, P., and Paz, S. (2006). Manipulation of the nuclear factor-kappaB pathway and the innate immune response by viruses. *Oncogene* 25, 6844-6867.
- Ho, B.C., Yu, I.S., Lu, L.F., Rudensky, A., Chen, H.Y., Tsai, C.W., Chang, Y.L., Wu, C.T., Chang, L.Y., Shih, S.R., *et al.* (2014). Inhibition of miR-146a prevents enterovirus-induced death by restoring the production of type I interferon. *Nat Commun* 5, 3344.
- Holm, C.K., Paludan, S.R., and Fitzgerald, K.A. (2013). DNA recognition in immunity and disease. *Curr Opin Immunol* 25, 13-18.

Kuo, R.L., Kao, L.T., Lin, S.J., Wang, R.Y., and Shih, S.R. (2013). MDA5 plays a crucial role in enterovirus 71 RNA-mediated IRF3 activation. *PLoS One* 8, e63431.

Lei, X., Liu, X., Ma, Y., Sun, Z., Yang, Y., Jin, Q., He, B., and Wang, J. (2010). The 3C protein of enterovirus 71 inhibits retinoid acid-inducible gene I-mediated interferon regulatory factor 3 activation and type I interferon responses. *J Virol* 84, 8051-8061.

Li, C., Ma, L., and Chen, X. (2011). Interferon regulatory factor 3-CL, an isoform of IRF3, antagonizes activity of IRF3. *Cell Mol Immunol* 8, 67-74.

Li, D., Chen, S., Zhang, W., Zhang, C., Sun, T., Du, Y., Ding, R., Gao, Y., Jin, Y., and Duan, G. (2020). MicroRNA-628-5p Facilitates Enterovirus 71 Infection by Suppressing TRAF3 Signaling. *Cell Mol Immunol*.

Li, X.D., Sun, L., Seth, R.B., Pineda, G., and Chen, Z.J. (2005). Hepatitis C virus protease NS3/4A cleaves mitochondrial antiviral signaling protein off the mitochondria to evade innate immunity. *Proc Natl Acad Sci U S A* 102, 17717-17722.

Li, Y., Xie, J., Xu, X., Wang, J., Ao, F., Wan, Y., and Zhu, Y. (2013). MicroRNA-548 down-regulates host antiviral response via direct targeting of IFN-lambda1. *Protein Cell* 4, 130-141.

Mahmood, N., and Xie, J. (2015). An endogenous 'non-specific' protein detected by a His-tag antibody is human transcription regulator YY1. *Data Brief* 2, 52-55.

Peng, N., Yang, X., Zhu, C., Zhou, L., Yu, H., Li, M., Lin, Y., Wang, X., Li, Q., She, Y., *et al.* (2018). MicroRNA-302 Cluster Downregulates Enterovirus 71-Induced Innate Immune Response by Targeting KPNA2. *J Immunol* 201, 145-156.

Perez-Ruiz, M., Navarro-Mari, J.M., Palacios Del Valle, E., and Rosa-Fraile, M. (2003). Human rhabdomyosarcoma cells for rapid detection of enteroviruses by shell-vial assay. *J Med Microbiol* 52, 789-791.

Rodriguez-Madoz, J.R., Bernal-Rubio, D., Kaminski, D., Boyd, K., and Fernandez-Sesma, A. (2010). Dengue virus inhibits the production of type I interferon in primary human dendritic cells. *J Virol* 84, 4845-4850.

Saul, V.V., Niedenthal, R., Pich, A., Weber, F., and Schmitz, M.L. (2015). SUMO modification of TBK1 at the adaptor-binding C-terminal coiled-coil domain contributes to its antiviral activity. *Biochim Biophys Acta* 1853, 136-143.

Taylor, P., Tamura, T., and Ozato, K. (2006). IRF family proteins and type I interferon induction in dendritic cells. *Cell Res* 16, 134-140.

Takeda, K., and Akira, S. (2005). Toll-like receptors in innate immunity. *Int Immunol* *17*, 1-14.

Takeuchi, O., and Akira, S. (2010). Pattern recognition receptors and inflammation. *Cell* *140*, 805-820.

Varga, Z.T., Grant, A., Manicassamy, B., and Palese, P. (2012). Influenza virus protein PB1-F2 inhibits the induction of type I interferon by binding to MAVS and decreasing mitochondrial membrane potential. *J Virol* *86*, 8359-8366.

Wang, C., Ji, L., Yuan, X., Jin, Y., Cardona, C.J., and Xing, Z. (2016). Differential Regulation of TLR Signaling on the Induction of Antiviral Interferons in Human Intestinal Epithelial Cells Infected with Enterovirus 71. *PLoS One* *11*, e0152177.

Xu, C., He, X., Zheng, Z., Zhang, Z., Wei, C., Guan, K., Hou, L., Zhang, B., Zhu, L., Cao, Y., *et al.* (2014). Downregulation of microRNA miR-526a by enterovirus inhibits RIG-I-dependent innate immune response. *J Virol* *88*, 11356-11368.

Yamayoshi, S., Watanabe, M., Goto, H., and Kawaoka, Y. (2016). Identification of a Novel Viral Protein Expressed from the PB2 Segment of Influenza A Virus. *J Virol* *90*, 444-456.

Zhang, W., Zhang, L., Wu, Z., and Tien, P. (2014). Differential interferon pathway gene expression patterns in Rhabdomyosarcoma cells during Enterovirus 71 or Coxsackievirus A16 infection. *Biochem Biophys Res Commun* *447*, 550-555.

Zhou, F., Wan, Q., Lu, J., Chen, Y., Lu, G., and He, M.L. (2019). Pim1 Impacts Enterovirus A71 Replication and Represents a Potential Target in Antiviral Therapy. *iScience* *19*, 715-727.

Reviewers' Comments:

Reviewer #3:

Remarks to the Author:

The authors have provided a very comprehensive answer to my queries.

1. The authors have addressed the discrepancies with the findings of Koop et al. well and now included a reference to the paper.
2. The authors state that they sequence verified the different flag- and V5-tagged expression constructs again and found no issues that could explain the different apparent molecular weights of the V5- versus the flag-tagged versions. This is good. They also provide some speculation for potential reasons that could explain this observation, mainly centred on different post-translational modifications that might occur with different IKKe versions and different tags. This is possible, but also justifies concerns about potential overexpression artefacts that could affect the results.
3. The authors now provide a lengthy discussion/speculation as to why they do not observe well-established V1-mediated effects and interactions, mostly relating to immune evasion mechanisms in virus-infected cells (which makes sense) and the cell line they used for infection models expressing an IRF3 inhibitor. I am not sure this fully explains all the data, seeing as several experiments have been carried out in HEK293T cells and with overexpressed signalling molecules in uninfected cells. But at the same time there may not be anything they can do about this, if these are the results they obtained. It may not be necessary to include the entire explanation/speculation provided for review purposes into the final manuscript.
4. The authors have expanded the discussion about potential mechanisms of isotype switching as requested.